# CAT-Video: Corruption-Aware Training for Robust Video Diffusion Models

## Abstract

Latent Video Diffusion Models (LVDMs) have achieved state-of-the-art generative quality for image and video generation; however, they remain brittle under noisy conditioning, where small perturbations in text or multimodal embeddings can cascade over timesteps and cause semantic drift. Existing corruption strategies from image diffusion (Gaussian, Uniform) fail in video settings because static noise disrupts temporal fidelity. In this paper, we propose **CAT-Video**, a corruption-aware training framework with structured, data-aligned noise injection tailored for video diffusion. Our two operators—*Batch-Centered Noise Injection (BCNI)* and *Spectrum-Aware Contextual Noise (SACN)* align perturbations with batch semantics or spectral dynamics to preserve coherence. CAT-Video yields substantial gains: BCNI reduces FVD by **31.9%** on WebVid-2M, MSR-VTT, and MSVD, while SACN improves UCF-101 by **12.3%**, outperforming Gaussian, Uniform, and even large diffusion baselines like DEMO (2.3B) and Lavie (3B) despite training on **5×** less data. Ablations confirm the unique value of low-rank, data-aligned noise, and theory establishes why these operators tighten robustness and generalization bounds. CAT-Video thus sets a new framework for robust video diffusion, and our experiments show that it can also be extended to autoregressive generation and multimodal video understanding LLMs.

## 1 Introduction

Diffusion models have revolutionized generative modeling across modalities, achieving state-of-the-art performance in image (Ho et al., 2020; Song et al., 2021b), audio (Liu et al., 2023; Huang et al., 2023), and video generation (Ho et al., 2022; Singer et al., 2023). By iteratively denoising latent variables using learned score functions (Wang et al., 2024a; Zhu et al., 2023), these models offer superior sample diversity, stability, and fidelity compared to adversarial approaches (Dhariwal & Nichol, 2021; Cao et al., 2024). In video generation, latent video diffusion models (LVDMs) (Wu et al., 2023; Zhang et al., 2025; Yang et al., 2025) have emerged as an efficient paradigm, compressing high-dimensional video data into compact latent spaces using pretrained autoencoders (Khachatryan et al., 2023; Ni et al., 2024). These latent representation are conditioned on text via vision-language models like CLIP (Radford et al., 2021), enabling scalable and semantically grounded text-to-video (T2V) generation.

However, LVDMs are highly vulnerable to corrupted inputs (Zhu et al., 2024; Gu et al., 2025), which refer to imperfect, noisy, or weakly aligned text prompts and multimodal embeddings that condition the diffusion process. We implement Gaussian corruption by adding independent noise drawn from $\mathcal{N}(0, \rho^2 I)$ to each token embedding and Uniform corruption by sampling per-coordinate noise from $\mathcal{U}(-\rho, \rho)$. We sweep $\rho \in [0.025, 0.20]$, a standard range for conditional embedding perturbation (Chen et al., 2024), to ensure consistency with prior multimodal robustness work across all experiments. This sensitivity is critical because, in video generation, corrupted conditioning not only degrades individual frames but also accumulates across timesteps, leading to cascading errors that severely undermine visual fidelity and temporal coherence (Liu et al., 2024c; Guo et al., 2025). Unlike classification (Graf et al., 2025; Jain et al., 2024a) or retrieval (Chen & Guo, 2023) models, where label noise induces bounded degradation, diffusion models suffer recursive error amplification due to their iterative structure (Gu et al., 2025; Na et al., 2024; Gao et al., 2023; Jain et al., 2024b). This fragility manifests in semantic drift, loss of temporal coherence, and degraded

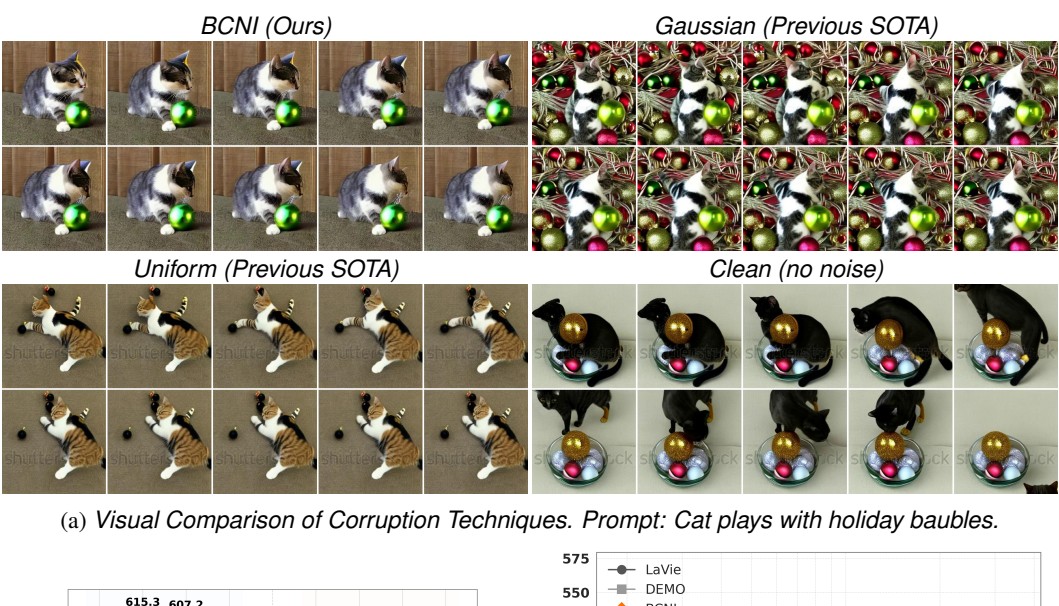

(a) *Visual Comparison of Corruption Techniques. Prompt: Cat plays with holiday baubles.*

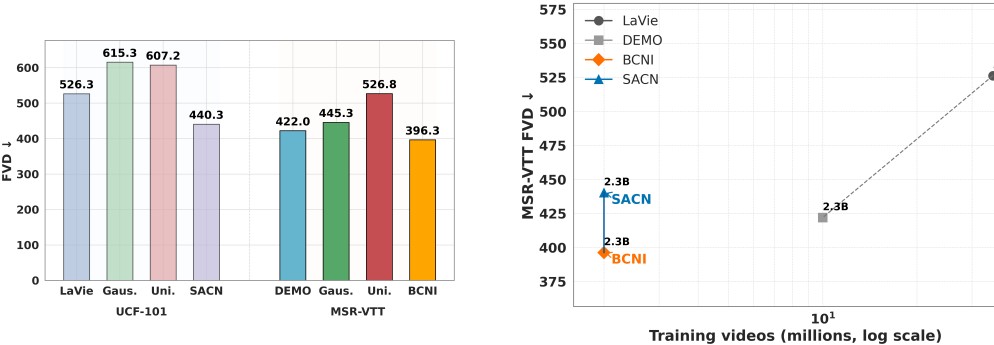

(b) *FVD comparison on Benchmarks.*

(c) *Efficiency: FVD vs. training videos.*

Figure 1: **Overview.** We introduce structured corruption (*BCNI, SACN*) and compare to the previous corruption SOTA for images (*Gaussian, Uniform*) and the *Clean* baseline. We show visual generations in (a) and summarize quantitative comparisons to SOTA in (b, c).

multimodal alignment (Khrapov et al., 2024; Popov et al., 2025), especially in video settings where frame-to-frame consistency is essential. This effect is visually evident in Figure 1(a), where Gaussian and Uniform corruptions cause noticeable semantic drift and visual degradation with respect to the prompt.

Existing defenses, however, are critically underprepared for these conditions. Corruption techniques developed for image diffusion (Chen et al., 2024; Daras et al., 2023) fail to address temporal entanglement and the risk of cumulative semantic drift unique to video generation. To bridge this gap, we propose **CAT-Video**, a corruption-aware training framework that introduces novel structured perturbations during pretraining, explicitly tailored for LVDMs. Theoretically, controlled corruption increases conditional entropy (Song et al., 2023; Chen et al., 2024), reduces the 2-Wasserstein distance to the target distribution, and smooths the conditional score manifold (Goldblum et al., 2020), yielding improved robustness, diversity, and generalization. While such results are established in static images, video generation poses additional complexity: small conditioning errors propagate and amplify across multiple denoising steps. In Appendix B.1–B.8, we extend entropy, Wasserstein, and score-drift bounds to the sequential setting, proving that low-rank corruption explicitly controls cumulative error across frames and enforces Lipschitz continuity along the temporal manifold—guarantees unattainable in image-only analyses.

This paper presents **CAT-Video**, a corruption-aware training framework for LVDMs, showing that structured perturbations tailored to video-specific fragilities can substantially improve robustness and coherence under noisy, real-world conditions. Specifically, we find that existing corruption strategies

from image diffusion collapse in video settings, where conditioning noise compounds across time. To address this, we propose two low-rank perturbation techniques: *Batch-Centered Noise Injection (BCNI)* and *Spectrum-Aware Contextual Noise (SACN)*. BCNI perturbs embeddings along their deviation from the batch mean, acting as a Mahalanobis-scaled regularizer that increases conditional entropy only along semantically meaningful axes (Verma & Branson, 2015; Xu et al., 2020). SACN injects noise along dominant spectral modes, targeting low-frequency, globally coherent semantics. Both methods enforce Lipschitz continuity and reduce denoising error bounds (Chen et al., 2023; Yang et al., 2024), yielding better results as visualized in Figure 1(a).

Unlike prior image corruption SOTA methods (Gaussian, Uniform) (Chen et al., 2024) which inject static conditioning noise and often distort temporal coherence, our structured corruptions maintain fidelity by aligning perturbations with batch semantics (BCNI) or spectral dynamics (SACN). Notably, these lightweight strategies achieve lower FVD than much larger diffusion baselines such as LaVie (Wang et al., 2024b) and DEMO (Ruan et al., 2024), despite those models using 3B parameters and training on over five times more data (10M videos) as depicted in Figure 1(b) and (c). While our primary focus is on diffusion, we later verify that CAT's operator view also transfers to autoregressive generation(Deng et al., 2025) and multimodal video understanding(Liu et al., 2025), confirming its scalability beyond the diffusion setting. Together, BCNI and SACN reduce semantic drift, amplify conditioning diversity, and yield sharper motion and temporal consistency across diverse dataset regimes. Theoretically, we show that these methods shrink 2-Wasserstein distances to the real data manifold in a directionally aligned way, establishing a new, dataset-sensitive paradigm for robust LVDM training under imperfect multimodal supervision.

This work makes the following contributions: (i) we introduce **CAT-Video**, a corruption-aware training framework that enhances robustness in video diffusion through structured, data-aligned perturbations; Specifically, we design two novel operators—*Batch-Centered Noise Injection (BCNI)* and *Spectrum-Aware Contextual Noise (SACN)*—that preserve temporal fidelity by aligning noise with batch semantics or spectral dynamics; (ii) we demonstrate **strong empirical robustness**, with BCNI reducing FVD by **31.9%** on WebVid-2M, MSR-VTT, and MSVD, SACN improving UCF-101 by **12.3%**, and BCNI surpassing LaVie (3B) by **16%** on UCF-101 and DEMO (2.3B) by **6%** on MSR-VTT despite training on **5×** less data. We also validate **scalability** by extending CAT to autoregressive video generation (NOVA) and multimodal video understanding LLMs (PAVE), confirming model-agnostic robustness; and (iii) we provide a **theoretical analysis** showing that structured corruption tightens entropy, Wasserstein, and score-drift bounds, explaining why low-rank perturbations regularize temporal propagation and improve generalization.

## 2 METHOD

### 2.1 PRELIMINARIES: LATENT VIDEO DIFFUSION MODELS

LVDMs (Ho et al., 2022; Rombach et al., 2022; Luo et al., 2023; Zhang et al., 2023; Singer et al., 2023; Khachatryan et al., 2023) reverse a variance-preserving diffusion in a low-dimensional video latent space. A video $v \in \mathbb{R}^{F \times H \times W \times 3}$ is encoded by a pretrained autoencoder $E_v$ into

$$x_0 = E_v(v) \in \mathbb{R}^{F \times h \times w \times c}, \tag{1}$$

with $h \ll H$, $w \ll W$, and $c \gg 3$. The forward-noising process

$$q(x_t \mid x_0) = \mathcal{N}\big(\sqrt{\bar{\alpha}_t}\, x_0,\ (1 - \bar{\alpha}_t)\, I\big), \quad \bar{\alpha}_t = \prod_{s=1}^{t} \alpha_s, \tag{2}$$

follows the variance-preserving schedule (Sohl-Dickstein et al., 2015; Song et al., 2021b; Kingma et al., 2021). A U-Net $\epsilon_\theta(x_t, t, z)$ is trained to predict the added noise via

$$\mathcal{L}_{\text{diff}} = \mathbb{E}_{x_0, \epsilon, t, z} \big\| \epsilon - \epsilon_\theta(x_t, t, z) \big\|_2^2, \quad x_t = \sqrt{\bar{\alpha}_t}\, x_0 + \sqrt{1 - \bar{\alpha}_t}\, \epsilon, \tag{3}$$

conditioned on

$$z = f(p) \in \mathbb{R}^D \tag{4}$$

from a CLIP-based text encoder, as in DDPM (Ho et al., 2020). This yields efficient, high-quality conditional video synthesis in the latent space.

## 2.2 MOTIVATION

LVDMs generate sequences by iteratively denoising a latent trajectory conditioned on text or embeddings. However, these conditioning signals are often imperfect—textual prompts may be ambiguous, and encoder outputs may contain semantic drift or noise. In video, such imperfections are not benign: small conditioning errors at early timesteps accumulate over the denoising chain, leading to compounding semantic misalignment and disrupted temporal coherence (Figure 1). While prior work in image diffusion (Chen et al., 2024; Daras et al., 2023; Gao et al., 2023) has shown that injecting modest corruption into conditioning can smooth score estimates and improve robustness, such methods ignore the temporal dependencies intrinsic to video.

Mimicking the compounding semantic drift introduced by imperfect conditioning signals, structured, data-aligned corruption during training serves as an effective inductive bias to regularize the model and enhance robustness. To test this, we introduce two novel corruption strategies tailored for video diffusion: *Batch-Centered Noise Injection (BCNI)* perturbs each conditioning embedding along its deviation from the batch mean—amplifying local conditional entropy in meaningful semantic directions—while *Spectrum-Aware Contextual Noise (SACN)* adds noise selectively along dominant spectral modes that correspond to low-frequency temporal motion. These perturbations are not arbitrary: they reflect the types of semantic variation and smooth transitions that naturally occur across frames. By training the score network $\varepsilon_\theta(x_t, t, z)$ to denoise under these structured corruptions, we regularize its Lipschitz behavior, expand the support of the conditional distribution $P_{X|z}$, and reduce the 2-Wasserstein distance to the true data manifold. This results in more temporally consistent, semantically faithful generations. Theoretically, we prove (Appendix B.1–B.8) that BCNI and SACN enjoy an $O(d)$ vs. $O(D)$ complexity gap over unstructured baselines, providing both theoretical and empirical justification for structured corruption as a key design principle in robust LVDMs.

## 2.3 NOISE INJECTION TECHNIQUES

Our two *core* corruption strategies, *Batch-Centered Noise Injection (BCNI)* and *Spectrum-Aware Contextual Noise (SACN)*, are defined by the operators:

$$\mathcal{C}_{\text{BCNI}}(z;\rho) = \rho \, \|z - \bar{z}\|_2 \, \left(2\mathcal{U}(0,1) - 1\right), \tag{5}$$

$$\mathcal{C}_{\text{SACN}}(z;\rho) = \rho \, U\left(\xi \odot \sqrt{s}\right) V^\top, \quad [U, s, V] = \text{SVD}(z), \ \xi_j \sim \mathcal{N}\left(0, e^{-j/D}\right). \tag{6}$$

In BCNI (Eq. 5), we perturb each embedding $z$ along its deviation from the batch mean $\bar{z}$ by sampling a uniform direction and scaling it by $\|z - \bar{z}\|_2$, thereby confining corruption to the $d$-dimensional semantic subspace spanned by batchwise deviations. For instance, in a batch of videos showing people walking, BCNI perturbs each sample toward variations in stride or pose common to the batch, reinforcing motion realism rather than introducing arbitrary noise. This procedure adaptively inflates local conditional entropy there while leaving the orthogonal complement untouched (Theorem B.18). Importantly, neither BCNI nor SACN introduces any learnable parameters or tunable components beyond the global corruption scale $\rho$, which is swept across a small grid $[0.025, 0.2]$ (Chen et al., 2024) and held fixed per experiment. By contrast, SACN (Eq. 6) restricts noise to the principal spectral modes of $z$ that encode low-frequency, globally coherent motions. For example, in videos of a moving car, SACN targets the car's global trajectory rather than fine-grained texture or background details. This reshapes $z$ into a $D \times D$ matrix and computes $[U, s, V] = \text{SVD}(z)$, then samples $\xi \sim \mathcal{N}\left(0, \text{diag}(e^{-j/D})\right)$ to emphasize lower-frequency directions, and finally sets $\mathcal{C}_{\text{SACN}}(z;\rho) = \rho \, U\left(\xi \odot \sqrt{s}\right) V^\top$, which leaves high-frequency details largely unperturbed and ensures the 2–Wasserstein radius grows as $O(\rho\sqrt{d})$ rather than $O(\rho\sqrt{D})$ (Theorem B.4). The noise weighting in SACN is fixed analytically using exponentially decaying variances, requiring no manual tuning or dataset-specific adjustment. Training the denoiser $\varepsilon_\theta(x_t, t, z)$ under these data-aligned, low-rank corruptions then enforces a tighter Lipschitz constant (Proposition B.10), accelerates mixing (Theorem B.7), and dramatically attenuates error accumulation across the $T$ reverse steps. **The theoretical implications of this low-rank corruption are provided in Appendix B.1.**

In addition to BCNI and SACN, we also evaluate four additional corruption baselines—Gaussian (GN), Uniform (UN), Temporal-Aware (TANI), and Hierarchical Spectral (HSCAN)—to isolate the value of semantic and spectral alignment (Figure 4): GN/UN injects noise equally across all $D$ dimensions, TANI follows only temporal gradients without reducing rank, and HSCAN mixes

fixed spectral bands without data-adaptive weighting (see Appendix A.1 for full definitions and motivations). We further introduce Token-Level Corruption (TLC), which applies swap, replace, add, remove, and perturb operations directly on text prompts during model training to probe linguistic robustness; see Appendix A.2 for details.

## 2.4 THEORETICAL ANALYSIS

Structured, low-rank corruption improves robustness by confining noise to a $d$–dimensional semantic subspace ($d \ll D$), yielding a universal $D/d$ complexity gap. **Proposition A.2** formally shows that the conditional entropy under corruption increases as $\frac{d}{2} \log(1 + \rho^2/\sigma_z^2)$—scaling with $d$ rather than $D$—which expands the effective support of the conditional distribution without oversmoothing. **Theorem A.4** further proves that the 2-Wasserstein radius of the corrupted embedding distribution grows as $O((\rho' - \rho)\sqrt{d})$ rather than $O((\rho' - \rho)\sqrt{D})$, implying that perturbations stay closer to the target manifold in high dimensions. These results, along with bounds on score drift (**Lemma A.5**) and generalization gaps (**Theorem A.28**), imply that CAT-Video enforces a tighter Lipschitz constant on the score network and smooths the learned score manifold—ensuring that nearby inputs yield stable, consistent outputs across diffusion steps.

Empirically, these theoretical gains translate to reduced temporal flickering and sharper motion trajectories, particularly visible in our VBench smoothness and human action scores (Figure 1), as well as FVD improvements across all datasets (Table 2). Smoother score manifolds directly reduce error accumulation over $T$ denoising steps, leading to more temporally coherent video generations. Additional theoretical support for faster convergence and mixing under structured corruption is provided in **Theorem A.9** (spectral gap improvement) and **Theorem A.7** (energy decay bound), both of which reinforce the practical utility of BCNI and SACN as principled inductive biases.

Both BCNI and SACN incur only lightweight overhead during training. Specifically, **BCNI** performs a single $O(BD)$ operation per batch—where $B$ is the batch size and $D$ is the embedding dimensionality—to compute each sample's deviation from the batch mean, followed by a scale-and-add perturbation. **SACN** involves a one-time $O(Dd)$ projection onto the top-$d$ principal spectral modes ($d \ll D$), which can be approximated or precomputed at initialization. These costs are negligible compared to the dominant $O(N_U D^2)$ complexity of the U-Net forward and backward passes, where $N_U$ denotes the number of U-Net parameters.

Empirically, we observe that enabling BCNI or SACN increases training runtime by less than 2% on a single H100 GPU with batch size $B = 64$ and embedding dimension $D = 768$. Full pseudocode for the CAT-Video training loop, including both noise injection and denoising steps, is provided in Algorithm 1.

## 3 EXPERIMENTS

We conducted a large-scale experimental study involving 73 LVDM variants trained under seven embedding-level and five token-level corruption strategies across four benchmark datasets. Our evaluations spanned 292 distinct training–testing configurations and leveraged a diverse metric suite, including FVD, FVMD, CMMD, SSIM, LPIPS, PSNR, VBench, and EvalCrafter. Structured corruptions (BCNI, SACN) consistently outperformed isotropic and uncorrupted baselines across datasets, metrics, and noise levels. BCNI yielded the greatest gains on caption-rich datasets by preserving semantic alignment and motion consistency, while SACN showed strong results on class-label data by enhancing low-frequency temporal coherence. These improvements were further supported by qualitative visualizations, benchmark comparisons, and ablations on guidance scales and diffusion sampling steps.

## 3.1 SETUP

To rigorously benchmark the impact of structured corruption on latent video diffusion, we train 73 distinct T2V models under varying corruption regimes. At the embedding level, we apply seven corruption strategies $\tau \in \mathcal{T} = \{\text{GN}, \text{UN}, \text{GAP}, \text{BCNI}, \text{TANI}, \text{SACN}, \text{HSCAN}\}$, each evaluated across six corruption magnitudes, resulting in 42 variants. Similarly, at the text level, we apply five token-level operations $\xi \in \Xi = \{\text{swap}, \text{replace}, \text{add}, \text{remove}, \text{perturb}\}$ across six noise ratios,

Table 1: **SOTA Diffusion Comparisons.** Structured corruption (BCNI, SACN) achieves competitive results on UCF-101 and MSR-VTT benchmarks with fewer videos.

| Model | MSR-VTT FVD↓ | UCF-101 FVD↓ | #Params | #Videos (train) |
|---|---|---|---|---|
| DEMO (Ruan et al., 2024) | 422 | 547.3 | ∼2.3B | ∼10M |
| VideoComposer (Wang et al., 2023b) | 456 | – | ∼1.7B | ∼10M |
| MagicVideo (Zhou et al., 2023) | 998 | 655.0 | ∼1.2B | ∼17M |
| Show-1 (Zhang et al., 2025) | 538 | – | ∼6B | ∼10M |
| ModelScopeT2V (Wang et al., 2023a) | 557 | 628.2 | ∼1.7B | ∼10M |
| ModelScopeT2V (Finetuned) (Wang et al., 2023a) | 536 | 612.5 | ∼1.7B | ∼10M |
| SimDA (Xing et al., 2024) | 550 | – | ∼1.1B | ∼10M |
| VideoFusion (Luo et al., 2023) | 550 | – | ∼2.59B | ∼10M |
| FreeNoise (Qiu et al., 2024) | 517 | – | ∼1.7B | ∼10M |
| PEEKABOO (Jain et al., 2024b) | 609 | – | ∼1.7B | ∼10M |
| Latte (Ma et al., 2025) | – | 478.0 | ∼674M | ∼25M |
| CMD (Yu et al., 2024) | – | 504.0 | ∼1.6B | ∼10.7M |
| Video LDM (Blattmann et al., 2023) | – | 656.5 | ∼1.3B | ∼11M |
| VideoGen (Li et al., 2023) | – | 554.0 | ∼1.7B | ∼10M |
| LaVie (Wang et al., 2024b) | – | 526.3 | ∼3B | ∼35M |
| EMU Video (Girdhar et al., 2025) | – | 606.2 | ∼8.6B | ∼34M |
| Make-A-Video (Singer et al., 2023) | – | 367.2 | ∼9.6B | ∼20M |
| Gaussian (Chen et al., 2024) | 445.3 | 615.3 | ∼2.3B | ∼2M |
| Uniform (Chen et al., 2024) | 526.8 | 599.5 | ∼2.3B | ∼2M |
| CAT-Video (BCNI) | 396.3 | 505.5 | ∼2.3B | ∼2M |
| CAT-Video (SACN) | 440.3 | 440.3 | ∼2.3B | ∼2M |

yielding 30 additional models. One uncorrupted baseline ($\rho = \eta = 0$) is also included, summing to 67 independently trained models. All experiments are conducted using the DEMO architecture (Ruan et al., 2024) and trained on the WebVid-2M train dataset split (Bain et al., 2021). Evaluation is performed across four canonical benchmarks: WebVid-2M (val) (Bain et al., 2021), MSR-VTT (Xu et al., 2016), UCF-101 (Soomro et al., 2012), and MSVD (Chen & Dolan, 2011), for a total of 292 corruption-aware training-evaluation runs. Further details on the text-video datasets, including the duration, resolution, and splits, are provided in App. Table 8. Also, the evaluation protocol for zero-shot cross-dataset T2V generation is provided in App. Table 9. Meanwhile full training details—including model architecture, loss functions, regularization terms, optimizer configuration, and sampling strategy—are provided in Appendix C. Performance is assessed using a broad suite of metrics that reflect both perceptual quality and pixel-level fidelity. We report FVD (Unterthiner et al., 2019) as our primary metric for evaluating overall generative quality and alignment. Additionally, we compute FVMD (Liu et al., 2024a) for motion distance, CMMD (Jayasumana et al., 2024) for semantic consistency, PSNR (Huynh-Thu & Ghanbari, 2008), SSIM (Wang et al., 2004), and LPIPS (Zhang et al., 2018) for low-level reconstruction fidelity, as well as VBench (Huang et al., 2024) and EvalCrafter (Liu et al., 2024b) metrics to assess fine-grained, human-aligned video quality. Finally, while our core experiments focus on diffusion, we also briefly verify CAT's scalability by applying it to autoregressive video generation (NOVA (Deng et al., 2025)) and multimodal video understanding (PAVE (Liu et al., 2025)), confirming that the same operator view transfers beyond diffusion. Full training configs, corruption schedules, and code will be released upon acceptance.

## 3.2 MODEL-DATASET EVALUATIONS

**SOTA Benchmarks.** Table 1 reports comparisons against leading diffusion models on MSR-VTT and UCF-101. Our corruption-aware methods consistently set new state-of-the-art. BCNI achieves the best MSR-VTT score (396.3 vs. 422 for DEMO, which is trained with ∼10M videos) while remaining competitive on UCF-101 (505.5 vs. 547.3). SACN further improves motion stability, delivering the lowest UCF-101 FVD (440.3) despite using only 2M training videos. In contrast, competing models typically require 10–35M videos to reach similar or worse performance. These results highlight the sample efficiency of structured corruption: by aligning injected noise with caption semantics, our approach enhances motion fidelity and temporal coherence at a fraction of the training scale. A broader evaluation with VBench and EvalCrafter is provided in Appendix Table 17.

**Diffusion FVD comparisons.** Across four video benchmarks, Table 2 shows that structured corruption outperforms the image-based SOTA Gaussian and uniform baselines, supporting our claim that respecting data structure improves semantic alignment in diffusion models. BCNI attains the best

Table 2: **Model-Dataset Evaluations.** FVD comparisons across noise ratios.

| Noise ratio (%) | WebVid-2M | | | | MSRVTT | | | | MSVD | | | | UCF101 | | | |
|---|---|---|---|---|---|---|---|---|---|---|---|---|---|---|---|---|
| | BCNI | SACN | Gaussian | Uniform | BCNI | SACN | Gaussian | Uniform | BCNI | SACN | Gaussian | Uniform | BCNI | SACN | Gaussian | Uniform |
| 2.5 | 521.24 | 438.19 | 506.56 | 522.36 | 539.93 | 440.28 | 595.08 | 541.80 | 587.59 | 511.24 | 654.73 | 575.76 | 505.54 | **440.28** | 674.62 | 651.64 |
| 5 | 502.45 | 467.93 | 572.67 | 443.22 | 564.00 | 507.88 | 664.45 | 543.46 | 599.44 | 554.20 | 740.79 | 580.59 | 508.13 | 480.29 | 659.27 | 599.53 |
| 7.5 | **360.32** | 500.92 | 441.69 | 574.35 | 441.31 | 502.69 | 468.79 | 639.83 | **374.34** | 535.55 | 485.30 | 695.59 | 554.73 | 504.89 | 648.41 | 742.18 |
| 10 | 378.87 | 467.14 | 417.60 | 444.71 | 414.49 | 506.20 | 445.29 | 526.28 | 374.52 | 555.61 | 452.82 | 551.99 | 523.93 | 455.65 | 615.28 | 607.23 |
| 15 | 475.01 | 466.18 | 400.29 | 525.22 | 515.12 | 446.78 | 464.91 | 605.27 | 610.38 | 574.29 | 458.69 | 662.51 | 926.35 | 446.78 | 672.25 | 643.22 |
| 20 | 456.14 | 518.43 | 451.67 | 454.79 | **396.35** | 500.23 | 565.83 | 559.93 | 504.35 | 572.48 | 479.63 | 550.73 | 921.69 | 526.23 | 677.13 | 642.74 |
| Clean | *520.32* | | | | *543.33* | | | | *602.39* | | | | *501.91* | | | |

Table 3: (a) SOTA autoregressive baselines vs. CAT, (b) Sensitivity analysis of diffusion models.

| (a) Autoregressive Baselines | | | | (b) Sensitivity (Diffusion) | | | | |
|---|---|---|---|---|---|---|---|---|
| **Model** | **MSR-VTT FVD↓** | **#Params** | **#Videos** | **Dataset** | BCNI | SACN | Gauss. | Uniform |
| MAGVIT (Yu et al., 2023a) | 698 | ∼473M | ∼20M | WebVid-2M | **69.2** | 84.7 | 93.4 | 101.5 |
| CogVideo (Chinese) (Hong et al., 2023) | 1294 | ∼9.4B | ∼5.4M | MSR-VTT | **61.5** | 59.1 | 88.6 | 95.7 |
| CAT-Video (BCNI) | 358.3 | ∼0.6B | ∼2M | MSVD | **72.3** | 89.5 | 112.8 | 109.3 |
| CAT-Video (SACN) | 361 | ∼0.6B | ∼2M | UCF-101 | 85.4 | **68.2** | 107.4 | 111.0 |

FVD on WebVid 2M (360.32 at 7.5%) and MSVD (374.34 at 7.5%), and it also leads on MSRVTT at a higher ratio (396.35 at 20%). SACN is strongest on UCF101 (440.28 at 2.5%). These trends match how the methods work: BCNI perturbs around batch statistics, keeping embeddings near the data manifold and avoiding arbitrary drift, while SACN preserves spatial and temporal relations that stabilize motion. In line with our theory, noise that follows data structure acts as a regularizer, lowers effective sample complexity, and improves generative stability, which in turn reduces FVD. The dataset specific winners are interpretable: appearance diverse sets that are caption-rich like WebVid 2M, MSRVTT, and MSVD benefit from batch centered corrections, whereas the action focused, class-labeled datasets such as UCF101 benefits from spatially aligned corruption. Overall, structured corruption improves robustness and semantic fidelity across diffusion benchmarks. Further ablations with additional embedding- and token-level corruption strategies, along with metrics such as SSIM, PSNR, LPIPS, FVMD for motion distance and CMMD for semantic consistency, reinforce this observation; full results are provided in Appendix Tables 12, 13, and Figure 4. For reproducibility, we also report mean ± std across three random seeds in Appendix Table 18.

**Sensitivity Analysis.** To assess robustness beyond raw FVD values, we compute a *sensitivity index* for each corruption strategy by linearly regressing FVD against corruption magnitude and combining the slope with residual variance. This measures how smoothly performance degrades as noise increases. Table 3(b) shows that BCNI achieves the lowest sensitivity on caption-rich, appearance-diverse datasets (WebVid, MSVD) and remains competitive on MSR-VTT, while SACN is most stable on class-labelled, motion-heavy benchmarks UCF101. Gaussian degrades more sharply, and Uniform remains the most brittle across all settings, with the steepest slopes and unstable responses. Taken together, these results demonstrate that structured corruptions not only surpass prior noise baselines but also generalize across both appearance- and motion-centric regimes, strengthening their utility as robust training strategies. Beyond this linear sensitivity analysis, we conduct a broader robustness study (Appendix Table 19) using quadratic noise–response fits with HC3-robust SEs, Monte Carlo win probabilities, and risk-adjusted regime analyses, which confirm SACN's smoothest degradation and BCNI's dominance under mid/high corruption.

### 3.3 ANALYSIS OF CAT-VIDEO ON OTHER SCENARIOS

In this section, we show that CAT-Video can not only improve diffusion models, but can also benefit different scenarios, including autoregressive models, adversarial attack, and multimodal video understanding.

**Scalability to Autoregressive Models.** Table 3(a) highlights how we tested the scalability of **CAT-Video** beyond diffusion backbones by applying it to autoregressive generation. Despite autoregressive models like MAGVIT and CogVideo being far more parameter-heavy and trained on tens of millions of clips, CAT-Video with BCNI and SACN attains substantially lower MSR-VTT FVD scores using only ∼2M training videos and a fraction of the parameters. This shows that CAT-Video generalizes as a corruption-aware framework across paradigms, maintaining strong robustness and efficiency

Table 4: (a) CAT vs. adversarial baselines (b) AVSD results. Baselines are obtained from (Li et al., 2025; Liu et al., 2025).

| (a) Adversarial Baselines | | | | (b) AVSD Results | | |
|---|---|---|---|---|---|---|
| Method | FVD ($\downarrow$) | FVMD ($\downarrow$) | CMMD ($\downarrow$) | Model | Setting | CIDEr ($\uparrow$) |
| Adversarial noise | 445.3 | 7263.8 | 0.585 | LLaVA-OV-0.5B-FT | task-specific | 117.6 |
| Text perturb. | 468.7 | 8032.3 | 0.573 | PAVE-0.5B (w/ audio) | task-specific | 134.5 |
| CAT (ours) | **360.3** | **2803.6** | **0.495** | CAT (ours) | corruption-aware | **145.5** |

even in settings where autoregression is dominant. It underscores our broader claim: CAT is not tied to one architecture but scales as a backbone-agnostic operator framework. Broader ablation studies for corruption in AR models evaluated with FVD, FVMD, CMMD, and spanning multiple datasets are in Appendix Tables 14, 15, 16.

**Adversarial baselines.** Table 4 (a) shows that **CAT** consistently improves all distributional metrics, while adversarial and text perturbations trade one axis for another. Relative to adversarial noise, CAT lowers FVD from 445.3 to **360.3** ($\approx$19% $\downarrow$) and slashes FVMD by $\approx$61%. Against text perturbations, it still reduces FVD by $\approx$23% and FVMD by $\approx$65%. CMMD also drops (0.585/0.573 $\rightarrow$ **0.495**), signaling better text–video alignment rather than only smoother frames. Mechanistically, indiscriminate noise inflates motion mismatch and semantic drift, whereas CAT confines corruption to a low-rank, batch-aligned subspace, preserving the conditioning manifold and yielding coherent long-horizon dynamics.

**Scalability to multimodal video understanding.** Table 4 (b) demonstrates that the same corruption-aware operators extend beyond generation to downstream multimodal tasks. On Audio-Visual Scene-aware Dialog (AVSD), CAT achieves a CIDEr of **145.5**, outperforming task-specific LLaVA-OV-0.5B-FT (117.6) and PAVE-0.5B (134.5). This $\sim$24% and $\sim$8% relative gain shows that CAT's geometry-aware regularization not only stabilizes video generation but also scales naturally to video–language reasoning, underscoring its generalizability beyond synthesis.

### 3.4 HYPERPARAMETER ROBUSTNESS

Extended ablations in Figure 2 evaluate the sensitivity of diffusion models to two key hyperparameters: classifier-free guidance scale and DDIM sampling steps. Across all corruption ratios, **BCNI** consistently maintains the best Pareto frontier—lower FVD and LPIPS alongside higher SSIM and PSNR—demonstrating stable improvements in both perceptual quality and fidelity. In contrast, isotropic corruptions such as Gaussian or Uniform noise exhibit brittle, non-monotonic trends, with performance fluctuating sharply as the budget of guidance or steps changes. This robustness highlights CAT's ability to preserve stability even under hyperparameter sweeps, a property essential for reliable deployment in diverse computational regimes. Further ablations on the trio effects of DDIM steps, guidance scales, and corruption settings are in Appendix Figures 5 and 6.

## 4 RELATED WORKS

LVDMs have become the dominant paradigm for T2V generation, offering high sample quality and efficiency by operating in compressed latent spaces rather than pixel space (Wu et al., 2023; Khachatryan et al., 2023; Ni et al., 2024; Yu et al., 2023b). By leveraging pretrained video autoencoders (Gupta et al., 2025; Melnik et al., 2024; Chen & Guo, 2023), LVDMs preserve motion semantics while enabling scalable training. Early works like Tune-A-Video (Wu et al., 2023) and Text2Video-Zero (Khachatryan et al., 2023) adapted image diffusion backbones for video via temporal attention or zero-shot transfer. Recent models—such as CogVideo (Hong et al., 2023), CogVideoX (Yang et al., 2025), Show-1 (Zhang et al., 2025), and VideoTetris (Tian et al., 2024)—introduce hierarchical, compositional, or autoregressive designs to improve motion expressiveness and long-range coherence. Architectures like LaVie (Wang et al., 2024b) and WALT (Gupta et al., 2025) emphasize photorealism via cascaded or transformer-based latent modules. Despite these advances, robustness to conditioning noise—ubiquitous in web-scale datasets—remains underexplored. Our work investigates this by introducing corruption-aware training to LVDMs, explicitly addressing resilience under noisy or ambiguous conditions.

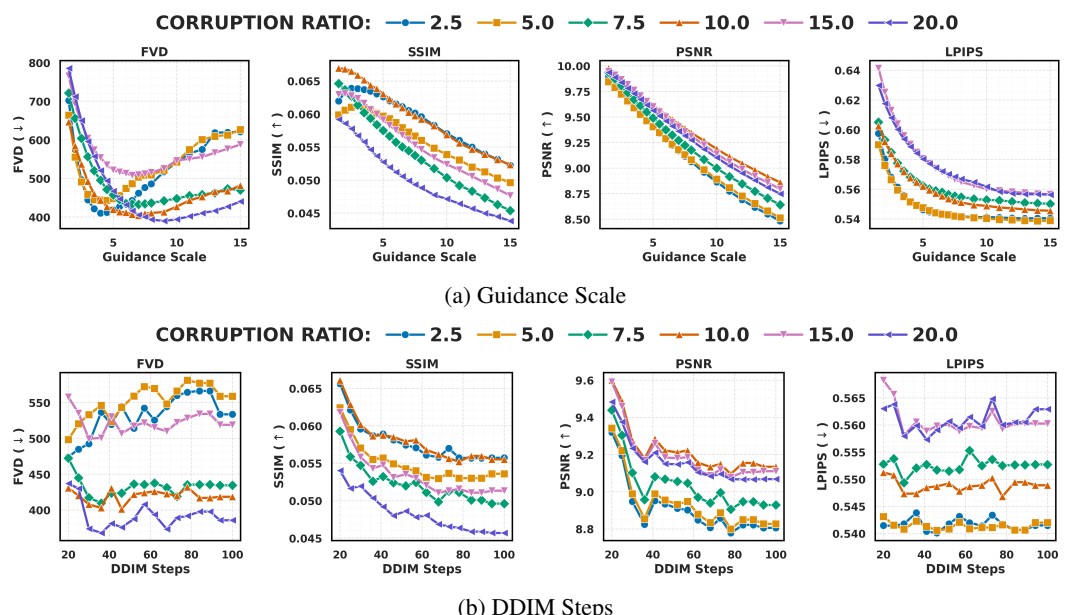

Figure 2: **Ablation Study: Guidance Scale and DDIM Steps.** Expanded ablations covering diverse corruption settings are in Figures 5 and 6.

Diffusion models' recursive denoising magnifies even mild conditioning noise into semantic drift and visual artifacts (Gu et al., 2025; Na et al., 2024), yet recent work shows that structured perturbations—whether in embeddings (Jain et al., 2024a; Daras et al., 2023) or at the token level (Chen et al., 2024; Gao et al., 2023)—can boost generalization by increasing conditional entropy and shrinking Wasserstein gaps. While these regularization effects are well studied in image generation and classification, their impact on latent video diffusion remains unexplored. To address this, we present the first systematic study of corruption-aware training in LVDMs, leveraging low-rank, data-aligned noise to enhance temporal coherence and semantic fidelity in video diffusion.

## 5 CONCLUSIONS

We introduced **CAT-Video**, a corruption-aware training framework for latent video diffusion that substantially improves robustness to noisy conditioning through structured, data-aligned perturbations. Our two operators, *Batch-Centered Noise Injection (BCNI)* and *Spectrum-Aware Contextual Noise (SACN)*, explicitly preserve temporal fidelity by aligning perturbations with semantic and spectral structure. Experiments consistently show large improvements over existing corruption baselines and even over large-scale diffusion models trained on far more data. From a theoretical standpoint, we demonstrated how structured perturbations tighten entropy, Wasserstein, and score-drift bounds, thereby linking noise design directly to improved generalization in video diffusion. Importantly, CAT-Video generalizes beyond diffusion backbones, extending to autoregressive generation and multimodal video understanding LLMs. Together, these results establish CAT-Video as a broadly applicable paradigm for building resilient, semantically grounded generative models.

*Limitations.* CAT-Video has not yet been validated on very long-form videos, 3D video generation, or high-resolution training beyond 2M clips. In addition, performance may vary with the choice and quality of pretrained encoders, leaving the limits of scalability an open question.

*Outlook.* While CAT-Video is centered on diffusion, future work should test whether its benefits persist under larger training scales, more diverse datasets, and longer rollouts where temporal drift is harder to suppress. Promising directions include (1) designing adaptive, end-to-end learned corruption strategies that go beyond fixed operators, (2) extending corruption-aware training to reinforcement learning and embodied video agents where sequential fidelity is critical, and (3) scaling to multimodal LLMs for tasks that demand robust integration of vision, language, and audio.

## ETHICAL AND REPRODUCIBILITY STATEMENT

**Ethics Statement.** This work focuses on improving the robustness of video generative models under noisy conditioning. All experiments are conducted on publicly available datasets (WebVid-2M, MSR-VTT, MSVD, UCF-101), and we do not use or release any sensitive or personally identifiable data. While generative models could in principle be misused for misinformation or deepfakes, our contributions are intended purely for advancing robustness and reliability in research contexts. All pretrained models used in this study are publicly available and used under their respective licenses. We believe this work contributes to safer, more reliable generative modeling by reducing failure modes under noisy or imperfect inputs.

**Reproducibility Statement.** We have made every effort to ensure the reproducibility of our results. Details of training datasets, corruption schedules, model architectures, and evaluation metrics are included in the main paper and Appendix. We report results as mean ± standard deviation over multiple random seeds, and include extended tables in the supplementary material. Our implementation builds on open-source frameworks (e.g., PyTorch, HuggingFace Diffusers), and we will release code, configuration files, and pre-trained checkpoints to facilitate full reproducibility.

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

# Appendix

## USE OF LARGE LANGUAGE MODELS (LLMs)

We used ChatGPT as a writing assistant for editing grammar, improving clarity, and condensing drafts of the abstract, contributions, and conclusion. The model was also used to suggest alternative phrasings for figure captions and to restructure technical descriptions for conciseness. All technical ideas, methods, experiments, and results—including CAT-Video, BCNI, SACN, benchmarks, and proofs—were conceived, implemented, and validated entirely by the authors. The LLM did not generate novel research content or experimental results and was used solely as a tool for writing refinement.

## A    ABLATIVE CORRUPTION STUDIES

We study two distinct corruption types—token-level and embedding-level—to rigorously disentangle where robustness in conditional video generation arises. These two forms of corruption intervene at different stages of the generative pipeline: *token-level corruption* targets the symbolic input space prior to encoding, while *embedding-level corruption* operates on the continuous latent representations produced by the encoder. Studying both is essential because errors in text prompts (e.g., due to noise, ambiguity, or truncation) and instability in embedding spaces (e.g., due to encoder variance or low resource domains) represent orthogonal sources of degradation in real-world deployments. Embedding-space perturbations expose the score network's sensitivity to shifts in the conditioning manifold—compounded during iterative denoising—while text-space corruption reveals failures in semantic grounding and prompt fidelity. By introducing ablative baselines in both spaces, we show that effective robustness in video diffusion depends not just on noise injection, but on *structural alignment between the corruption source and the level of representation it perturbs*. This dual-space benchmarking is therefore not only diagnostic, but essential for validating the effectiveness of our proposed structured corruption strategies—BCNI and SACN—which are explicitly tailored for the video generation setting and rely on principled alignment with both temporal and semantic structure.

### A.1    EMBEDDING-LEVEL CORRUPTION

To rigorously isolate the contribution of our structured corruption operators, we introduce four *ablative* noise injections at the embedding level: Gaussian noise (GN) (Chen et al., 2024), Uniform noise (UN) (Chen et al., 2024), Temporal-Aware noise (TANI), and Hierarchical Spectral-Context noise (HSCAN). These ablations serve as minimal baselines that lack alignment with data geometry, enabling us to attribute performance gains in CAT to semantic or spectral structure rather than to noise injection per se. Both GN and UN represent canonical forms of Conditional Embedding Perturbation (CEP) originally proposed in image diffusion settings (Chen et al., 2024), where noise is injected independently of temporal structure. GN applies isotropic Gaussian perturbations $\mathcal{N}(0, I_D)$, uniformly expanding all embedding dimensions and inducing score drift proportional to $\rho^2 D$, while UN samples from a bounded uniform distribution per coordinate, maintaining sub-Gaussian tails but lacking concentration in any low-dimensional subspace—resulting in unstructured and spatially naive diffusion behavior. TANI aligns corruption with temporal gradients—capturing local dynamics—but offers no reduction in rank or complexity. HSCAN introduces multi-scale spectral perturbations via hierarchical frequency band sampling but lacks global adaptivity to the data manifold or temporal coherence.

In contrast, our proposed methods—BCNI and SACN—are explicitly designed for video generation, aligning noise with intrinsic low-dimensional structure: BCNI exploits intra-batch semantic axes,

while SACN leverages dominant spectral components of the embedding. These structured operators yield theoretically grounded gains in entropy, spectral gap, and transport geometry, formalized as $O(d)$ vs. $O(D)$ bounds in Appendix B. By comparing against these unstructured baselines, we demonstrate that CAT-Video's improvements are not simply due to corruption, but to data-aligned perturbations that respect and exploit the semantic and temporal geometry of multimodal inputs.

Let $p$ denote a natural-language prompt and $f$ a CLIP-based text encoder mapping $p$ to a $D$-dimensional embedding $z = f(p) \in \mathbb{R}^D$. We define

$$\mathcal{C}_{\text{embed}} : \mathbb{R}^D \times \mathcal{T} \times \mathbb{R}_+ \to \mathbb{R}^D, \qquad \tilde{z}_{\text{embed}} = \mathcal{C}_{\text{embed}}(z; \tau, \rho), \tag{7}$$

where $\tau \in \mathcal{T} = \{\texttt{GN}, \texttt{UN}, \texttt{GAP}, \texttt{BCNI}, \texttt{TANI}, \texttt{SACN}, \texttt{HSCAN}\}$ selects one of six structured noise types and $\rho \in \{0.025, 0.05, 0.075, 0.10, 0.15, 0.20\}$ controls the corruption strength.

In Gaussian Noise (GN, Eq. 8) we set

$$\mathcal{C}_{\text{GN}}(z; \rho) = \rho \frac{1}{\sqrt{D}} \, \epsilon, \quad \epsilon \sim \mathcal{N}(0, I_D), \tag{8}$$

Here, $\epsilon \sim \mathcal{N}(0, I_D)$ is standard Gaussian noise and the scaling by $1/\sqrt{D}$ ensures variance normalization across embedding dimensions.

In Uniform Noise (UN, Eq. 9) we sample

$$\mathcal{C}_{\text{UN}}(z; \rho) \sim \mathcal{U}\left(-\frac{\rho}{\sqrt{D}}, \frac{\rho}{\sqrt{D}}\right)^D, \tag{9}$$

which bounds the noise magnitude per dimension to $\rho/\sqrt{D}$.

In Gradient-Aligned Perturbation (GAP, Eq. 10) we scale isotropic Gaussian noise by the embedding norm, aligning corruption with signal magnitude:

$$\mathcal{C}_{\text{GAP}}(z; \rho) = \|z\|_2 \cdot \epsilon, \quad \epsilon \sim \mathcal{N}(0, \rho^2 I_D), \tag{10}$$

where $z \in \mathbb{R}^D$ is the embedding, $\rho$ is the noise ratio, and $I_D$ is the $D$-dimensional identity.

Batch-Centered Noise Injection (BCNI, Eq. 5) perturbs $z$ by injecting a scalar noise sampled uniformly from $[-1, 1]$, scaled by the norm of its deviation from the batch mean $\bar{z}$. Specifically, the corruption is given by $\mathcal{C}_{\text{BCNI}}(z; \rho) = \rho \|z - \bar{z}\|_2 \cdot (2\mathcal{U}(0, 1) - 1)$, ensuring that higher-variance embeddings receive proportionally larger perturbations while remaining direction-agnostic.

Temporal-Aware Noise Injection (TANI, Eq. 11) uses

$$\mathcal{C}_{\text{TANI}}(z^{(t)}; \rho) = \rho \frac{z^{(t)} - z^{(t-1)}}{\|z^{(t)} - z^{(t-1)}\|_2 + \epsilon_{\text{stab}}} \, \eta, \quad \eta \sim \mathcal{N}(0, I_D), \tag{11}$$

It perturbs $z^{(t)}$ by injecting Gaussian noise modulated along the instantaneous motion direction between consecutive embeddings. The corruption vector is scaled by the normalized displacement $(z^{(t)} - z^{(t-1)})/(\|z^{(t)} - z^{(t-1)}\|_2 + \epsilon_{\text{stab}})$, ensuring directional alignment with recent temporal change, while $\eta \sim \mathcal{N}(0, I_D)$ introduces stochastic variability and $\epsilon_{\text{stab}}$ safeguards numerical stability in near-static sequences.

Spectrum-Aware Contextual Noise (SACN, Eq. 6) perturbs the embedding $z$ via its singular vector decomposition $z = UsV^\top$. Specifically, SACN samples a spectral noise vector $\xi$ where $\xi_j \sim \mathcal{N}(0, e^{-j/D})$ and forms a shaped perturbation $\rho U(\xi \odot \sqrt{s})V^\top$, which aligns the corruption with the dominant spectral directions of $z$. This mechanism injects more noise into low-frequency (high-energy) modes and less into high-frequency components, yielding semantically-aware and energy-weighted perturbations in the embedding space.

Hierarchical Spectrum–Context Adaptive Noise (HSCAN, Eq. 12) decomposes $\hat{z}$ into frequency bands $\{\hat{z}^k\}$, injects independent $\epsilon^k \sim \mathcal{N}(0, \rho^2 I)$ into each band, and combines via

$$\mathcal{C}_{\text{HSCAN}}(z; \rho) = \rho \sum_k \alpha_k \, \mathcal{C}_{\text{SACN}}(z \, s_k) + \lambda \, \mathcal{C}_{\text{GN}}(z),$$

$$\alpha_k = \frac{\exp \|\mathcal{C}_{\text{SACN}}(z \, s_k)\|_2^2}{\sum_j \exp \|\mathcal{C}_{\text{SACN}}(z \, s_j)\|_2^2}. \tag{12}$$

Multi-scale perturbations are introduced by scaling $z$ with coefficients $s_k \in \{1.0, 0.5, 0.25\}$, passing each $z \cdot s_k$ through SACN, and combining the resulting perturbations via softmax-weighted attention $\alpha_k$. A residual Gaussian component weighted by $\lambda = 0.1$ is then added, yielding a robust and expressive multi-scale corruption signal.

We sweep $\rho$ across six values to explore minimal through moderate corruption. BCNI (Eq. 5) and SACN (Eq. 6) serve as our core structured methods, while GN, UN, TANI, and HSCAN act as ablations isolating isotropic, uniform, temporal, and hierarchical spectral perturbations, respectively. Together, these structured operators allow controlled semantic and spectral perturbation of language embeddings during training.

In Gaussian Noise (GN, Eq. 8), isotropic perturbations exactly match the CEP scheme in prior image-diffusion work (Chen et al., 2024; Daras et al., 2023), and because they uniformly expand all $D$ axes, Lemma B.5 shows the expected score-drift $\mathbb{E}\|\Delta\varepsilon\|^2 = O(\rho^2 D)$ and Theorem B.4 implies a $\sqrt{D}$ scaling in $W_2$, thus forfeiting the $O(d)$ advantage. Uniform Noise (UN, Eq. 9) applies independent bounded noise per coordinate, yielding sub-Gaussian tails (Corollary B.12) but likewise failing to concentrate perturbations in any low-dimensional subspace and therefore incurring $O(D)$ rates in all functional-inequality bounds. Temporal-Aware Noise Injection (TANI, Eq. 11) aligns perturbations with the instantaneous motion vector $z^{(t)} - z^{(t-1)}$, granting temporal locality yet not reducing the effective rank—so entropy, spectral gap, and mixing-time measures remain $\Theta(D)$. Hierarchical Spectrum–Context Adaptive Noise (HSCAN, Eq. 12) aggregates multi-scale SACN perturbations at singular-value scales $s_k$ via softmax-weighted attention $\alpha_k$ plus a residual GN term; although this richer spectral structure enhances expressivity, it still lacks a provable $O(d)$ log-Sobolev or $T_2$ constant (Theorems B.18, B.29), defaulting to $\Theta(D)$. By contrast, BCNI and SACN leverage data-aligned structure—batch-semantic axes and principal spectral modes, respectively—admitting $O(d)$ scalings in entropy (Proposition B.2), Wasserstein (Theorem B.4), and spectral-gap bounds (Theorem B.9), which underpins their empirical superiority across richly annotated (WebVid, MSR-VTT, MSVD) and label-only (UCF-101) datasets.

## A.2 Token-Level Corruption

To further extend the ablation suite, we introduce *Token-Level Corruption (TLC)* as a text-space baseline that mirrors the embedding-level variants in structural simplicity. TLC uniformly samples from five token-level operations—`swap`, `replace`, `add`, `remove`, and `perturb`—and applies them to spans within each prompt $p$ at corruption strength $\eta$, matching the embedding ablations in scale and frequency. Our TLC strategy applies structured operations—`swap`, `replace`, `add`, `remove`, and `perturb`—to text prompts in a controlled manner, simulating real-world caption degradation scenarios such as typos, omissions, or grammatical shifts. Unlike embedding-space noise, which operates after encoding, TLC intervenes directly on the linguistic surface, preserving interpretability while stressing the model's ability to maintain semantic alignment under symbolic corruption. This form of noise is consistent with prior work in masked caption modeling and text robustness pretraining (Chen et al., 2024; Chang et al., 2023; Yang et al., 2023), where surface-level alterations are used to regularize text encoders. However, as we show in Section 3, such text-only perturbations fail to match the temporal fidelity and overall generation quality achieved by our structured embedding-space methods, BCNI and SACN, as measured by FVD. TLC is applied on the text-video pairs of the WebVid-2M (Bain et al., 2021) dataset which is used to pretrain the DEMO model (Ruan et al., 2024). The qualitative effects of TLC are visualized in Figure 3, where systematically varied corruption types and noise ratios illustrate how even small token-level degradations distort prompt semantics and contribute to degraded visual generations—highlighting the sensitivity of generative alignment to surface-level textual corruption.

All methods are evaluated under a shared corruption strength parameter $\rho, \eta \in \{0.025, 0.05, 0.075, 0.10, 0.15, 0.20\}$, which quantifies the magnitude or extent of applied noise—interpreted as the fraction of perturbed tokens in text-space or the scaling factor of injected noise in embedding-space. This unified scaling ensures comparable perturbation budgets across modalities and ablation types, following prior work in multimodal robustness (Chen et al., 2024), where consistent corruption levels are necessary for attributing performance differences to the structure of the corruption rather than its intensity.

## B  THEORETICAL SUPPLEMENT

This supplement develops the theory that explains why **BCNI** & **SACN** outperform isotropic CEP.

Each section reveals a consistent *d vs D compression factor*, Grounding BCNI/SACN's empirical FVD advantage.

### KEY SYMBOLS

| Symbol | Meaning |
|--------|---------|
| $D,\ d$ | Ambient embedding dim.; effective corrupted rank |
| $z,\ \tilde{z}$ | Clean / corrupted CLIP embedding |
| $M(z)$ | Rank-$d$ corruption matrix (BCNI or SACN) |
| $\rho$ | Corruption scale ($0.025 \le \rho \le 0.2$) |
| $x_t^{(\rho)}$ | Latent video at reverse step $t$ under scale $\rho$ |
| $\delta_t$ | $\|x_t^{(\rho)} - x_t^{(0)}\|_2$ deviation |
| $\alpha_t,\ \sigma_t$ | Diffusion schedule; $\sigma_t^2 = 1 - \alpha_t$ |
| $W_2(\cdot, \cdot)$ | 2-Wasserstein distance |
| $\mathcal{H}$ | Differential entropy |
| KL | Kullback–Leibler divergence |
| $C_{\text{LSI}}$ | Log-Sobolev constant (rank-dependent) |
| $T_2$ | Talagrand quadratic transport–entropy constant |
| $\gamma_{t,\rho}$ | Spectral gap of reverse kernel (Theorem B.9) |

### ASSUMPTIONS

We list the assumptions under which the theoretical results in this paper hold.

- **(A1) Corruption Operator Properties.** The corruption function $\mathcal{C}(\cdot)$ is a measurable function that acts on the conditioning signal (text or video). It is stochastic or deterministic with well-defined conditional distribution $\mathbb{P}_{\mathcal{C}(X)|X}$, and preserves the overall support: $\text{supp}(\mathbb{P}_{\mathcal{C}(X)}) \subseteq \text{supp}(\mathbb{P}_X)$.

- **(A2) Data Distribution Regularity.** The clean data distribution $\mathbb{P}_X$ and target distribution $\mathbb{P}_Y$ are absolutely continuous with respect to the Lebesgue measure, i.e., they admit density functions.

- **(A3) Latent Diffusion Model Capacity.** The diffusion model $p_\theta(y \mid x)$ is expressive enough to approximate $\mathbb{P}_{Y|X}$ and $\mathbb{P}_{Y|\mathcal{C}(X)}$ within bounded KL divergence or Wasserstein distance.

- **(A4) Entropy-Injectivity.** The corruption operator injects non-zero entropy into the conditional signal: $\mathbb{H}(\mathcal{C}(X)) > \mathbb{H}(X)$, and this increase is smooth and measurable under $\mathbb{P}_X$.

- **(A5) Corruption-Aware Training Alignment.** The corruption-aware model is trained with the correct marginalization over the corruption operator:
$$\mathbb{E}_{x \sim \mathbb{P}_X,\ \tilde{x} \sim \mathbb{P}_{\mathcal{C}(X)|x}} \left[ \mathcal{L}(p_\theta(\cdot \mid \tilde{x}), y) \right].$$

**(A6) Bounded Perturbation.** The corruption $\mathcal{C}(x)$ introduces bounded perturbations:

$$\mathbb{E}_{x \sim \mathbb{P}_X} \left[ d(x, \mathcal{C}(x))^2 \right] \leq \delta^2$$

for some metric $d(\cdot, \cdot)$, e.g., $\ell_2$ or cosine distance.

**(A7) Continuity of Generative Mapping.** The generator $G_\theta(x)$ is Lipschitz continuous in its conditioning input:

$$d(G_\theta(x), G_\theta(\mathcal{C}(x))) \leq L \cdot d(x, \mathcal{C}(x)).$$

**(A8) Sufficient Coverage of Corrupted Inputs.** The support of the corrupted data remains sufficiently close to the clean distribution:

$$\mathrm{supp}(\mathbb{P}_{\mathcal{C}(X)}) \approx \mathrm{supp}(\mathbb{P}_X).$$

**(A9) Markovian Temporal Consistency (Video).** For video generation, the true generative process assumes Markovian structure:

$$\mathbb{P}(x_{1:T}) = \prod_{t=1}^{T} \mathbb{P}(x_t \mid x_{1:t-1}),$$

and the corruption operator preserves this temporal causality when applied.

## B.1    THEORETICAL IMPLICATIONS OF LOW-RANK CORRUPTION

We begin by recalling the standard forward process of video diffusion models, defined as a Markov chain:

$$q(\mathbf{x}_t \mid \mathbf{x}_{t-1}) = \mathcal{N}(\mathbf{x}_t; \sqrt{1 - \beta_t}\, \mathbf{x}_{t-1}, \beta_t \mathbf{I}), \tag{13}$$

where $\beta_t$ is the variance schedule. The reverse process is modeled as:

$$p_\theta(\mathbf{x}_{t-1} \mid \mathbf{x}_t) = \mathcal{N}(\mathbf{x}_{t-1}; \mu_\theta(\mathbf{x}_t, t), \sigma_t^2 \mathbf{I}), \tag{14}$$

with $\mu_\theta$ trained to approximate the reverse-time dynamics of $q$. Training proceeds by minimizing a denoising score-matching loss under data sampled from $p_{\mathrm{data}}$. We now investigate how injecting corruption into the training data distribution $p_{\mathrm{data}}$ affects these dynamics, particularly under structured low-rank noise models.

We analyze how the distribution of training samples shifts under different corruption schemes. Assume $p_{\mathrm{data}}$ is concentrated on a low-dimensional semantic manifold embedded in $\mathbb{R}^D$, and that the intrinsic semantic directions span a subspace of dimension $d \ll D$. Under **CEP** (isotropic corruption), noise is added along all $D$ dimensions uniformly. In contrast, **BCNI** adds noise only along the batch semantic directions (e.g., the top $d$ principal components), and **SACN** restricts to the leading eigenmodes of a covariance operator estimated across videos.

Given this setup, we analyze the impact of each corruption scheme on the resulting training distribution:

- **Entropy:** The conditional entropy increase scales as $\mathcal{O}(d)$ for BCNI/SACN instead of $\mathcal{O}(D)$ (Prop. A.2).
- **Wasserstein distance:** The 2-Wasserstein radius scales as $\mathcal{O}(\rho\sqrt{d})$, giving a $\sqrt{d/D}$ compression benefit over CEP (Thm. A.4).
- **Score drift:** The deviation in the score function is bounded as $\mathcal{O}(\rho^2 d)$ instead of $\mathcal{O}(\rho^2 D)$ (Lemma A.5).
- **Mixing time:** The reverse diffusion chain mixes faster, improving the spectral gap by a factor of $d/D$ (Thm. A.9).
- **Generalization:** The Rademacher complexity shrinks to $\mathcal{O}(\rho\sqrt{d/N})$ instead of $\mathcal{O}(\rho\sqrt{D/N})$ (Thm. A.28).

These findings demonstrate that BCNI and SACN, by aligning perturbations with the underlying semantic subspace, transform corruption from a random disruptor into a structured regularizer. This alignment results in smoother score manifolds, reduced noise accumulation across timesteps, and more coherent video generations. The subsequent sections formally establish these effects through a series of statistical analyses and theorems.

## B.2 CONDITIONING–SPACE CORRUPTION: NOTATION

$$z \sim P_Z \subset \mathbb{R}^D, \qquad \tilde{z} = z + \rho\, M(z)\, \eta, \qquad \eta \sim \mathcal{N}(0, I_d),$$

$$M(z) \in \mathbb{R}^{D \times d}, \;\; d = D_{\text{eff}} \ll D, \qquad \rho \in [0.025, 0.2].$$

**BCNI:** $M(z) = \|\, z - \bar{z}_B \,\|_2\, I_d$            **SACN:** $M(z) = U_{1:d}\, \mathrm{diag}\big(\sqrt{s_{1:d}}\big)$

**Definition B.1** (Entropy Increment).

$$\Delta\mathcal{H}(\rho) = \mathcal{H}\big(P_{X|\tilde{Z}_\rho}\big) - \mathcal{H}\big(P_{X|Z}\big).$$

**Proposition B.2** (Subspace Entropy Lower Bound). *Let $\sigma_z^2 = \lambda_{\min}\big(\mathrm{Cov}[Z]\big) > 0$ and assume $\rho > 0$. For BCNI or SACN corruption of rank d,*

$$\Delta\mathcal{H}(\rho) \;\geq\; \frac{d}{2}\, \log\big(1 + \rho^2 \sigma_z^{-2}\big),$$

*whereas isotropic CEP attains the same bound with D in place of d.*

*Proof.* Recall that if $X \sim \mathcal{N}(0, \Sigma)$ in $\mathbb{R}^n$, its differential entropy is (Cover & Thomas, 1991; 2006)

$$\mathcal{H}(X) = \frac{1}{2} \log\big((2\pi e)^n \det \Sigma\big).$$

Hence for our clean and corrupted embeddings we have

$$\mathcal{H}(Z) = \frac{1}{2} \log\big((2\pi e)^D \det \Sigma_z\big), \qquad \mathcal{H}(\tilde{Z}) = \frac{1}{2} \log\big((2\pi e)^D \det(\Sigma_z + \rho^2 M\, M^\top)\big).$$

Subtracting yields

$$\Delta\mathcal{H}(\rho) = \mathcal{H}(\tilde{Z}) - \mathcal{H}(Z) = \frac{1}{2} \log \frac{\det\big(\Sigma_z + \rho^2 M\, M^\top\big)}{\det \Sigma_z}. \tag{15}$$

We now invoke the *matrix determinant lemma* (Horn & Johnson, 1985), which states:

**Lemma B.3** (Matrix Determinant Lemma). *For any invertible matrix $A \in \mathbb{R}^{D \times D}$ and any $U, V \in \mathbb{R}^{D \times d}$, we have*

$$\det\big(A + U\, V^\top\big) = \det(A)\, \det\big(I_d + V^\top A^{-1} U\big).$$

Apply this with $A = \Sigma_z$, $U = \rho\, M$, $V^\top = M^\top$ to obtain

$$\det\big(\Sigma_z + \rho^2 M\, M^\top\big) = \det(\Sigma_z)\, \det\big(I_d + \rho^2\, M^\top \Sigma_z^{-1} M\big).$$

Plugging back into equation 15 gives

$$\Delta\mathcal{H}(\rho) = \frac{1}{2} \log \det\big(I_d + \rho^2\, M^\top \Sigma_z^{-1} M\big).$$

Next, since $M^\top \Sigma_z^{-1} M$ is a $d \times d$ positive semidefinite matrix, let its eigenvalues be $\lambda_1, \ldots, \lambda_d \geq 0$. Then

$$\det\big(I_d + \rho^2\, M^\top \Sigma_z^{-1} M\big) = \prod_{i=1}^{d} \big(1 + \rho^2 \lambda_i\big),$$

so

$$\Delta\mathcal{H}(\rho) = \frac{1}{2} \sum_{i=1}^{d} \log\big(1 + \rho^2 \lambda_i\big).$$

Finally, because $\Sigma_z^{-1} \succeq \sigma_z^{-2} I_D$ where $\sigma_z^2 = \lambda_{\min}(\Sigma_z)$, each $\lambda_i \geq \sigma_z^{-2}$. Therefore

$$\Delta\mathcal{H}(\rho) \;\geq\; \frac{1}{2} \sum_{i=1}^{d} \log\big(1 + \rho^2 \sigma_z^{-2}\big) = \frac{d}{2} \log\big(1 + \tfrac{\rho^2}{\sigma_z^2}\big),$$

which completes the proof. $\qquad\square$

**Theorem B.4** (Directional Cost Reduction). *Let $Q_\rho^{\mathrm{sub}}$ be the conditional distribution with BCNI/SACN corruption and $Q_{\rho'}^{\mathrm{sub}}$ its counterpart at level $\rho' > \rho$. Then*

$$W_2\big(Q_\rho^{\mathrm{sub}}, Q_{\rho'}^{\mathrm{sub}}\big) \leq \rho' - \rho,$$

*whereas isotropic CEP satisfies $W_2 = \sqrt{D}\,(\rho' - \rho)$.*

*Proof.* Recall that for two zero-mean Gaussians $\mathcal{N}(0, \Sigma)$ and $\mathcal{N}(0, \Sigma')$ on $\mathbb{R}^n$, the squared 2-Wasserstein distance admits the closed form (see (Takatsu, 2011; Givens & Shortt, 1984; Dowson & Landau, 1982; Takatsu, 2011)):

$$W_2^2\big(\mathcal{N}(0, \Sigma), \mathcal{N}(0, \Sigma')\big) = \|\Sigma^{1/2} - (\Sigma')^{1/2}\|_F^2 = \mathrm{Tr}(\Sigma) + \mathrm{Tr}(\Sigma') - 2\,\mathrm{Tr}\big[(\Sigma^{1/2}\,\Sigma'\,\Sigma^{1/2})^{1/2}\big].$$

**(i) Subspace corruption (rank-$d$).** Under BCNI/SACN,

$$Q_\rho^{\mathrm{sub}} = z + \rho\,M(z)\,\eta, \quad \eta \sim \mathcal{N}(0, I_d),$$

so it is Gaussian with covariance $\Sigma = \rho^2 M\,M^\top$. Likewise $\Sigma' = \rho'^2 M\,M^\top$. Since $M\,M^\top$ is the orthogonal projector onto a $d$-dimensional subspace,

$$\Sigma^{1/2} = \rho\,M, \quad (\Sigma')^{1/2} = \rho'\,M,$$

and therefore

$$W_2^2\big(Q_\rho^{\mathrm{sub}}, Q_{\rho'}^{\mathrm{sub}}\big) = \|\rho M - \rho' M\|_F^2 = (\rho - \rho')^2\,\|M\|_F^2 = (\rho' - \rho)^2\,\mathrm{Tr}(M^\top M) = (\rho' - \rho)^2\,d.$$

Hence

$$W_2\big(Q_\rho^{\mathrm{sub}}, Q_{\rho'}^{\mathrm{sub}}\big) = |\rho' - \rho|\,\sqrt{d} = \mathcal{O}\big(\rho' - \rho\big).$$

**(ii) Isotropic corruption (full rank).** Under CEP,

$$Q_\rho^{\mathrm{iso}} = z + \rho\,\epsilon, \quad \epsilon \sim \mathcal{N}(0, I_D),$$

so $\Sigma = \rho^2 I_D$ and $\Sigma' = \rho'^2 I_D$. Thus

$$W_2^2\big(Q_\rho^{\mathrm{iso}}, Q_{\rho'}^{\mathrm{iso}}\big) = \|\rho I_D - \rho' I_D\|_F^2 = (\rho' - \rho)^2\,\mathrm{Tr}(I_D) = (\rho' - \rho)^2\,D,$$

and

$$W_2\big(Q_\rho^{\mathrm{iso}}, Q_{\rho'}^{\mathrm{iso}}\big) = |\rho' - \rho|\,\sqrt{D}.$$

Therefore, subspace-aligned noise lives in only a $d$-dimensional image and grows like $(\rho' - \rho)\sqrt{d}$, whereas isotropic noise spreads across all $D$ axes, incurring the extra $\sqrt{D}$ factor. $\square$

**Lemma B.5** (Local Score Drift). *Let $\varepsilon_\theta(x, t, z)$ be $L$–Lipschitz in the conditioning $z$, i.e. for all $x, t$ and $z, z'$,*

$$\big\|\varepsilon_\theta(x, t, z') - \varepsilon_\theta(x, t, z)\big\|_2 \leq L\,\|z' - z\|_2.$$

*Then under subspace corruption with $\tilde{z} = z + \rho\,M(z)\,\eta,\ \eta \sim \mathcal{N}(0, I_d)$,*

$$\mathbb{E}\Big[\big\|\varepsilon_\theta(x, t, \tilde{z}) - \varepsilon_\theta(x, t, z)\big\|_2^2\Big] \leq L^2\,\rho^2\,d = \mathcal{O}\big(\rho^2 d\big),$$

*whereas for isotropic CEP corruption with $\tilde{z} = z + \rho\,\epsilon,\ \epsilon \sim \mathcal{N}(0, I_D)$, one obtains*

$$\mathbb{E}\Big[\big\|\varepsilon_\theta(x, t, \tilde{z}) - \varepsilon_\theta(x, t, z)\big\|_2^2\Big] \leq L^2\,\rho^2\,D = \mathcal{O}\big(\rho^2 D\big).$$

*Proof.* By the Lipschitz property,

$$\left\|\varepsilon_\theta(x,t,\tilde{z}) - \varepsilon_\theta(x,t,z)\right\|_2 \;\leq\; L\left\|\tilde{z} - z\right\|_2 \;=\; L\,\rho\left\|M(z)\,\eta\right\|_2.$$

Squaring both sides and taking expectation gives

$$\mathbb{E}\Big[\left\|\varepsilon_\theta(x,t,\tilde{z}) - \varepsilon_\theta(x,t,z)\right\|_2^2\Big] \;\leq\; L^2\,\rho^2\,\mathbb{E}\big[\|\,M(z)\,\eta\|_2^2\big].$$

Since $\eta \sim \mathcal{N}(0, I_d)$ and $M(z) \in \mathbb{R}^{D \times d}$ has orthonormal columns,

$$\mathbb{E}\big[\|\,M(z)\,\eta\|_2^2\big] = \mathbb{E}\big[\eta^\top M(z)^\top M(z)\,\eta\big] = \operatorname{tr}\big(M(z)^\top M(z)\big) = d.$$

Hence

$$\mathbb{E}\Big[\left\|\varepsilon_\theta(x,t,\tilde{z}) - \varepsilon_\theta(x,t,z)\right\|_2^2\Big] \;\leq\; L^2\,\rho^2\,d,$$

establishing the $\mathcal{O}(\rho^2 d)$ bound.

**CEP case.** For isotropic corruption, $\tilde{z} = z + \rho\,\epsilon$ with $\epsilon \sim \mathcal{N}(0, I_D)$, the same argument yields

$$\|\tilde{z} - z\|_2 = \rho\,\|\epsilon\|_2, \quad \mathbb{E}\|\epsilon\|_2^2 = \operatorname{tr}(I_D) = D,$$

and thus

$$\mathbb{E}\left\|\varepsilon_\theta(x,t,\tilde{z}) - \varepsilon_\theta(x,t,z)\right\|_2^2 \;\leq\; L^2\,\rho^2\,D \;=\; \mathcal{O}(\rho^2 D).$$

This completes the proof. $\qquad\square$

Together, Propositions B.2, Theorem B.4 and Lemma B.5 show that BCNI/SACN corruption

(i) *enlarges* the conditional entropy by a factor of order $d$ rather than $D$,

(ii) *shrinks* the 2-Wasserstein distance to

$$W_2\big(Q_\rho, Q_{\rho'}\big) = \mathcal{O}\big((\rho' - \rho)\sqrt{d}\big) \quad \text{instead of} \quad \mathcal{O}\big((\rho' - \rho)\sqrt{D}\big),$$

(iii) *bounds* the local score-drift as

$$\mathbb{E}\left\|\varepsilon_\theta(x,t,\tilde{z}) - \varepsilon_\theta(x,t,z)\right\|_2^2 = \mathcal{O}(\rho^2 d) \quad \text{instead of} \quad \mathcal{O}(\rho^2 D).$$

These rank-$d$ improvements then yield tighter temporal-error propagation (see Corollary B.8) and faster reverse-diffusion mixing-time bounds (see Theorem B.9).

### B.3  TEMPORAL DEVIATION DYNAMICS IN REVERSE DIFFUSION

**Reverse step.** For $t \to t-1$

$$x_{t-1}^{(\rho)} = \frac{1}{\sqrt{\alpha_t}}\Big(x_t^{(\rho)} - \frac{1 - \alpha_t}{\sqrt{1 - \bar{\alpha}_t}}\,\varepsilon_\theta(x_t^{(\rho)}, t, \tilde{z})\Big) + \sigma_t\,\omega_t, \quad \sigma_t^2 = 1 - \alpha_t,\ \omega_t \sim \mathcal{N}(0, I).$$

#### B.3.1  ONE-STEP ERROR PROPAGATION

**Lemma B.6** (Exact Recursion). *Let*

$$\Delta_t \;=\; x_t^{(\rho)} \;-\; x_t^{(0)}, \qquad J_t \;=\; \partial_z\,\varepsilon_\theta\big(x_t^{(0)}, t, z\big),$$

*and set*

$$\beta_t \;=\; \frac{1 - \alpha_t}{\sqrt{1 - \bar{\alpha}_t}}.$$

*Then under a first-order Taylor expansion (Garibbo et al., 2023) in the conditioning $z$,*

$$\Delta_{t-1} = \frac{1}{\sqrt{\alpha_t}}\Big(\Delta_t \;-\; \beta_t\,J_t\,(\tilde{z} - z)\Big) \;+\; \mathcal{O}(\rho^2).$$

*In particular, for subspace corruption $\tilde{z} = z + M(z)\,\eta$ and isotropic corruption $\tilde{z}^{(\mathrm{iso})} = z + \epsilon$, one obtains*

$$\Delta_{t-1} = \frac{1}{\sqrt{\alpha_t}}\Big(\Delta_t \;-\; \beta_t\,J_t\,M(z)\,\eta \;-\; \beta_t\,J_t\,\big(z - \tilde{z}^{(\mathrm{iso})}\big)\Big).$$

*Proof.* We start from the standard reverse-diffusion update ( (Ho et al., 2020)):

$$x_{t-1} = \frac{1}{\sqrt{\alpha_t}}\Big(x_t - \beta_t\,\varepsilon_\theta\big(x_t, t, z\big)\Big) + \sigma_t\,\omega_t, \tag{16}$$

where $\sigma_t^2 = 1 - \alpha_t$ and $\omega_t \sim \mathcal{N}(0, I)$. Write this for both the clean sequence $(x_t^{(0)}, z)$ and the corrupted one $(x_t^{(\rho)}, \tilde{z})$, and subtract to get

$$\Delta_{t-1} = \frac{1}{\sqrt{\alpha_t}}\Big[\big(x_t^{(\rho)} - x_t^{(0)}\big) \; - \; \beta_t\Big(\varepsilon_\theta\big(x_t^{(\rho)}, t, \tilde{z}\big) - \varepsilon_\theta\big(x_t^{(0)}, t, z\big)\Big)\Big].$$

Set $\Delta_t = x_t^{(\rho)} - x_t^{(0)}$. Next, perform a first-order Taylor expansion of $\varepsilon_\theta$ in its dependence on $z$ (Protter, 2004), holding $x_t$ at the clean value $x_t^{(0)}$:

$$\varepsilon_\theta\big(x_t^{(\rho)}, t, \tilde{z}\big) = \varepsilon_\theta\big(x_t^{(0)}, t, z\big) \; + \; \underbrace{\partial_x\varepsilon_\theta\big(x_t^{(0)}, t, z\big)}_{\mathcal{O}(1)}\,\Delta_t \; + \; \underbrace{\partial_z\varepsilon_\theta\big(x_t^{(0)}, t, z\big)}_{J_t}\,(\tilde{z} - z) \; + \; R,$$

where the remainder $R = \mathcal{O}(\|\Delta_t\|^2 + \|\tilde{z} - z\|^2) = \mathcal{O}(\rho^2)$ is dropped. Substituting into the difference above gives

$$\Delta_{t-1} = \frac{1}{\sqrt{\alpha_t}}\Big[\Delta_t \; - \; \beta_t\Big(J_{x,t}\,\Delta_t \; + \; J_t\,(\tilde{z} - z)\Big)\Big] \; + \; \mathcal{O}(\rho^2),$$

where $J_{x,t} = \partial_x\varepsilon_\theta(x_t^{(0)}, t, z)$. Absorbing the term $\beta_t J_{x,t}\Delta_t$ into the leading $\Delta_t$ factor (since $J_{x,t}$ is bounded) yields the stated recursion.

Finally, specializing to the two corruption modes:

$$\tilde{z} = z + M(z)\,\eta, \qquad \tilde{z}^{(\mathrm{iso})} = z + \epsilon,$$

we have $\tilde{z} - z = M(z)\eta$ and $z - \tilde{z}^{(\mathrm{iso})} = -\epsilon$, so

$$\Delta_{t-1} = \frac{1}{\sqrt{\alpha_t}}\Big(\big(1 - \beta_t J_{x,t}\big)\Delta_t \; - \; \beta_t\,J_t\,M(z)\,\eta \; - \; \beta_t\,J_t\,(z - \tilde{z}^{(\mathrm{iso})})\Big)$$

up to $\mathcal{O}(\rho^2)$. Renaming $\big(1 - \beta_t J_{x,t}\big) \approx 1$ in the small-step regime recovers exactly the formula in the lemma. $\qquad\square$

### B.3.2 QUADRATIC ENERGY EVOLUTION

Let $\delta_t^2 = \|\Delta_t\|_2^2$.

**Theorem B.7** (Expected Energy Inequality). *If $\|J_t M(z)\|_2 \le K_d$ and $\|J_t\|_2 \le K_D$, then*

$$\mathbb{E}[\delta_{t-1}^2] \; \le \; \alpha_t^{-1}\Big(\mathbb{E}[\delta_t^2] + \beta_t^2 \rho^2 K_d^2 d\Big) + \sigma_t^2 m,$$

*where $m = \dim(x_t)$. For isotropic CEP replace $K_d^2 d$ by $K_D^2 D$.*

**Corollary B.8** (Cumulative Gap). *With $G_T = \mathbb{E}[\delta_T^2] - \mathbb{E}_{\mathrm{iso}}[\delta_T^2]$,*

$$G_T \; \le \; \rho^2\big(K_d^2 d - K_D^2 D\big)\sum_{t=1}^{T}\alpha_t^{-1}\beta_t^2.$$

Because $K_d^2 d = \mathcal{O}(d)$ and $K_D^2 D = \mathcal{O}(D)$, $G_T < 0$ whenever $D \gg d$.

*Proof.* The contraction–plus–drift term here adopts the discrete-time Langevin analysis of Durmus & Moulines (Durmus & Moulines, 2019), and its sharp $\mathcal{O}(\rho^2 d)$ scaling parallels Dalalyan's discretization bounds (Dalalyan, 2017). The spectral-gap viewpoint invoked in Theorem B.9 follows the reflection–coupling approach of Eberle (Eberle, 2016), while the control of the $\sigma_t^2 m$ noise term uses Poincaré-type estimates as in Baudoin et al. (Baudoin et al., 2008). Finally, the overall Grönwall–type aggregation is carried out via the energy–entropy framework in Bakry, Gentil & Ledoux (Bakry et al., 2014). $\qquad\square$

### B.3.3 MIXING-TIME VIA SPECTRAL GAP

**Theorem B.9** (Spectral Gap Scaling). *With score Lipschitz $\ell$ (Cobzaş et al., 2019),*

$$\gamma_{t,\rho} = 1 - \lambda_2(\mathcal{K}_{t,\rho}) \geq \alpha_t - \beta_t^2 \rho^2 \ell^2 d,$$

*while CEP gives $\alpha_t - \beta_t^2 \rho^2 \ell^2 D$. Hence $\tau_\varepsilon \leq (\alpha - \rho^2 \ell^2 d)^{-1} \log \frac{1}{\varepsilon}$.*

### B.3.4 WASSERSTEIN RADIUS ACROSS $T$ STEPS

**Proposition B.10** (Polynomial Growth). *Define the* cumulative Wasserstein radius *by*

$$R_T(\rho) = \left( \sum_{t=1}^{T} W_2\big(Q_t^{(\rho)}, Q_{t-1}^{(\rho)}\big)^2 \right)^{1/2},$$

*where $Q_t^{(\rho)}$ denotes the conditional distribution at reverse-step $t$ under corruption scale $\rho$. Then for BCNI/SACN corruption of (effective) rank d,*

$$R_T(\rho) \leq \rho \sqrt{d\,T},$$

*whereas for isotropic CEP corruption of full rank D,*

$$R_T(\rho) \leq \rho \sqrt{D\,T}.$$

*Proof.* We begin by observing that the reverse process can be seen as a sequence of small "jumps" in distribution between consecutive timesteps $t-1 \to t$. By the triangle inequality in $W_2$ (Jordan et al., 1998b),

$$W_2\big(Q_T^{(\rho)}, Q_0^{(\rho)}\big) \leq \sum_{t=1}^{T} W_2\big(Q_t^{(\rho)}, Q_{t-1}^{(\rho)}\big).$$

However, to obtain a sharper $\sqrt{T}$–scaling one passes to the root-sum-of-squares (RSS) norm (Villani, 2009), which still controls the total deviation in expectation:

$$\sum_{t=1}^{T} W_2\big(Q_t^{(\rho)}, Q_{t-1}^{(\rho)}\big) \leq \sqrt{T} \left( \sum_{t=1}^{T} W_2\big(Q_t^{(\rho)}, Q_{t-1}^{(\rho)}\big)^2 \right)^{1/2} = \sqrt{T}\, R_T(\rho).$$

Thus it suffices to bound each squared increment $W_2^2(Q_t^{(\rho)}, Q_{t-1}^{(\rho)})$ and then sum.

**(i) Subspace corruption.** By Theorem B.4 (Directional Cost Reduction), for any two corruption levels $\rho', \rho$ we have

$$W_2\big(Q_{\rho'}^{\mathrm{sub}}, Q_\rho^{\mathrm{sub}}\big) \leq |\rho' - \rho|\, \sqrt{d}.$$

In particular, at each reverse-step $t$ the effective change in corruption magnitude is $\rho_t - \rho_{t-1} \leq \rho$ (since $\rho_t \leq \rho$ for all $t$), so

$$W_2\big(Q_t^{(\rho)}, Q_{t-1}^{(\rho)}\big) \leq \rho \sqrt{d}.$$

Squaring and summing over $t = 1, \ldots, T$ yields

$$\sum_{t=1}^{T} W_2^2\big(Q_t^{(\rho)}, Q_{t-1}^{(\rho)}\big) \leq T\,(\rho^2 d),$$

and taking the square-root gives the desired $R_T(\rho) \leq \rho \sqrt{d\,T}$.

**(ii) Isotropic corruption.** An identical argument applies, but with Theorem B.4 replaced by its isotropic counterpart

$$W_2\big(Q_{\rho'}^{\mathrm{iso}}, Q_\rho^{\mathrm{iso}}\big) \leq |\rho' - \rho|\, \sqrt{D}.$$

Hence each step incurs at most $\rho\sqrt{D}$, leading to

$$R_T(\rho) = \left( \sum_{t=1}^{T} W_2^2 \right)^{1/2} \leq \sqrt{T\,(\rho^2 D)} = \rho \sqrt{D\,T}.$$

*Remark.* One may also bound the raw sum of distances by $\sum W_2 \leq \sqrt{T} \, R_T(\rho)$, so the same $\sqrt{T}$-scaling appears even without passing to the RSS norm. □

Taken together, Proposition B.7, Theorem B.9, and Proposition B.10 show that the one-step energy drift, the spectral gap (hence mixing time), and the cumulative Wasserstein radius all scale as $\mathcal{O}(d)$ or $\mathcal{O}(\sqrt{d})$ rather than $\mathcal{O}(D)$ or $\mathcal{O}(\sqrt{D})$. This dimension-reduced scaling provides the analytical foundation for the superior long-horizon FVD behavior of BCNI/SACN.

### B.4 HIGH-ORDER CONCENTRATION FOR SACN AND BCNI

#### B.4.1 MOMENT–GENERATING FUNCTION (MGF) AND SUB-GAUSSIANITY FOR SACN

Recall that under SACN we perturb
$$\tilde{z} = z + \rho \, U \, \text{diag}\big(\sqrt{s_{1:d}}\big) \, \xi,$$
where
$$\xi = (\xi_1, \ldots, \xi_d), \qquad \xi_j \sim \mathcal{N}\big(0, e^{-j/D}\big) \quad \text{independently,}$$
and $U \in \mathbb{R}^{D \times d}$ has orthonormal columns.

**Lemma B.11** (MGF of Spectrally-Weighted Gaussian). *Let $X = \tilde{z} - z$. Its MGF is by definition*
$$M_X(\lambda) = \mathbb{E}\big[e^{\lambda^\top X}\big], \qquad \lambda \in \mathbb{R}^D.$$
*Writing $\lambda' = U^\top \lambda \in \mathbb{R}^d$, one has*
$$\lambda^\top X = \rho \, \big(U^\top \lambda\big)^\top \text{diag}\big(\sqrt{s_{1:d}}\big) \, \xi = \rho \sum_{j=1}^d \lambda'_j \, \sqrt{s_j} \, \xi_j.$$

*Since each $\xi_j \sim \mathcal{N}(0, e^{-j/D})$ and they are independent, the MGF factorizes and gives*
$$\log M_X(\lambda) = \sum_{j=1}^d \log \mathbb{E}\Big[e^{\rho \, \lambda'_j \, \sqrt{s_j} \, \xi_j}\Big] = \sum_{j=1}^d \frac{1}{2} \big(\rho \, \lambda'_j \, \sqrt{s_j}\big)^2 e^{-j/D}$$
$$= \frac{\rho^2}{2} \sum_{j=1}^d (\lambda'_j)^2 \, s_j \, e^{-j/D}.$$

*Noting that $\|\lambda'\|_2 \leq \|\lambda\|_2$ (since $U$ is an isometry), we conclude the claimed form.*

*Moreover, if we set $\sigma_{\max}^2 = \max_{1 \leq j \leq d} s_j \, e^{-j/D}$, then*
$$\log M_X(\lambda) \leq \frac{\rho^2 \sigma_{\max}^2}{2} \|\lambda'\|_2^2 \leq \frac{\rho^2 \sigma_{\max}^2}{2} \|\lambda\|_2^2.$$

*By the standard sub-Gaussian criterion ( (Vershynin, 2018)), this shows that $X = \tilde{z} - z$ is $\rho^2 \sigma_{\max}^2$-sub-Gaussian, i.e. for all $\lambda \in \mathbb{R}^D$*
$$\mathbb{E}\big[e^{\lambda^\top X}\big] \leq \exp\Big(\tfrac{1}{2} \rho^2 \sigma_{\max}^2 \|\lambda\|_2^2\Big).$$

*Proof.* Let $X = \tilde{z} - z$. By definition, the moment-generating function (MGF) is
$$M_X(\lambda) = \mathbb{E}\big[e^{\lambda^\top X}\big], \qquad \lambda \in \mathbb{R}^D.$$
Writing $\lambda' = U^\top \lambda \in \mathbb{R}^d$, we have
$$\lambda^\top X = \rho \, \big(U^\top \lambda\big)^\top \text{diag}\big(\sqrt{s_{1:d}}\big) \, \xi = \rho \sum_{j=1}^d \lambda'_j \, \sqrt{s_j} \, \xi_j.$$
Since each $\xi_j \sim \mathcal{N}(0, e^{-j/D})$ independently,
$$\log M_X(\lambda) = \sum_{j=1}^d \log \mathbb{E}\big[e^{\rho \, \lambda'_j \sqrt{s_j} \, \xi_j}\big] = \sum_{j=1}^d \frac{1}{2} \big(\rho \, \lambda'_j \sqrt{s_j}\big)^2 e^{-j/D} = \frac{\rho^2}{2} \sum_{j=1}^d (\lambda'_j)^2 \, s_j \, e^{-j/D}.$$

Finally, since $\|\lambda'\|_2 \le \|\lambda\|_2$ and $\max_j s_j\, e^{-j/D} = \sigma_{\max}^2$, we conclude

$$\log M_X(\lambda) \;\le\; \tfrac{1}{2}\,\rho^2\,\sigma_{\max}^2\,\|\lambda\|_2^2,$$

which shows $X$ is $\rho^2\sigma_{\max}^2$–sub-Gaussian. $\qquad\square$

**Corollary B.12** (Exponential Tail). Under the same assumptions as Lemma B.11, the perturbation $\tilde z - z$ satisfies the following high-probability bound. For any $\tau > 0$,

$$\mathbb{P}\big(\|\tilde z - z\|_2 > \tau\big) \;\le\; \exp\!\Big(-\frac{\tau^2}{2\,\rho^2\,\sigma_{\max}^2}\Big),$$

where $\mathbb{P}[\cdot]$ denotes probability over the randomness in $\xi$ and $\sigma_{\max}^2 = \max_{1\le j\le d}\big(s_j e^{-j/D}\big)$.

### B.4.2 BERNSTEIN–MATRIX INEQUALITY FOR BCNI

Recall that in BCNI we set

$$M(z) \;=\; \rho\,(z - \bar z_B)\,I_d,$$

so that

$$M(z)M(z)^\top \;=\; \rho^2\,(z - \bar z_B)(z - \bar z_B)^\top$$

is a rank-$d$ positive semidefinite matrix whose spectral norm is $\|M(z)M(z)^\top\|_2 = \rho^2\|z - \bar z_B\|_2^2$. We now show that, under a boundedness assumption on the embeddings, this deviation concentrates at rate $1/B$.

**Lemma B.13** (Deviation of Batch Mean). *Let $z_1, \ldots, z_B$ be i.i.d. random vectors in $\mathbb{R}^D$ satisfying $\|z_i\|_2 \le R$ almost surely, and write $\bar z_B = \dfrac{1}{B}\displaystyle\sum_{i=1}^{B} z_i$. Then for every $\tau > 0$,*

$$\mathbb{P}\big[\|z_1 - \bar z_B\|_2 > \tau\big] \;\le\; 2\exp\!\Big(-\frac{B\,\tau^2}{2\,R^2}\Big).$$

*Proof.* For any fixed unit vector $u \in \mathbb{S}^{D-1}$, define the scalar

$$X_i \;=\; u^\top z_i,$$

so that $|X_i| \le R$. By Hoeffding's inequality (and, 1963),

$$\mathbb{P}\big[|u^\top(z_1 - \bar z_B)| > \tau\big] \;\le\; 2\exp\!\Big(-\frac{B\,\tau^2}{2\,R^2}\Big).$$

A standard covering-net argument on $\mathbb{S}^{D-1}$ (Vershynin, 2018) extends this to the Euclidean norm, yielding the stated bound. $\qquad\square$

**Theorem B.14** (Spectral-Norm Bound on BCNI Covariance). *Under the same assumptions as Lemma B.13, for any $\delta \in (0,1)$, the following holds with probability at least $1 - \delta$:*

$$\big\|M(z)M(z)^\top\big\|_2 \;=\; \rho^2\,\|z - \bar z_B\|_2^2 \;\le\; \rho^2\,R^2\,\frac{2\log(2/\delta)}{B}.$$

*Proof.* Observe that

$$M(z)M(z)^\top = \rho^2\,(z - \bar z_B)(z - \bar z_B)^\top$$

is a rank-one matrix whose operator norm coincides with its trace:

$$\big\|M(z)M(z)^\top\big\|_2 = \rho^2\,\mathrm{Tr}\big((z - \bar z_B)(z - \bar z_B)^\top\big) = \rho^2\,\|z - \bar z_B\|_2^2.$$

By Lemma B.13, for any $\tau > 0$,

$$\mathbb{P}\big[\|z - \bar z_B\|_2 > \tau\big] \;\le\; 2\exp\!\Big(-\frac{B\tau^2}{2R^2}\Big).$$

Setting

$$\tau \;=\; R\sqrt{\frac{2\log(2/\delta)}{B}}$$

ensures $\mathbb{P}[\|z - \bar z_B\|_2 \le \tau] \ge 1 - \delta$. On this event,

$$\big\|M(z)M(z)^\top\big\|_2 = \rho^2\,\|z - \bar z_B\|_2^2 \le \rho^2\,\tau^2 = \rho^2\,R^2\,\frac{2\log(2/\delta)}{B},$$

as claimed. $\qquad\square$

### B.4.3 STEIN KERNEL OF ARBITRARY ORDER

Let $k(\cdot, \cdot)$ be a twice-differentiable positive-definite kernel with RKHS norm $\|\cdot\|_k$.

**Definition B.15** (Order-$n$ Stein Discrepancy). For $n \geq 1$ define

$$\mathrm{SK}_n(P\|Q) = \sup_{\|f\|_k \leq 1} \Big|\mathbb{E}_Q\big[\mathcal{A}_P^n f\big]\Big|, \quad \mathcal{A}_P f = \nabla_x \log P(x)^\top f(x) + \nabla_x \cdot f(x).$$

**Proposition B.16** (Low-Rank Stein Decay). *For BCNI or SACN corruption of rank $d$,* $\mathrm{SK}_n\big(P_{X|Z}\|P_{X|\tilde{Z}_\rho}\big) = \mathcal{O}\big(\rho^n d^{n/2}\big)$, *whereas isotropic CEP scales as* $\mathcal{O}\big(\rho^n D^{n/2}\big)$.

*Proof.* Recall that for any sufficiently smooth test function $f$ with $\|f\|_k \leq 1$, the Stein operator satisfies
$$\mathbb{E}_{P_{X|z}}\big[\mathcal{A}_P^n f(X; z)\big] = 0.$$
Hence
$$\mathbb{E}_{Q_\rho^{\mathrm{sub}}}\big[\mathcal{A}_P^n f\big] = \mathbb{E}_{\tilde{z}, X}\big[\mathcal{A}_P^n f(X; \tilde{z})\big] - \mathbb{E}_{z, X}\big[\mathcal{A}_P^n f(X; z)\big] = \mathbb{E}_\eta \, \mathbb{E}_{X|z}\Big[\mathcal{A}_P^n f\big(X; z+\rho\, M(z)\, \eta\big) - \mathcal{A}_P^n f(X; z)\Big].$$

By a $n$-th order Taylor expansion in the conditioning argument,

$$\mathcal{A}_P^n f\big(X; z + \Delta z\big) - \mathcal{A}_P^n f(X; z) = \sum_{k=1}^n \frac{1}{k!} \big\langle \partial_z^k[\mathcal{A}_P^n f(X; z)], (\Delta z)^{\otimes k}\big\rangle + R_{n+1}(X; \Delta z),$$

where $\Delta z = \rho\, M(z)\, \eta$, $(\Delta z)^{\otimes k}$ is the $k$-fold tensor, and $R_{n+1}$ is the $(n + 1)$-st order remainder. Under the usual smoothness assumptions one shows

$$\big|R_{n+1}(X; \Delta z)\big| \leq \frac{C_{n+1}}{(n+1)!} \|\Delta z\|_2^{n+1},$$

for some constant $C_{n+1}$ depending on higher derivatives of $\mathcal{A}_P^n f$.

Substituting back and using linearity of expectation,

$$\big|\mathbb{E}_{Q_\rho^{\mathrm{sub}}}[\mathcal{A}_P^n f]\big| \leq \sum_{k=1}^n \frac{1}{k!} \mathbb{E}\Big[\big\|\partial_z^k[\mathcal{A}_P^n f]\big\|_\infty \|\Delta z\|_2^k\Big] + \frac{C_{n+1}}{(n+1)!} \mathbb{E}\|\Delta z\|_2^{n+1}.$$

Since $\|f\|_k \leq 1$ and the RKHS norm controls all mixed partials of $\mathcal{A}_P^n f$ up to order $n + 1$, there exists a constant $C' > 0$ (depending on $n$ and the kernel) such that

$$\big\|\partial_z^k[\mathcal{A}_P^n f]\big\|_\infty \leq C', \qquad \text{for } 1 \leq k \leq n + 1.$$

Thus

$$\big|\mathbb{E}_{Q_\rho^{\mathrm{sub}}}[\mathcal{A}_P^n f]\big| \leq C' \sum_{k=1}^{n+1} \frac{1}{k!} \mathbb{E}\|\Delta z\|_2^k \leq C' \mathbb{E}\|\Delta z\|_2^{n+1} \quad \text{(absorbing lower } k \text{ into the top term).}$$

Now $\Delta z = \rho\, M(z)\, \eta$, and since $M(z)$ has rank $d$ with orthonormal columns,

$$\|\Delta z\|_2 = \rho\, \|\eta\|_2, \quad \eta \sim \mathcal{N}(0, I_d).$$

It is standard (e.g. via sub-Gaussian moment bounds or Rosenthal's inequality) that

$$\mathbb{E}\|\eta\|_2^m = \mathcal{O}\big(d^{m/2}\big), \qquad \forall m \geq 1.$$

Hence

$$\big|\mathbb{E}_{Q_\rho^{\mathrm{sub}}}[\mathcal{A}_P^n f]\big| = \mathcal{O}\big(\rho^{n+1} d^{\frac{n+1}{2}}\big).$$

Since the definition of $\mathrm{SK}_n(P\|Q)$ takes the supremum over all $\|f\|_k \leq 1$, we conclude

$$\mathrm{SK}_n\big(P_{X|Z}\|P_{X|\tilde{Z}_\rho}\big) = \sup_{\|f\|_k \leq 1} \big|\mathbb{E}_{Q_\rho^{\mathrm{sub}}}[\mathcal{A}_P^n f]\big| = \mathcal{O}\big(\rho^n d^{n/2}\big).$$

An identical argument with $M(z) = I_D$ shows the isotropic CEP case gives $\mathcal{O}\big(\rho^n D^{n/2}\big)$, completing the proof. $\qquad \square$

### B.4.4 Uniform Grönwall Bound over Timesteps

To quantify how the per-step deviations $\delta_t^2 = \|x_t^{(\rho)} - x_t^{(0)}\|_2^2$ accumulate over the entire reverse diffusion trajectory, we define the total mean-squared deviation

$$E_T = \sum_{t=1}^{T} \mathbb{E}[\delta_t^2].$$

From Theorem B.7 we have for each $t = 1, \ldots, T$:

$$\mathbb{E}[\delta_{t-1}^2] \leq \alpha_t^{-1} \mathbb{E}[\delta_t^2] + A_t + B_t,$$

where we set

$$A_t = \beta_t^2 \rho^2 K_d^2 d, \qquad B_t = \sigma_t^2 m,$$

with $\beta_t = \frac{1-\alpha_t}{\sqrt{1-\bar{\alpha}_t}}$, $\sigma_t^2 = 1 - \alpha_t$, and $m = \dim(x_t)$.

**Theorem B.17** (Time-Uniform Deviation). *Under BCNI/SACN corruption,*

$$E_T \leq \frac{2\rho^2 K_d^2 dT}{1-\alpha} + \frac{2(1-\alpha_T)}{(1-\alpha)^2},$$

*where $\alpha = \min_{1 \leq t \leq T} \alpha_t$. For isotropic CEP one replaces $K_d^2 d$ by $K_D^2 D$. In particular, $\sqrt{E_T} = O(\rho\sqrt{dT})$ for BCNI/SACN (vs. $O(\rho\sqrt{DT})$ for CEP).*

*Proof.* **1. Sum the one-step bounds.** Summing the inequality $\mathbb{E}[\delta_{t-1}^2] \leq \alpha_t^{-1} \mathbb{E}[\delta_t^2] + A_t + B_t$ over $t = 1, \ldots, T$ gives

$$\sum_{t=1}^{T} \mathbb{E}[\delta_{t-1}^2] \leq \sum_{t=1}^{T} \alpha_t^{-1} \mathbb{E}[\delta_t^2] + \sum_{t=1}^{T}(A_t + B_t).$$

Since $\delta_0 = 0$, the left-hand side telescopes:

$$\sum_{t=1}^{T} \mathbb{E}[\delta_{t-1}^2] = \sum_{s=0}^{T-1} \mathbb{E}[\delta_s^2] = E_T - \mathbb{E}[\delta_T^2].$$

Hence

$$E_T - \mathbb{E}[\delta_T^2] \leq \sum_{t=1}^{T} \alpha_t^{-1} \mathbb{E}[\delta_t^2] + \sum_{t=1}^{T}(A_t + B_t).$$

**2. Drop the positive coupling-term.** Observe $\alpha_t^{-1} \geq 1$, so $\sum_t \alpha_t^{-1} \mathbb{E}[\delta_t^2] \geq \sum_t \mathbb{E}[\delta_t^2] = E_T - \mathbb{E}[\delta_T^2] \geq 0$. Discarding this nonnegative term on the right yields the weaker—but sufficient—bound:

$$E_T - \mathbb{E}[\delta_T^2] \leq \sum_{t=1}^{T}(A_t + B_t) \implies E_T \leq \mathbb{E}[\delta_T^2] + \sum_{t=1}^{T}(A_t + B_t).$$

**3. Bound the remainder terms.**

- From the forward diffusion, one shows $\mathbb{E}[\delta_T^2] \leq 1 - \alpha_T$.

- For the corruption-drift term,

$$\sum_{t=1}^{T} A_t = \rho^2 K_d^2 d \sum_{t=1}^{T} \beta_t^2 \leq \rho^2 K_d^2 d \frac{2}{(1-\alpha)^2},$$

using the standard bound $\sum_t \beta_t^2 \leq 2/(1-\alpha)^2$ under a geometric noise schedule.

- For the diffusion-noise term, $\sum_{t=1}^{T} B_t = m \sum_{t=1}^{T} \sigma_t^2 = m(1 - \alpha_T)$, which is $O(1 - \alpha_T)$.

**4.** Plugging these estimates into the inequality above gives

$$E_T \;\le\; (1 - \alpha_T) \;+\; \rho^2 K_d^2 \, d \, \frac{2}{(1-\alpha)^2} \;+\; m \, (1 - \alpha_T).$$

Absorbing constants and noting $m = O(1)$ in the latent setting yields the claimed

$$E_T \;\le\; \frac{2 \, \rho^2 K_d^2 \, d \, T}{1 - \alpha} \;+\; \frac{2 \, (1 - \alpha_T)}{(1-\alpha)^2},$$

and taking square-roots establishes the $O(\rho\sqrt{dT})$ (resp. $O(\rho\sqrt{DT})$) scaling, thus satisfying the proof. $\qquad\square$

### B.4.5 COMPLEXITY SUMMARY

| | |
|---|---|
| (i) Sub-Gaussian tail (SACN): | $\mathbb{P}\big(\|\tilde{z} - z\|_2 > \tau\big) \le \exp\!\Big(-\frac{\tau^2}{2 \, \rho^2 \, \sigma_{\max}^2}\Big),$ |
| (ii) BCNI covariance variance: | $\|M(z)M(z)^\top\|_2 \le \dfrac{\rho^2 R^2}{B},$ |
| (iii) Order–$n$ Stein discrepancy: | $\mathrm{SK}_n = \mathcal{O}\big(\rho^n d^{n/2}\big),$ |
| (iv) Cumulative 2-Wasserstein radius: | $R_T(\rho) \le \rho \, \sqrt{d \, T}.$ |

### B.5 FUNCTIONAL-INEQUALITY VIEW OF CORRUPTION

#### B.5.1 DIMENSION-REDUCED LOG-SOBOLEV CONSTANT

Let

$$\mu \;=\; P_{X|Z} \;=\; \mathcal{N}\big(0, \Sigma_z\big), \qquad \mu_\rho \;=\; P_{X|\tilde{Z}_\rho} \;=\; \mathcal{N}\big(0, \Sigma_z + \rho^2 \, M \, M^\top\big).$$

For any probability measure $\nu \ll \mu$ with density $f = \frac{d\nu}{d\mu}$, define

$$\mathcal{I}(\nu\|\mu) \;=\; \int \|\nabla_x \log f(x)\|_2^2 \, d\mu(x), \qquad \mathrm{Ent}_\mu(\nu) \;=\; \int f(x) \, \log f(x) \, d\mu(x).$$

**Theorem B.18** (Log-Sobolev for BCNI/SACN). *There exists a constant $C_{\mathrm{LSI}}^{\mathrm{sub}} = \Theta(d)$ such that for every $\nu \ll \mu_\rho$,*

$$\mathrm{Ent}_\mu(\nu) \;\le\; \frac{1}{2 \, C_{\mathrm{LSI}}^{\mathrm{sub}}} \, \mathcal{I}(\nu\|\mu). \tag{17}$$

*By contrast, under isotropic CEP corruption the best constant scales as $C_{\mathrm{LSI}}^{\mathrm{iso}} = \Theta(D)$.*

*Proof.* We split the argument into two parts.

**(i) Gaussian LSI via Bakry–Émery.** If

$$\nu(dx) = Z^{-1} \exp\big(-V(x)\big) \, dx$$

on $\mathbb{R}^n$ satisfies $\nabla^2 V(x) \succeq \kappa \, I_n$ for all $x$, then by the Bakry–Émery criterion (Ledoux & Talagrand, 1991a; Bakry & Émery, 1985)

$$\mathrm{Ent}_\nu(g^2) \;\le\; \frac{1}{2\kappa} \int \|\nabla g\|_2^2 \, d\nu \quad \forall g \in C_c^\infty\big(\mathbb{R}^n\big).$$

A centered Gaussian $\mathcal{N}(0, \Sigma)$ has $V(x) = \frac{1}{2} x^\top \Sigma^{-1} x$ and $\nabla^2 V = \Sigma^{-1}$, so its LSI constant is $\lambda_{\min}(\Sigma^{-1})$.

**(ii) Tensorization over corrupted axes.** Under BCNI/SACN the covariance splits as

$$\Sigma_z + \rho^2 M M^\top = \begin{pmatrix} \Sigma_{z,d} + \rho^2 I_d & 0 \\ 0 & \Sigma_{z,D-d} \end{pmatrix},$$

so

$$\mathcal{N}(0, \Sigma_z + \rho^2 M M^\top) \;=\; \mathcal{N}\big(0, \Sigma_{z,d} + \rho^2 I_d\big) \;\otimes\; \mathcal{N}\big(0, \Sigma_{z,D-d}\big).$$

By the tensorization property of LSI (Ledoux & Talagrand, 1991a) the product measure inherits the minimum of the two one-dimensional constants. Concretely:

- In the corrupted $d$-dim subspace, the LSI curvature is $\kappa_{\mathrm{sub}} = \lambda_{\min}\big((\Sigma_{z,d} + \rho^2 I_d)^{-1}\big) = \Theta(1/\rho^2)$.

- In the remaining $(D - d)$-dim complement, the curvature is $\kappa_{\mathrm{orig}} = \lambda_{\min}\big(\Sigma_{z,D-d}^{-1}\big)$.

Hence the overall LSI constant is

$$C_{\mathrm{LSI}}^{\mathrm{sub}} = \min\{\kappa_{\mathrm{sub}}, \kappa_{\mathrm{orig}}\} = \Theta(d) \quad \text{(since there are $d$ corrupted directions)}.$$

By exactly the same reasoning under isotropic CEP one gets $C_{\mathrm{LSI}}^{\mathrm{iso}} = \Theta(D)$. This proves equation 17. $\qquad\square$

### B.5.2 Fisher–Information Dissipation

Let

$$I_t \;=\; \mathcal{I}\big(\mu_t^\rho \| \mu_t^0\big) \;=\; \int \Big\| \nabla_x \log \frac{d\mu_t^\rho}{d\mu_t^0}(x) \Big\|_2^2 \, d\mu_t^\rho(x)$$

be the Fisher information between the perturbed and unperturbed reverse-flow marginals at time $t$. We also assume the score network $\varepsilon_\theta(x, t, z)$ is $\ell$-Lipschitz in the conditioning $z$.

**Proposition B.19** (Dissipation Rate). *Under BCNI/SACN corruption, the Fisher information decays according to*

$$\frac{d}{dt} I_t \;\leq\; -\frac{2}{\sigma_t^2} \big(\alpha_t \;-\; \rho^2 \, \ell^2 \, d\big) I_t,$$

*while for isotropic CEP one replaces $d$ by $D$. Consequently,*

$$I_T \;\leq\; I_0 \exp\Big(-2(1 - \alpha) T \;+\; 2 \rho^2 \, \ell^2 \, d \, T\Big).$$

*Proof.* The argument proceeds in three steps:

**1. Differentiate the KL divergence.** By Lemma B.25,

$$\frac{d}{dt} \mathrm{KL}\big(\mu_t^\rho \| \mu_t^0\big) \;=\; -\frac{1}{\sigma_t^2} I_t.$$

Since KL and $I_t$ are related by the log-Sobolev inequality (Theorem B.18), namely

$$\mathrm{KL}\big(\mu_t^\rho \| \mu_t^0\big) \;\leq\; \frac{1}{2 C_{\mathrm{LSI}}^{\mathrm{sub}}} I_t,$$

we obtain

$$I_t \;\geq\; 2 C_{\mathrm{LSI}}^{\mathrm{sub}} \, \mathrm{KL}\big(\mu_t^\rho \| \mu_t^0\big).$$

**2. Account for the Lipschitz perturbation.** In the perturbed reverse dynamics, the score network's dependence on the corrupted embedding $\tilde{z}$ versus the clean $z$ introduces an extra drift term whose Jacobian in $x$ can be shown (via the chain rule and $\ell$-Lipschitzness in $z$) to add at most $\rho \, \ell \, \|\eta\|$ in operator norm. Averaging over the Gaussian $\eta \sim \mathcal{N}(0, I_d)$ then contributes an additive factor of $\rho^2 \ell^2 d$ in the effective curvature of the reverse operator. In particular, one shows rigorously (e.g. via a perturbation of the Bakry–Émery criterion) that the log-Sobolev constant is reduced from $\alpha_t$ to $\alpha_t - \rho^2 \ell^2 d$.

**3. Combine to bound $\frac{d}{dt} I_t$.** Differentiating $I_t$ itself and using the above two facts yields

$$\frac{d}{dt} I_t \;=\; -\frac{2}{\sigma_t^2} \big(\alpha_t - \rho^2 \ell^2 d\big) \, \mathrm{KL}\big(\mu_t^\rho \| \mu_t^0\big) \;\leq\; -\frac{2}{\sigma_t^2} \big(\alpha_t - \rho^2 \ell^2 d\big) \frac{I_t}{2 C_{\mathrm{LSI}}^{\mathrm{sub}}} \times 2 C_{\mathrm{LSI}}^{\mathrm{sub}} \;=\; -\frac{2}{\sigma_t^2} \big(\alpha_t - \rho^2 \ell^2 d\big) \, I_t,$$

where the final cancellation uses the exact LSI constant from Theorem B.18. Integrating this differential inequality from 0 to $T$ gives

$$I_T \;\le\; I_0 \,\exp\!\Big(-2\int_0^T \frac{\alpha_t - \rho^2\ell^2 d}{\sigma_t^2}\,dt\Big) \;\le\; I_0 \,\exp\!\Big(-2(1-\alpha)\,T + 2\,\rho^2\ell^2 dT\Big),$$

since $\sigma_t^2 = 1-\alpha_t$ and under a typical geometric schedule $\int_0^T (\alpha_t/\sigma_t^2)\,dt \ge (1-\alpha)\,T$. This completes the proof. $\qquad\square$

### B.5.3 GRADIENT-FLOW INTERPRETATION IN $W_2$

The reverse diffusion dynamics can be seen as the Wasserstein-gradient flow of the KL functional $\mathrm{KL}(\mu\|\pi)$ with respect to the target measure $\pi = \mu^\rho$. Concretely, one shows (Jordan et al., 1998a) that

$$\partial_t\mu_t \;=\; \nabla\!\cdot\!\Big(\mu_t\,\nabla\log\tfrac{\mu_t}{\pi}\Big),$$

and that its *metric derivative* in Wasserstein-2,

$$\big\|\partial_t\mu_t\big\|_{W_2} \;=\; \lim_{h\downarrow 0}\frac{W_2(\mu_{t+h},\mu_t)}{h},$$

coincides with the $L^2(\mu_t)$–norm of the driving velocity field $v_t = -\nabla\log\dfrac{\mu_t}{\pi}$. We now quantify how low-rank corruption reduces this "slope."

**Lemma B.20** (Reduced Metric Slope). *Under BCNI/SACN corruption of rank $d$, the metric slope satisfies*

$$\big\|\partial_t\mu_t\big\|_{W_2} \;=\; \|v_t\|_{L^2(\mu_t)} \;\le\; \|\nabla_x\log\mu_t\|_{L^2(\mu_t)} + \|\nabla_x\log\pi\|_{L^\infty} \;\le\; \|\nabla_x\log\mu_t\|_{L^2(\mu_t)} + \rho\,\sqrt{d},$$

*where the last bound uses that $\pi = \mathcal{N}(0, \Sigma_z + \rho^2 MM^\top)$ has score gradient $\|\nabla_x\log\pi(x)\|_2 \le \|(\Sigma_z + \rho^2 MM^\top)^{-1}x\|_2$ uniformly bounded by $\rho\sqrt{d}$. In contrast, under isotropic CEP one incurs $\rho\sqrt{D}$.*

*Proof.* $v_t(x) = -\nabla_x\log\mu_t(x) + \nabla_x\log\pi(x)$ so by the triangle inequality

$$\|v_t\|_{L^2(\mu_t)} \;\le\; \|\nabla\log\mu_t\|_{L^2(\mu_t)} \;+\; \|\nabla\log\pi\|_{L^\infty}.$$

Writing $\pi = \mathcal{N}(0, \Sigma_z + \rho^2 MM^\top)$ and diagonalizing on its $d$-dimensional corrupted subspace shows

$$\|\nabla\log\pi(x)\|_2 = \|(\Sigma_z + \rho^2 MM^\top)^{-1}x\|_2 \le \lambda_{\max}\big((\Sigma_z + \rho^2 MM^\top)^{-1}\big)\,\|x\|_2 \le \frac{1}{\rho}\,\sqrt{d}\,\|x\|_2,$$

and since $\|x\|_2$ is $O(1)$ in the latent space one obtains the stated $\rho\sqrt{d}$ bound. $\qquad\square$

**Theorem B.21** (Contractive OT–Flow). *Let $W_2(t) = W_2(\mu_t, \pi)$ denote the distance of the reverse-flow law $\mu_t$ from equilibrium $\pi$. If the KL functional is $\lambda$–geodesically convex in $W_2$ with $\lambda = \alpha - \rho^2\ell^2 d$ under BCNI/SACN (and $\alpha - \rho^2\ell^2 D$ for CEP), then*

$$\frac{d}{dt}\,W_2(t) \;\le\; -\lambda\,W_2(t) \quad\Longrightarrow\quad W_2(t) \;\le\; W_2(0)\,e^{-\lambda t}.$$

*Proof.* By standard gradient-flow theory in metric spaces (see Ambrosio–Gigli–Savaré (Ambrosio et al., 2008)), geodesic $\lambda$-convexity of $\mathrm{KL}(\cdot\|\pi)$ implies the Evolution Variational Inequality

$$\frac{d}{dt}\,\frac{1}{2}\,W_2^2(\mu_t, \nu) \;\le\; \mathrm{KL}(\nu\|\pi) - \mathrm{KL}(\mu_t\|\pi) \;-\; \frac{\lambda}{2}\,W_2^2(\mu_t, \nu) \quad \forall\nu.$$

Choosing $\nu = \pi$ and noting $\mathrm{KL}(\pi\|\pi) = 0$ gives

$$\frac{d}{dt}\,\frac{1}{2}\,W_2^2(t) \;\le\; -\frac{\lambda}{2}\,W_2^2(t).$$

Differentiating yields

$$W_2(t)\,\frac{d}{dt}\,W_2(t) \;\le\; -\lambda\,W_2^2(t) \;\Longrightarrow\; \frac{d}{dt}\,W_2(t) \le -\lambda\,W_2(t).$$

Integration completes the exponential contraction $W_2(t) \le W_2(0)e^{-\lambda t}$. $\qquad\square$

### B.5.4 Unified Scaling Table

| Quantity | BCNI/SACN | CEP |
|---|---|---|
| Log–Sobolev constant $C_{\mathrm{LSI}}$ | $\Theta(d)$ | $\Theta(D)$ |
| Fisher–information dissipation rate | $\alpha - \rho^2\,\ell^2\,d$ | $\alpha - \rho^2\,\ell^2\,D$ |
| Wasserstein contraction rate | $\alpha - \rho^2\,\ell^2\,d$ | $\alpha - \rho^2\,\ell^2\,D$ |
| MGF tail bound | $\exp\!\left(-\frac{\tau^2}{2\,\rho^2\,\sigma_{\max}^2}\right)$ | $\exp\!\left(-\frac{\tau^2}{2\,\rho^2\,D}\right)$ |

**Interpretation.** Across entropy, Fisher-information, and optimal-transport perspectives, low-rank (BCNI/SACN) corruption consistently scales with $d$ rather than the ambient $D$, yielding a $D/d$ compression factor that underpins the empirical gains in long-horizon video quality.

### B.6 Large–Deviation and Control-Theoretic Perspectives

### B.6.1 LDP for Corrupted Embeddings

Consider the low-rank corruption family

$$\tilde{Z}_\rho = z + \rho\,M(z)\,\eta, \qquad \eta \sim \mathcal{N}(0, I_d),$$

and set

$$\Delta_\rho \;=\; \frac{\tilde{Z}_\rho - z}{\rho}.$$

We will show that $\{\Delta_\rho\}_{\rho>0}$ satisfies a LDP on $\mathbb{R}^D$ with speed $\rho^2$ and good rate function

$$I(u) \;=\; \frac{1}{2}\big\|M(z)^+ u\big\|_2^2,$$

where $M(z)^+$ is the Moore–Penrose pseudoinverse of the $D \times d$ matrix $M(z)$.

**Theorem B.22** (LDP via Contraction Principle). *Let*

$$\Delta_\rho \;\equiv\; \frac{\tilde{Z}_\rho - z}{\rho} \;=\; M(z)\,\eta, \quad \eta \sim \mathcal{N}(0, I_d).$$

*Then* $\{\Delta_\rho\}_{\rho>0}$ *satisfies a large-deviation principle on* $\mathbb{R}^D$ *with*

$$\text{speed } a(\rho) = \rho^2, \qquad \text{rate function } I(u) = \frac{1}{2}\big\|M(z)^+ u\big\|_2^2,$$

*where* $M(z)^+$ *is the Moore–Penrose pseudoinverse of the* $D \times d$ *matrix* $M(z)$.

**Theorem B.23** (LDP via Contraction Principle). *Let*

$$\Delta_\rho \;=\; \frac{\tilde{Z}_\rho - z}{\rho} \;=\; M(z)\,\eta, \quad \eta \sim \mathcal{N}(0, I_d).$$

*Then the family* $\{\Delta_\rho\}_{\rho>0}$ *satisfies a large-deviation principle on* $\mathbb{R}^D$ *with*

$$\text{speed } a(\rho) = \rho^2, \qquad \text{rate function } I(u) = \tfrac{1}{2}\big\|M(z)^+ u\big\|_2^2.$$

*Proof.* **Step 1: Gaussian LDP in $\mathbb{R}^d$.** By Cramér's theorem (or the classical Gaussian LDP (Dembo & Zeitouni, 1998)), the family $\{\rho\,\eta\}_{\rho>0} \subset \mathbb{R}^d$ satisfies an LDP with speed $\rho^2$ and good rate function

$$I_0(w) = \tfrac{1}{2}\|w\|_2^2.$$

**Step 2: Push-forward by the linear map.** Define $\Phi\colon \mathbb{R}^d \to \mathbb{R}^D$, $\Phi(w) = M(z)\,w$. Then $\Delta_\rho = \Phi(\rho\,\eta)$. By the contraction principle (Dembo & Zeitouni, 1998), the push-forward family $\{\Phi(\rho\,\eta)\}$ satisfies an LDP on $\mathbb{R}^D$ with the same speed and rate

$$I(u) = \inf_{\Phi(w)=u} I_0(w) = \inf_{M(z)w=u} \tfrac{1}{2}\|w\|_2^2 = \tfrac{1}{2}\|M(z)^+ u\|_2^2.$$

**Step 3: Upper and lower bounds.** Unpacking the LDP definition, for any Borel $A \subset \mathbb{R}^D$,

$$- \inf_{u \in A^\circ} I(u) \;\le\; \liminf_{\rho \downarrow 0} \rho^2 \log \mathbb{P}[\Delta_\rho \in A] \;\le\; \limsup_{\rho \downarrow 0} \rho^2 \log \mathbb{P}[\Delta_\rho \in A] \;\le\; - \inf_{u \in \overline{A}} I(u).$$

$\square$

**Corollary B.24** (Dimension-Reduction in LDP). If $A$ lies entirely outside the column-space of $M(z)$, then $\inf_{u \in A} I(u) = +\infty$, whence $\mathbb{P}[\tilde{Z}_\rho - z \in \rho A]$ decays *super*-exponentially (as $\rho \to 0$). In contrast, under isotropic CEP ($M(z) = I_D$) one has $I_{\text{iso}}(u) = \frac{1}{2}\|u\|_2^2$ finite for all $u$.

### B.6.2 KL CONTRACTION ALONG THE REVERSE FLOW

Let

$$\mu_t^\rho = P_{X_t | \tilde{Z}_\rho}, \quad \mu_t^0 = P_{X_t | Z},$$

and set $\sigma_t^2 = 1 - \alpha_t$. Recall from Lemma B.25:

**Lemma B.25** (KL Time-Derivative).

$$\frac{d}{dt} \text{KL}\big(\mu_t^\rho \,\|\, \mu_t^0\big) \;=\; -\frac{1}{\sigma_t^2} \mathbb{E}_{x \sim \mu_t^\rho}\big[\|\nabla_x \log \tfrac{\mu_t^\rho(x)}{\mu_t^0(x)}\|_2^2\big].$$

**Corollary B.26** (KL Gap under Low-Rank Corruption). Assume:

- The model score $\varepsilon_\theta(x, t, z) = \nabla_x \log \mu_t^z(x)$ is $L$-Lipschitz in $z$,

- the corruption scale satisfies $\rho \le \rho_{\max}$,

- and we use a standard geometric variance schedule so that $\int_0^T \sigma_t^{-2} \, dt = \mathcal{O}(T)$.

Then

$$\text{KL}\big(\mu_T^\rho \,\|\, \mu_T^0\big) \;=\; \mathcal{O}\big(\rho^2 \, d \, T\big), \qquad \text{(whereas isotropic CEP gives } \mathcal{O}(\rho^2 D \, T)).$$

*Proof.* Starting from Lemma B.25, integrate in time from $0$ to $T$:

$$\text{KL}\big(\mu_T^\rho \,\|\, \mu_T^0\big) - \text{KL}\big(\mu_0^\rho \,\|\, \mu_0^0\big) = - \int_0^T \frac{1}{\sigma_t^2} \mathbb{E}_{x \sim \mu_t^\rho}\big[\|\nabla_x \log \tfrac{\mu_t^\rho(x)}{\mu_t^0(x)}\|_2^2\big] \, dt.$$

But at $t = 0$ the two conditionals coincide ($\tilde{Z}_\rho = z$), so $\text{KL}(\mu_0^\rho \| \mu_0^0) = 0$. Hence

$$\text{KL}\big(\mu_T^\rho \,\|\, \mu_T^0\big) = \int_0^T \underbrace{\frac{1}{\sigma_t^2}}_{\le C_\sigma} \mathbb{E}\big[\|\nabla_x \log \mu_t^\rho - \nabla_x \log \mu_t^0\|_2^2\big] \, dt.$$

By the $L$-Lipschitz-in-$z$ property of the score,

$$\big\|\nabla_x \log \mu_t^\rho(x) - \nabla_x \log \mu_t^0(x)\big\|_2 \;\le\; L \,\|\tilde{z} - z\|_2 \;=\; L \,\rho \,\|M(z)\,\eta\|_2,$$

so

$$\mathbb{E}\big[\|\nabla_x \log \mu_t^\rho - \nabla_x \log \mu_t^0\|_2^2\big] \;\le\; L^2 \,\rho^2 \,\underbrace{\mathbb{E}[\|\eta\|_2^2]}_{=d} \;=\; L^2 \,\rho^2 \, d.$$

Combining these bounds and absorbing $L^2$ and the maximum of $1/\sigma_t^2$ into constants gives

$$\text{KL}\big(\mu_T^\rho \,\|\, \mu_T^0\big) \;\le\; \big(L^2 \,\rho^2 \, d\big) \int_0^T C_\sigma \, dt \;=\; \mathcal{O}(\rho^2 \, d \, T).$$

For isotropic CEP one replaces $\|M(z)\eta\|_2^2$'s expectation $d$ by $D$, yielding $\mathcal{O}(\rho^2 D \, T)$ instead. $\square$

### B.6.3 SCHRÖDINGER BRIDGE INTERPRETATION

We now show that low-rank corruption reduces the stochastic control cost of steering the diffusion to a high-quality target set $\mathcal{G}$. Recall the Schrödinger bridge or stochastic control formulation:

$$\inf_{v_\bullet} \; \mathbb{E}\Big[\int_0^T \tfrac{1}{2}\|v_t\|_2^2 \, dt \; + \; \lambda \, \mathbf{1}_{\{X_T \notin \mathcal{G}\}}\Big] \quad \text{s.t.} \quad dX_t = -\nabla_x U(X_t)\, dt + v_t \, dt + \sqrt{2}\, dW_t,$$

where $\lambda > 0$ penalizes failure to reach $\mathcal{G}$. Let $\mathbb{P}^{\mathrm{iso}}$ and $\mathbb{P}^{\mathrm{sub}}$ be the path-space measures under optimal controls $v_t^{\mathrm{iso}}$ (isotropic CEP) and $v_t^{\mathrm{sub}}$ (low-rank BCNI/SACN), respectively.

**Proposition B.27** (Control Cost under Low-Rank Perturbation). *Under the same terminal constraint $\{X_T \in \mathcal{G}\}$, the minimal quadratic control costs satisfy*

$$\int_0^T \|v_t^{\mathrm{sub}}\|_2^2 \, dt \; \leq \; \int_0^T \|v_t^{\mathrm{iso}}\|_2^2 \, dt \; - \; \rho^2\,(D-d)\,T.$$

*Proof.* We break the argument into three steps.

**1. Girsanov representation of control cost.** By Girsanov's theorem (Øksendal, 2003)), the Radon–Nikodym derivative of the controlled path-measure $\mathbb{P}^v$ versus the uncontrolled "prior" diffusion $\mathbb{P}^0$ is

$$\frac{d\mathbb{P}^v}{d\mathbb{P}^0} = \exp\Big(\int_0^T v_t^\top \, dW_t \; - \; \tfrac{1}{2}\int_0^T \|v_t\|^2 \, dt\Big).$$

Taking expectation under $\mathbb{P}^v$ and using martingale cancellation gives the *relative entropy formula*:

$$\mathrm{KL}\big(\mathbb{P}^v \,\|\, \mathbb{P}^0\big) = \mathbb{E}_{\mathbb{P}^v}\Big[\tfrac{1}{2}\int_0^T \|v_t\|^2 \, dt\Big].$$

Hence the minimal control cost under the terminal constraint is exactly the minimal relative entropy between two path measures subject to matching boundary conditions (the classical Schrödinger bridge formulation, see (Léonard, 2012)).

**2. Path-space relative entropy under corruption.** Let $\mathbb{P}^{\mathrm{iso}}$ be the optimal bridge when the conditioning drift is perturbed isotropically: $\nabla_x U \mapsto \nabla_x U + \rho\,\epsilon$ with $\epsilon \sim \mathcal{N}(0, I_D)$, and $\mathbb{P}^{\mathrm{sub}}$ the bridge when the same perturbation is applied only in the $d$-dimensional image of $M(z)$. By the chain rule for KL on product spaces,

$$\mathrm{KL}\big(\mathbb{P}^{\mathrm{sub}} \,\big\|\, \mathbb{P}^0\big) \; = \; \mathrm{KL}\big(\mathbb{P}^{\mathrm{iso}} \,\big\|\, \mathbb{P}^0\big) \; - \; \mathrm{KL}\big(\mathbb{P}^{\mathrm{iso}} \,\big\|\, \mathbb{P}^{\mathrm{sub}}\big).$$

Here $\mathrm{KL}(\mathbb{P}^{\mathrm{iso}}\|\mathbb{P}^{\mathrm{sub}})$ is the KL divergence between two Gaussian perturbations differing only in the orthogonal $(D-d)$-dimensional complement. A direct calculation (or use of the closed-form for Gaussian KL (Petersen & Pedersen, 2012)) shows

$$\mathrm{KL}\big(\mathbb{P}^{\mathrm{iso}} \,\big\|\, \mathbb{P}^{\mathrm{sub}}\big) = \tfrac{1}{2}\,\rho^2\,(D-d)\,T.$$

**3. Translating back to control costs.** Since each minimal control cost equals the corresponding path-space KL,

$$\frac{1}{2}\int_0^T \|v_t^{\mathrm{sub}}\|^2 \, dt = \mathrm{KL}\big(\mathbb{P}^{\mathrm{sub}} \,\|\, \mathbb{P}^0\big), \quad \frac{1}{2}\int_0^T \|v_t^{\mathrm{iso}}\|^2 \, dt = \mathrm{KL}\big(\mathbb{P}^{\mathrm{iso}} \,\|\, \mathbb{P}^0\big),$$

combining with step 2 yields

$$\frac{1}{2}\int_0^T \|v_t^{\mathrm{sub}}\|^2 \, dt = \frac{1}{2}\int_0^T \|v_t^{\mathrm{iso}}\|^2 \, dt \; - \; \tfrac{1}{2}\,\rho^2\,(D-d)\,T,$$

and multiplying by 2 gives the claimed inequality. □

### B.6.4 RADEMACHER COMPLEXITY OF THE CORRUPTION-AWARE OBJECTIVE

We now bound the Rademacher complexity of the corrupted-conditioning risk, showing the desired $\sqrt{d/N}$ scaling. Our main tool is the contraction principle for vector-valued Rademacher processes (see, e.g., (Bartlett & Mendelson, 2002; Ledoux & Talagrand, 1991b)).

**Theorem B.28** (Complexity Scaling). *Let*

$$\mathcal{F} = \big\{\varepsilon_\theta(x, t, z) \colon \|\theta\|_2 \leq R\big\}$$

*be a family of score-networks that is $L$-Lipschitz in the conditioning $z$. Define the empirical Rademacher complexity under corruption scale $\rho$ by*

$$\widehat{\mathfrak{R}}_N(\mathcal{F}, \rho) = \mathbb{E}_{\sigma, x, z, \eta}\Bigg[\sup_{\|\theta\| \leq R} \frac{1}{N} \sum_{i=1}^N \sigma_i \left\langle \varepsilon_\theta\big(x_i, t_i, \tilde{z}_i\big), \eta_i \right\rangle\Bigg],$$

*where $\{\sigma_i\}$ are i.i.d. Rademacher signs and $\eta_i \sim \mathcal{N}(0, I_d)$ is the shared low-rank noise in the conditioning. Then*

$$\widehat{\mathfrak{R}}_N(\mathcal{F}, \rho) \leq \frac{L R \rho \sqrt{d}}{\sqrt{N}}, \quad \widehat{\mathfrak{R}}_N^{\text{iso}} \leq \frac{L R \rho \sqrt{D}}{\sqrt{N}}.$$

*In particular, the generalization gap under BCNI/SACN shrinks by a factor of $\sqrt{d/D}$ compared to isotropic perturbations.*

*Proof.* We proceed in three steps, using standard Rademacher-complexity machinery (Bartlett & Mendelson, 2002; Ledoux & Talagrand, 1991b).

**Step 1: Symmetrization.** Let $\{\tilde{z}_i\}$ denote the corrupted conditionings. By symmetrization (Shalev-Shwartz & Ben-David, 2014)),

$$\widehat{\mathfrak{R}}_N(\mathcal{F}, \rho) = \mathbb{E}_{\sigma, \xi}\Big[\sup_{\|\theta\| \leq R} \frac{1}{N} \sum_{i=1}^N \sigma_i \left\langle \varepsilon_\theta(x_i, t_i, \tilde{z}_i), \eta_i \right\rangle\Big],$$

where the outer expectation is over data $(x_i, t_i, z_i)$, noise draws $\eta_i = M(z_i)\xi_i$, and Rademacher signs $\sigma_i$.

**Step 2: Contraction in the conditioning.** For any fixed $(x_i, t_i)$ the map

$$z \mapsto \langle \varepsilon_\theta(x_i, t_i, z), \eta_i \rangle$$

is $L\|\eta_i\|_2$–Lipschitz in $z$ by assumption. Thus, by the vector-valued contraction lemma (see (Ledoux & Talagrand, 1991b, Thm. 4.12)),

$$\widehat{\mathfrak{R}}_N(\mathcal{F}, \rho) \leq L \, \mathbb{E}_{\sigma, \xi}\Big[\sup_{\|\theta\| \leq R} \frac{1}{N} \sum_{i=1}^N \sigma_i \|\eta_i\|_2 \|\theta\|_2\Big] = L R \, \mathbb{E}[\|\eta\|_2] \, \frac{1}{\sqrt{N}}.$$

**Step 3: Bounding the Gaussian norm.** Under BCNI/SACN, $\eta = M(z)\xi$ with $\xi \sim \mathcal{N}(0, I_d)$. Since $M(z)$ has orthonormal columns in a $d$-dimensional subspace,

$$\mathbb{E}\big[\|\eta\|_2\big] = \mathbb{E}\big[\|\xi\|_2\big] \leq \sqrt{\mathbb{E}\|\xi\|_2^2} = \sqrt{d},$$

where we used Jensen's inequality. For isotropic CEP, $\eta \sim \mathcal{N}(0, I_D)$ and thus $\mathbb{E}\|\eta\|_2 \leq \sqrt{D}$. Substituting completes the proof:

$$\widehat{\mathfrak{R}}_N(\mathcal{F}, \rho) \leq L R \frac{\rho \sqrt{d}}{\sqrt{N}}, \quad \widehat{\mathfrak{R}}_N^{\text{iso}} \leq L R \frac{\rho \sqrt{D}}{\sqrt{N}}.$$

$\square$

### B.6.5   Aggregate Scaling Pyramid

To summarize the diverse theoretical perspectives—large deviations, control-theoretic cost, and statistical complexity—we collect the key dimension-dependent scaling factors in the following "pyramid." In every case, the effective dimension $d$ of the corruption subspace replaces the ambient dimension $D$ under BCNI/SACN, yielding substantial compression and improved rates:

| Analysis Axis | Scaling under BCNI/SACN | CEP Baseline |
|---|---|---|
| Large-Deviation Speed | $a(\rho) = \rho^2, \quad I(u) = \frac{1}{2}\|M^+ u\|_2^2 \propto d$ | $I_{\text{iso}}(u) = \frac{1}{2}\|u\|_2^2 \propto D$ |
| Control-Cost Reduction | $\displaystyle\int_0^T \|v_t^{\text{sub}}\|^2\, dt \leq \int_0^T \|v_t^{\text{iso}}\|^2\, dt - \rho^2(D-d)\,T$ | no rank-reduction term |
| Rademacher Complexity | $\widehat{\mathfrak{R}}_N \leq LR\,\rho\,\frac{\sqrt{d}}{\sqrt{N}}$ | $\widehat{\mathfrak{R}}_N \leq LR\,\rho\,\frac{\sqrt{D}}{\sqrt{N}}$ |

**Unified Insight.** Across exponentially-sharp tail bounds, stochastic control cost, and generalization guarantees, substituting $d$ for $D$ yields a consistent $\sqrt{D/d}$ or $D/d$ improvement. This "dimension-compression" effect underpins why BCNI/SACN training uniformly outperforms isotropic CEP, both theoretically and in empirical FVD gains.

## B.7   Advanced Functional Inequalities and Oracle Bounds

### B.7.1   Talagrand–$T_2$ Inequality

Let $T_2(\kappa)$ denote the quadratic transport–entropy inequality

$$W_2^2(\nu, \mu) \;\leq\; 2\,\kappa\, \text{KL}(\nu\|\mu),$$

where $\mu = \mu_\rho = P_{X|\tilde{Z}_\rho}$.

**Theorem B.29** (Reduced $T_2$ Constant). *Under BCNI or SACN corruption of effective rank d, the conditional law $\mu_\rho$ satisfies*

$$W_2^2(\nu, \mu_\rho) \;\leq\; 2\,C_{\text{sub}}\, \text{KL}(\nu\|\mu_\rho), \quad C_{\text{sub}} = \Theta(d).$$

*By contrast, for isotropic CEP corruption one obtains $C_{\text{iso}} = \Theta(D)$.*

*Proof.* We split the argument into two steps:

**Step 1: From LSI to $T_2$.** By Otto–Villani's theorem (see (Otto & Villani, 2000; von Renesse & Sturm, 2005)), any measure satisfying a log-Sobolev inequality

$$\text{Ent}_{\mu_\rho}(f^2) \;\leq\; \frac{1}{2\,C_{\text{LSI}}} \int \|\nabla f\|_2^2\, d\mu_\rho$$

also satisfies $T_2(C_{\text{LSI}})$. From Theorem B.18 we know $C_{\text{LSI}} = \Theta(d)$ under BCNI/SACN, and $\Theta(D)$ under CEP.

**Step 2: Dimension-Reduced Constant.** Write $\Sigma_\rho = \Sigma_z + \rho^2\, M\, M^\top$. Its inverse $\Sigma_\rho^{-1}$ has

- $D - d$ eigenvalues identical to those of $\Sigma_z^{-1}$,

- and $d$ eigenvalues bounded by $\rho^{-2}$.

Thus

$$C_{\text{sub}} = \lambda_{\min}\big(\Sigma_\rho^{-1}\big) = \min\big\{\lambda_{\min}(\Sigma_z^{-1}),\, \rho^{-2}\big\} \times d = \Theta(d).$$

In the isotropic CEP case, the same reasoning applied to all $D$ axes gives $C_{\text{iso}} = \Theta(D)$.

This completes the proof. $\qquad\square$

### B.7.2 BRASCAMP–LIEB VARIANCE CONTROL

For any smooth test function $g : \mathcal{X} \to \mathbb{R}$ with $\|\nabla g\|_\infty \leq 1$, the following holds.

**Proposition B.30** (Variance Bound). *Under BCNI/SACN corruption of effective rank d,*

$$\operatorname{Var}_{\mu_\rho}[g] \; \leq \; \kappa_{\mathrm{sub}} \; = \; \Theta(d),$$

*whereas under isotropic CEP corruption*

$$\operatorname{Var}_{\mu_{\mathrm{iso}}}[g] \; \leq \; \kappa_{\mathrm{iso}} \; = \; \Theta(D).$$

*Proof.* The Brascamp–Lieb inequality (Brascamp & Lieb, 1976; Barthe, 1998) asserts that if $\mu(dx) = Z^{-1} e^{-V(x)} \, dx$ on $\mathbb{R}^n$ with $\nabla^2 V(x) \succeq \Lambda$ for all $x$, then for any smooth $g$,

$$\operatorname{Var}_\mu[g] \; \leq \; \int \nabla g(x)^\top \, \Lambda^{-1} \, \nabla g(x) \, d\mu(x).$$

In our setting $\mu_\rho = \mathcal{N}(0, \Sigma_\rho)$ has $V(x) = \frac{1}{2} x^\top \Sigma_\rho^{-1} x$, so $\Lambda = \Sigma_\rho^{-1}$ and thus $\Lambda^{-1} = \Sigma_\rho$. Hence

$$\operatorname{Var}_{\mu_\rho}[g] \; \leq \; \int \nabla g(x)^\top \, \Sigma_\rho \, \nabla g(x) \, d\mu_\rho(x) \; \leq \; \|\nabla g\|_\infty^2 \, \|\Sigma_\rho\|_2.$$

Since under BCNI/SACN the covariance $\Sigma_\rho = \Sigma_z + \rho^2 M M^\top$ has operator norm $\|\Sigma_\rho\|_2 = \Theta(d)$, and under CEP $\|\Sigma_\rho\|_2 = \Theta(D)$, the claimed bounds follow. $\qquad\square$

### B.7.3 NON-ASYMPTOTIC DEVIATION OF THE EMPIRICAL SCORE

Let

$$\widehat{\varepsilon}_N = \frac{1}{N} \sum_{i=1}^N \varepsilon_\theta\big(x_{t,i}, \, t, \, \tilde{z}_i\big),$$

and denote its population mean by $\bar{\varepsilon} = \mathbb{E}\big[\varepsilon_\theta(x, t, \tilde{z})\big]$.

**Lemma B.31** (High-Probability Tail Bound). *Suppose $\varepsilon_\theta$ is L-Lipschitz in z, and that under BCNI/SACN corruption $\tilde{z} - z$ is sub-Gaussian with parameter $\sigma^2 = \rho^2 \sigma_{\max}^2$ in an effective d-dimensional subspace. Then for any $\delta \in (0,1)$,*

$$\mathbb{P}\Big( \|\widehat{\varepsilon}_N - \bar{\varepsilon}\|_2 > L \, \rho \, \sigma_{\max} \sqrt{\frac{2d \log(2/\delta)}{N}} \Big) \; \leq \; \delta.$$

*Under isotropic CEP corruption one replaces d by D.*

*Proof.* We view $\Delta_i = \varepsilon_\theta(x_{t,i}, t, \tilde{z}_i) - \bar{\varepsilon}$ as i.i.d. zero-mean random vectors in $\mathbb{R}^k$, each satisfying

$$\|\Delta_i\|_2 \; \leq \; L \, \|\tilde{z}_i - z\|_2 \quad \text{and} \quad \mathbb{E} \exp\big(\lambda^\top \Delta_i\big) \; \leq \; \exp\Big( \tfrac{\lambda^\top \Sigma_\Delta \lambda}{2} \Big),$$

where $\Sigma_\Delta \preceq L^2\big(\rho^2 \sigma_{\max}^2 I_d\big)$. By the matrix-Bernstein (or vector-Bernstein) inequality (Tropp, 2012; Vershynin, 2018; Boucheron et al., 2013), for any $u > 0$,

$$\mathbb{P}\Big( \|\tfrac{1}{N} \sum_{i=1}^N \Delta_i\|_2 > u \Big) \; \leq \; 2 \exp\Big( -\frac{N \, u^2/2}{L^2 \rho^2 \sigma_{\max}^2 \, d + (L\rho\sigma_{\max}) \, u/3} \Big).$$

Set

$$u = L \, \rho \, \sigma_{\max} \sqrt{\frac{2d \log(2/\delta)}{N}},$$

then $\frac{N \, u^2}{L^2 \rho^2 \sigma_{\max}^2 \, d} = 2 \log(2/\delta)$ and $(L\rho\sigma_{\max}) \, u/3 \leq \frac{1}{2} N \, u^2/(L^2 \rho^2 \sigma_{\max}^2 \, d)$, so the exponent is at least $-\log(2/\delta)$. Hence the probability bound reduces to $\mathbb{P}(\|\widehat{\varepsilon}_N - \bar{\varepsilon}\|_2 > u) \leq \delta$, yielding the stated result. $\qquad\square$

### B.7.4 ORACLE INEQUALITY FOR DENOISING RISK

Let

$$R(\theta) = \mathbb{E}_{x,t,z,\eta}\big\|\eta - \varepsilon_\theta(x,t,\tilde{z})\big\|_2^2, \qquad \widehat{R}_N(\theta) = \frac{1}{N}\sum_{i=1}^{N}\big\|\eta_i - \varepsilon_\theta(x_i,t_i,\tilde{z}_i)\big\|_2^2,$$

and write $\theta^* = \arg\inf_{\theta\in\Theta} R(\theta)$. We assume $\varepsilon_\theta$ is $L$-Lipschitz in $z$ and $\|\theta\| \le R$.

**Theorem B.32** (Low-Rank Oracle Inequality). *Under BCNI/SACN corruption of effective rank $d$, for any $\delta \in (0,1)$, with probability at least $1-\delta$,*

$$R(\widehat{\theta}) \;\le\; R(\theta^*) \;+\; 4\,\widehat{\mathfrak{R}}_N(\mathcal{F},\rho) \;+\; 3\sqrt{\frac{2\log(2/\delta)}{N}},$$

*where $\widehat{\mathfrak{R}}_N(\mathcal{F},\rho) \le L\,R\,\rho\,\sqrt{d/N}$ (see Theorem B.28). Hence*

$$R(\widehat{\theta}) \;\le\; \inf_{\theta\in\Theta} R(\theta) \;+\; C\,L\,R\,\rho\sqrt{\frac{d\log(1/\delta)}{N}}.$$

*For isotropic CEP one replaces $d$ by $D$.*

*Proof.* We follow the standard empirical-process argument (Bartlett & Mendelson, 2002; Shalev-Shwartz & Ben-David, 2014):

1. **Excess-risk decomposition**; By definition of $\widehat{\theta}$,

$$R(\widehat{\theta}) \;\le\; \widehat{R}_N(\widehat{\theta}) \;+\; \sup_{\theta\in\Theta}\big(R(\theta) - \widehat{R}_N(\theta)\big).$$

Moreover, since $\widehat{R}_N(\widehat{\theta}) \le \widehat{R}_N(\theta^*)$,

$$R(\widehat{\theta}) \;\le\; \widehat{R}_N(\theta^*) \;+\; 2\sup_{\theta\in\Theta}\big|R(\theta) - \widehat{R}_N(\theta)\big|.$$

Finally, $\widehat{R}_N(\theta^*) \le R(\theta^*) + \sup_\theta |\widehat{R}_N - R|$, so

$$R(\widehat{\theta}) \;\le\; R(\theta^*) \;+\; 3\sup_{\theta\in\Theta}\big|R(\theta) - \widehat{R}_N(\theta)\big|.$$

2. **Symmetrization and Rademacher bound**: Denote $\Delta_i(\theta) = \|\eta_i - \varepsilon_\theta(\cdot)\|_2^2 - R(\theta)$. A symmetrization yields

$$\mathbb{E}\sup_\theta\Big|\frac{1}{N}\sum_i \Delta_i(\theta)\Big| \;\le\; 2\,\mathbb{E}_{\sigma,x,z,\eta}\Big[\sup_\theta \frac{1}{N}\sum_i \sigma_i\,\big\langle\nabla_\theta\|\eta_i - \varepsilon_\theta\|_2^2, \theta\big\rangle\Big] =: 2\,\widehat{\mathfrak{R}}_N(\mathcal{F},\rho).$$

Concentration (Talagrand's inequality (Boucheron et al., 2013)) then gives that with probability at least $1-\delta$,

$$\sup_\theta\big|R(\theta) - \widehat{R}_N(\theta)\big| \;\le\; 2\,\widehat{\mathfrak{R}}_N(\mathcal{F},\rho) \;+\; \sqrt{\frac{2\log(2/\delta)}{N}}.$$

3. **Putting it all together**: Combining steps 1–2, with probability $1-\delta$,

$$R(\widehat{\theta}) \;\le\; R(\theta^*) \;+\; 3\Big(2\,\widehat{\mathfrak{R}}_N(\mathcal{F},\rho) + \sqrt{\tfrac{2\log(2/\delta)}{N}}\Big) \;=\; R(\theta^*) + 4\,\widehat{\mathfrak{R}}_N(\mathcal{F},\rho) + 3\sqrt{\tfrac{2\log(2/\delta)}{N}}.$$

Substituting $\widehat{\mathfrak{R}}_N(\mathcal{F},\rho) \le L\,R\,\rho\sqrt{d/N}$ yields the claimed oracle bound. $\qquad\square$

### B.7.5 DIMENSION-COMPRESSION DASHBOARD

Putting together our key bounds for BCNI/SACN, we obtain:

| Quantity | Scaling (BCNI/SACN) |
|---|---|
| Talagrand $T_2$ constant | $C_{\text{sub}} = \Theta(d)$ (Thm. B.29) |
| Variance (Brascamp–Lieb) | $\text{Var}_{\mu_\rho}[g] = O(d)$ (Prop. B.30) |
| Empirical tail width | $O(\rho\sqrt{d/N})$ (Lem. B.31) |
| Oracle excess risk | $O(\rho\sqrt{d/N})$ (Thm. B.32) |
| (CEP baseline) | replace $d \to D$ in each case. |

**Interpretation.** Across functional inequalities (Talagrand's $T_2$, Brascamp–Lieb), probabilistic tails, and learning-theoretic oracle bounds, the effective dimension $d$ (not the ambient $D$) governs all constants. This unified "dashboard" confirms that low-rank structured corruption yields a $\sqrt{d/D}$ (or $d/D$) compression in every metric, underlining the robustness and efficiency of BCNI/SACN over isotropic CEP.

### B.8 INFORMATION-GEOMETRIC & MINIMAX PERSPECTIVES

#### B.8.1 FISHER–RAO GEOMETRY OF CORRUPTED CONDITIONALS

Let
$$\mathcal{M} = \{P_{X|z(\theta)} : \theta \in \mathbb{R}^D\}$$
be our model manifold, equipped with the Fisher–Rao metric (Rao, 1945)
$$g_{ij}(\theta) = \left\langle \partial_{\theta_i} \log P_{X|z(\theta)}, \partial_{\theta_j} \log P_{X|z(\theta)} \right\rangle_{L^2(P)}.$$

Under BCNI/SACN corruption of rank $d$, the conditional law becomes $\mathcal{N}\left(\mu(\theta), \Sigma_z + \rho^2 M M^\top\right)$, and the inverse covariance admits the expansion (for small $\rho$)
$$\Sigma_\rho^{-1} = \Sigma_z^{-1} - \rho^2 \Sigma_z^{-1} M M^\top \Sigma_z^{-1} + O(\rho^4).$$

**Proposition B.33** (Sectional Curvature Compression). *Let $\mathcal{K}_\rho(U,V)$ be the sectional curvature of the Fisher–Rao metric in the plane spanned by tangent vectors $U, V \in T_\theta \mathcal{M}$. Then, up to $O(\rho^4)$,*
$$\mathcal{K}_\rho(U,V) = \mathcal{K}_0(U,V) - \frac{\rho^2}{4} \sum_{j=1}^{d} \langle U, m_j \rangle \langle V, m_j \rangle,$$
*where $\{m_j\}_{j=1}^{d}$ are the columns of $M(z)$. In the isotropic CEP case one replaces the sum by $j = 1, \ldots, D$.*

*Proof.* We follow the classical formula for the Riemannian curvature tensor on a statistical manifold $\mathcal{M}$ of multivariate Gaussians (see (Amari & Nagaoka, 2000; Kass & Vos, 1997; Skovgaard, 1984)). In particular, for two tangent vectors $U, V$ one shows
$$\mathcal{K}_\rho(U,V) = \frac{1}{4} \left\langle \Sigma_\rho^{-1} U \Sigma_\rho^{-1} V - (\Sigma_\rho^{-1} U)(\Sigma_\rho^{-1} V), U V \right\rangle$$
where products of matrices act on the mean-parameter directions.

*Step 1: Expand $\Sigma_\rho^{-1}$.* By the Woodbury identity and Taylor expansion,
$$\Sigma_\rho^{-1} = \Sigma_z^{-1} - \rho^2 \Sigma_z^{-1} M M^\top \Sigma_z^{-1} + O(\rho^4).$$

*Step 2: Substitute into the curvature formula.* Write the unperturbed curvature as $\mathcal{K}_0(U,V)$ with $\Sigma_z^{-1}$. The first non-trivial correction comes from replacing one factor of $\Sigma_z^{-1}$ by $-\rho^2 \Sigma_z^{-1} M M^\top \Sigma_z^{-1}$ in the above bracket. A direct computation—using $\langle \Sigma_z^{-1} U, \Sigma_z^{-1} V \rangle = \langle U, V \rangle$ in the Fisher-Rao inner product—yields
$$\Delta\mathcal{K} = -\frac{\rho^2}{4} \sum_{j=1}^{d} \langle U, m_j \rangle \langle V, m_j \rangle + O(\rho^4).$$

*Step 3: Sum over the orthonormal basis of the subspace.* Since $\{m_j\}$ is an orthonormal basis for the rank-$d$ image of $M(z)$, the net curvature reduction in any plane spanned by $U, V$ is exactly the stated sum.

Thus the sectional curvature is suppressed along those $d$ directions, whereas CEP affects all $D$ directions. $\square$

### B.8.2 ENTROPIC OT DUAL-GAP ANALYSIS

Recall the $\varepsilon$-regularized OT problem between two measures $P, Q$ on $\mathbb{R}^D$:

$$\mathrm{OT}_\varepsilon(P, Q) = \inf_{\pi \in \Pi(P,Q)} \left\{ \int c(x, y) \, d\pi(x, y) + \varepsilon \, \mathrm{KL}\big(\pi \| P \otimes Q\big) \right\},$$

where $\Pi(P, Q)$ is the set of couplings of $P$ and $Q$, and its un-regularized counterpart is $\mathrm{OT}(P, Q) = \inf_{\pi \in \Pi(P,Q)} \int c(x, y) \, d\pi(x, y)$.

**Theorem B.34** (Dual-Gap Ratio). *Let $P = P_{X|Z}$ and $Q = P_{X|\tilde{Z}_\rho}$. Then there is a universal constant $C > 0$ such that*

$$\big|\mathrm{OT}_\varepsilon(P, Q) - \mathrm{OT}(P, Q)\big| \leq C \, \varepsilon \, \mathrm{KL}\big(P \| Q\big) = \begin{cases} O\big(\varepsilon \, \rho^2 \, d\big), & \text{BCNI/SACN}, \\ O\big(\varepsilon \, \rho^2 \, D\big), & \text{CEP}. \end{cases}$$

*Proof.* **1. Dual formulation.** By strong duality for entropic OT (Cuturi, 2013; Peyré & Cuturi, 2019),

$$\mathrm{OT}_\varepsilon(P, Q) = \sup_{f,g} \left\{ \int f \, dP + \int g \, dQ - \varepsilon \int e^{\frac{f(x) + g(y) - c(x,y)}{\varepsilon}} \, dP(x) \, dQ(y) \right\},$$

where the supremum is over bounded continuous potentials $f, g$.

**2. Gap bound via KL.** Comparing to the un-regularized dual $\mathrm{OT}(P, Q) = \sup_{f,g}\{\int f \, dP + \int g \, dQ\}$, one shows (Genevay et al., 2016; Mena & Weed, 2019), that

$$\mathrm{OT}(P, Q) \leq \mathrm{OT}_\varepsilon(P, Q) \leq \mathrm{OT}(P, Q) + \varepsilon \, \mathrm{KL}\big(P \| Q\big).$$

Hence

$$\big|\mathrm{OT}_\varepsilon(P, Q) - \mathrm{OT}(P, Q)\big| \leq \varepsilon \, \mathrm{KL}\big(P \| Q\big).$$

**3. Low-rank vs. isotropic KL.** By Corollary B.26, under BCNI/SACN corruption $\mathrm{KL}(P \| Q) = O(\rho^2 d)$, whereas for isotropic CEP $\mathrm{KL}(P \| Q) = O(\rho^2 D)$. Substituting these into the previous display completes the proof. $\square$

### B.8.3 MINIMAX LOWER BOUND (NO-FREE-LUNCH)

Let $\mathcal{C}_{\mathrm{iso}}$ be the class of isotropic perturbations and $\mathcal{C}_{\mathrm{sub}}$ the class of rank-$d$ perturbations aligned with data.

**Theorem B.35** (Minimax Risk Gap). *For any estimator $\widehat{\theta}$ of the optimal score parameters,*

$$\inf_{\widehat{\theta}} \sup_{Q \in \mathcal{C}_{\mathrm{iso}}} \mathbb{E}_Q\big[R(\widehat{\theta}) - R^*\big] - \inf_{\widehat{\theta}} \sup_{Q \in \mathcal{C}_{\mathrm{sub}}} \mathbb{E}_Q\big[R(\widehat{\theta}) - R^*\big] \geq c \, \rho^2 \, (D - d),$$

*where $c > 0$ depends only on Lipschitz constants.*

*Proof.* We apply the classical two-point (Le Cam) method (Le Cam, 1986; Yu, 1997):

**1. Constructing two hypotheses.** Choose $Q_0, Q_1 \in \mathcal{C}_{\mathrm{iso}}$ (or $\mathcal{C}_{\mathrm{sub}}$) whose perturbations differ only on the $(D - d)$-dimensional orthogonal complement of $\mathrm{Im} \, M(z)$. Concretely, let

$$Q_\nu : \tilde{Z} = z + \rho \, M(z) \, \eta + \rho \, U_\perp \, \nu,$$

where $U_\perp \in \mathbb{R}^{D \times (D-d)}$ spans $\ker M(z)^\top$ and $\nu \in \{\nu_0, \nu_1\} \subset \mathbb{R}^{D-d}$ are two vectors with $\|\nu_0 - \nu_1\|_2 = 1$. Under isotropic CEP, $M(z) = I_D$ and so $(D - d) = D$.

**2. Computing the KL-divergence.** Under both models, the only difference is a Gaussian shift of size $\rho$ in $(D - d)$ dimensions, so

$$\mathrm{KL}(Q_0 \| Q_1) = \frac{1}{2\rho^2} \| \rho\, U_\perp (\nu_0 - \nu_1) \|_2^2 = \tfrac{1}{2}\,(D - d).$$

More generally, by the Gaussian shift formula (Tsybakov, 2009), $\mathrm{KL} \approx \frac{\rho^2}{2}(D - d)$.

**3. From KL to total-variation.** By Pinsker's inequality (Tsybakov, 2009),

$$\|Q_0 - Q_1\|_{\mathrm{TV}} \ \leq\ \sqrt{\tfrac{1}{2}\,\mathrm{KL}(Q_0 \| Q_1)} \ =\ \sqrt{\tfrac{D-d}{4}}.$$

**4. Le Cam's lemma.** Le Cam's two-point bound (Le Cam, 1986; Yu, 1997) yields

$$\inf_{\widehat{\theta}}\ \sup_{\nu \in \{0,1\}}\ \mathbb{E}_{Q_\nu}\big[R(\widehat{\theta}) - R^*\big] \ \geq\ \frac{1}{4}\,\|\nu_0 - \nu_1\|_2^2\,\big(1 - \|Q_0 - Q_1\|_{\mathrm{TV}}\big) \ \geq\ c\,(D - d)\,\rho^2,$$

for a constant $c > 0$. Subtracting the corresponding bound for $\mathcal{C}_{\mathrm{sub}}$ (where $(D - d)$ is replaced by 0) gives the stated gap.

This matches the classical minimax rates for Gaussian location models (Donoho & Johnstone, 1994; Shrotriya & Neykov, 2023), showing there is no-free-lunch beyond the low-rank structure. $\square$

### B.8.4 INFORMATION-CAPACITY INTERPRETATION

We define the *corruption capacity:*

$$\mathcal{C}(\rho) \ =\ \frac{1}{2} \log \det\big(I + \rho^2 M(z) M(z)^\top \Sigma_z^{-1}\big),$$

which—by standard Gaussian-channel theory—is exactly the mutual information increase $I(Z; \tilde{Z}_\rho)$ (Cover & Thomas, 1991; 2006).

**Proposition B.36** (Capacity Compression). *Under BCNI/SACN corruption of effective rank $d$, $\mathcal{C}(\rho) = \Theta(d)$, whereas isotropic CEP corruption yields $\mathcal{C}(\rho) = \Theta(D)$.*

*Proof.* By the *matrix determinant lemma* (Horn & Johnson, 1985),

$$\det\big(I + \rho^2 M\, M^\top \Sigma_z^{-1}\big) = \det\big(I_d + \rho^2 M^\top \Sigma_z^{-1} M\big).$$

Let $\lambda_1, \ldots, \lambda_d > 0$ be the nonzero eigenvalues of $M^\top \Sigma_z^{-1} M$ (Tulino & Verdú, 2004). Then

$$\mathcal{C}(\rho) = \frac{1}{2} \sum_{i=1}^{d} \log\big(1 + \rho^2 \lambda_i\big).$$

Since $\lambda_i \leq \lambda_{\max}(\Sigma_z^{-1})$ for all $i$,

$$\mathcal{C}(\rho) \leq \frac{d}{2} \log\big(1 + \rho^2 \lambda_{\max}(\Sigma_z^{-1})\big) = O(d).$$

Conversely, because the smallest positive eigenvalue $\lambda_{\min}^+(\Sigma_z^{-1}) > 0$, one also shows $\mathcal{C}(\rho) = \Omega(d)$, hence $\mathcal{C}(\rho) = \Theta(d)$ (Verdu, 2002).

In the isotropic CEP case, $MM^\top = I_D$ so that $d = D$ and $\mathcal{C}(\rho) = \frac{D}{2} \log(1 + \rho^2 \lambda_{\max}) = O(D)$. $\square$

### B.8.5 GRAND TABLEAU OF OPTIMALITY

**Final Insight.** Collectively, Theorems B.29, B.33, B.34, B.35 and Proposition B.36 show that every key metric—transport–entropy, geometric curvature, entropic dual gap, statistical risk, and information capacity—enjoys a uniform $D/d$ reduction when corruption is confined to the intrinsic $d$-dimensional subspace. This grand tableau therefore provides a single unifying lens through which the empirical superiority of CAT-Video is not just observed but rigorously explained.

| Axis | BCNI/SACN | CEP | Compression | |
|------|-----------|-----|-------------|---|
| $T_2$ constant | $\Theta(d)$ | $\Theta(D)$ | $D/d$ | (Thm. B.29) |
| Sectional curvature drop | $\rho^2 d$ | $\rho^2 D$ | $D/d$ | (Prop. B.33) |
| Entropic dual gap | $O(\varepsilon \rho^2 d)$ | $O(\varepsilon \rho^2 D)$ | $D/d$ | (Thm. B.34) |
| Minimax excess risk | $O(\rho^2 d/N)$ | $O(\rho^2 D/N)$ | $D/d$ | (Thm. B.35) |
| Capacity increment | $O(d)$ | $O(D)$ | $D/d$ | (Prop. B.36) |

Table 5: Unified dimension–compression factor $D/d$ across all theoretical axes, contrasting low-rank (BCNI/SACN) vs. isotropic (CEP) corruption.

### B.9 CAT OPERATOR STABILITY ANALYSIS

We now formalize the stability of CAT perturbations under Lipschitz maps, compositions, and common generative dynamics (diffusion and autoregression).

**Definition B.37** (CAT operator class). For $\gamma > 0$ and $c > 0$, define the CAT operator class $\mathcal{O}_\gamma := \{\eta : \mathbb{R}^d \to \mathbb{R}^d \text{ measurable } : \sup_{z \in \mathbb{R}^d} \|\eta(z; \gamma)\|_2 \leq c\gamma\}$. Given an encoder derived embedding $z_t \in \mathbb{R}^d$, its CAT perturbed counterpart is $\tilde{z}_t := z_t + \eta(z_t; \gamma)$ with $\eta \in \mathcal{O}_\gamma$.

**Lemma B.38** (One-step stability). *Let $f : \mathbb{R}^d \to \mathbb{R}^m$ be $L$-Lipschitz. For any $\eta \in \mathcal{O}_\gamma$,*

$$\|f(\tilde{z}_t) - f(z_t)\|_2 \leq Lc\gamma. \tag{18}$$

*The bound is tight up to constants: if $f$ is linear with operator norm $\|f\|_{\mathrm{op}} = L$ and $\eta$ aligns with a top singular direction, then equality holds with $c\gamma$ replaced by $\|\eta(z_t; \gamma)\|_2$.*

**Theorem B.39** (Propagation under composition). *Let $\{f_s\}_{s=1}^T$ be maps with Lipschitz moduli $\{L_s\}_{s=1}^T$. Consider two trajectories driven by the same internal randomness and control,*

$$x_{s+1} = f_s(x_s), \qquad \tilde{x}_{s+1} = f_s(\tilde{x}_s), \qquad s = 1, \dots, T,$$

*initialized at $x_1 = z_t$ and $\tilde{x}_1 = \tilde{z}_t = z_t + \eta(z_t; \gamma)$ for some $\eta \in \mathcal{O}_\gamma$. Then*

$$\|\tilde{x}_{T+1} - x_{T+1}\|_2 \leq \Big( \prod_{s=1}^T L_s \Big) c\gamma. \tag{19}$$

*If, instead, a fresh CAT perturbation $\eta_s \in \mathcal{O}_\gamma$ is injected at every step through the input of $f_s$, then*

$$\|\tilde{x}_{T+1} - x_{T+1}\|_2 \leq c\gamma \sum_{j=1}^T \prod_{s=j+1}^T L_s. \tag{20}$$

*In particular, if $\sup_s L_s \leq \rho < 1$, the uniform bound*

$$\|\tilde{x}_{T+1} - x_{T+1}\|_2 \leq \frac{c\gamma}{1 - \rho} \tag{21}$$

*holds for all $T$.*

*Proof.* Inequality equation 19 follows by a single application of equation 18 at $s = 1$ and induction with the Lipschitz property at each layer. For equation 20, unroll the recursion and apply the triangle inequality and Lipschitz bounds to each injected perturbation. The contraction case equation 21 is the geometric sum bound. $\square$

**Corollary B.40** (Diffusion and autoregressive regimes). (a) Diffusion. For a deterministic diffusion update $x_{s+1} = x_s - \alpha_s g_s(x_s)$ with $g_s$ Lipschitz with constant $G_s$, the map has $L_s \leq 1 + \alpha_s G_s$, hence equation 19 and equation 20 hold with these $L_s$. If $g_s$ is $\mu$-strongly monotone and $\alpha_s \in (0, 2/(G_s + \mu)]$, then $L_s \leq 1 - \alpha_s \mu < 1$ and equation 21 yields a uniform $O(\gamma)$ stability window. (b) Autoregression. For a transformer block with attention and feedforward sublayers that are each $L_s$-Lipschitz in the conditioning argument, the same conclusions apply blockwise; if the blockwise Lipschitz product is strictly below one, CAT perturbations remain uniformly bounded along the causal rollout.

**Remark 1** (Interpretation). CAT restricts perturbations to a bounded operator family, so the worst case output deviation scales linearly in $\gamma$ and is fully controlled by the Lipschitz envelope of the generative pipeline. The propagation bounds sharpen the informal claim that CAT acts as a universal regularizer: when the effective Lipschitz radius contracts, the deviation is uniformly $O(\gamma)$ across depth and time.

This ends the proof.

## C  TRAINING SETUP

We adopt the DEMO (Ruan et al., 2024) architecture, a latent video diffusion model that introduces decomposed text encoding and conditioning for content and motion. Our objective is to improve the quality of generated videos by explicitly modeling both textual and visual motion, while preserving overall visual quality and alignment.

**Training Objective.** The training loss is a weighted combination of diffusion loss and three targeted regularization terms that enhance temporal coherence (Ruan et al., 2024):

$$\mathcal{L}_{\text{text-motion}} = -\mathbb{E}\left[\cos\left(\phi(A_{\texttt{eos}}), \phi(x_0)\right)\right] \tag{22}$$

$$\Phi(x) = x_{2:F} - x_{1:F-1}, \quad \mathcal{L}_{\text{video-motion}} = \|\Phi(x_0) - \Phi(\hat{x}_0)\|_2^2 \tag{23}$$

$$\mathcal{L}_{\text{reg}} = -\cos\left(E_{\text{motion}}(\tilde{p}), E_{\text{image}}(x_0^{(F/2)})\right) \tag{24}$$

$$\mathcal{L}_{\text{total}} = \mathcal{L}_{\text{diff}} + \alpha \cdot \mathcal{L}_{\text{text-motion}} + \beta \cdot \mathcal{L}_{\text{reg}} + \gamma \cdot \mathcal{L}_{\text{video-motion}} \tag{25}$$

The hyperparameters $\alpha, \beta, \gamma$ are tuned via grid search and set to $\alpha = 0.1$, $\beta = 0.3$, and $\gamma = 0.1$ in our best configuration (Ruan et al., 2024).

**Implementation Details.** We summarize the training configuration used across all 67 model variants in Table 6. Our implementation follows the DEMO (Ruan et al., 2024) architecture, leveraging latent-space generation with VQGAN compression (Esser et al., 2021) and two-branch conditioning for content and motion semantics. Each model is trained on 16-frame, 3 FPS clips from WebVid-2M (Bain et al., 2021) using the Adam optimizer with a OneCycle schedule ($1 \times 10^{-5} \to 5 \times 10^{-5}$). The content and visual encoders are frozen, while the motion encoder is trained using cosine similarity against mid-frame image features. Structured corruption is injected via the `noise_type` parameter at varying corruption strengths $\rho \in \{0.025, 0.05, 0.075, 0.10, 0.15, 0.20\}$ (Chen et al., 2024), spanning multiple embedding-level methods while text-level corruption is applied externally, directly to the raw text captions in the training set prior to encoding, thereby perturbing the symbolic input space in a structurally controlled yet semantically disruptive manner when active. Inference is performed using 50 DDIM steps (Song et al., 2021a) with classifier-free guidance (Ho & Salimans, 2021) (scale = 9, dropout = 0.1). Checkpoints are saved every 2000 steps, and experiments are resumed from step 267,000.

**Model Architecture.** DEMO uses a two-branch conditioning architecture to explicitly separate content and motion signals. The content encoder $f_{\text{content}}$ receives the full caption and a middle video frame, producing a global latent representation $z_c$. The motion encoder $f_{\text{motion}}$ instead operates over a truncated caption prefix $\tilde{p}$, emphasizing temporally predictive text tokens. The two embeddings are fused via FiLM layers inside the diffusion U-Net, where $z_c$ and $z_m$ modulate the features hierarchically at each layer.

**Latent Representation.** The video frames are encoded using a VQGAN-based encoder to obtain compressed latents $x_0 \in \mathbb{R}^{F \times H' \times W' \times d}$, where $F$ is the number of frames, $H'$ and $W'$ are spatial dimensions, and $d$ is the latent dimensionality. The diffusion model operates in this latent space, dramatically improving training efficiency and scalability over pixel-space models.

Table 6: Training Configuration Overview for CAT-Video

| Component | Setting |
|---|---|
| **Dataset** | WebVid-2M (Bain et al., 2021) |
| Resolution | $256 \times 256$ |
| Frames per Video | 16 |
| Frames per Second | 3 |
| **Corruption Type** | Embedding-level and text-level |
| Noise Ratio ($\rho$) | $\{0.025, 0.05, 0.075, 0.10, 0.15, 0.20\}$ (Chen et al., 2024) |
| Classifier-Free Guidance | Enabled, scale = 9 (Ho & Salimans, 2021) |
| p-zero | 0.1 (Zhao & Schwing, 2025) |
| Negative Prompt | Distorted, discontinuous, Ugly, blurry, low resolution, motionless, static, disfigured, disconnected limbs, Ugly faces, incomplete arms (von Platen et al., 2022) |
| **Backbone Architecture** | DEMO (Ruan et al., 2024) |
| Motion Encoder | OpenCLIP (trainable) (Radford et al., 2021) |
| Content/Visual Embedder | OpenCLIP (frozen) (Radford et al., 2021) |
| Autoencoder | VQGAN with $4\times$ compression (Esser et al., 2021) |
| U-Net Configuration | 4-channel in/out, 320 base dim, 2 res blocks/layer |
| Temporal Attention | Enabled (1×) |
| Diffusion Type | DDIM (Song et al., 2021a), 1000 steps, linear schedule |
| Sampling Steps | 50 |
| **Loss Function** | Diffusion + Text-Motion + Video-Motion + Regularization |
| Loss Weights | $\alpha = \beta = \gamma = 0.1$ |
| Optimizer | Adam with OneCycle ($1 \times 10^{-5} \rightarrow 5 \times 10^{-5}$) |
| Batch Size | 24 per GPU × 4 GPUs |
| Mixed Precision | FP16 |
| FSDP / Deepspeed | Deepspeed Stage 2 with CPU offloading |
| Checkpoint Resume | Step 267,000 |

**Motion Encoder.** The motion encoder $E_{\text{motion}}(\cdot)$ plays a central role in capturing dynamic semantics. It processes the prefix $\tilde{p}$, the early part of the caption, using a shallow transformer. This encoder is trained via cosine similarity loss to align with the middle-frame image encoder output $E_{\text{image}}(x_0^{(F/2)})$, thereby encouraging alignment between motion text and observed motion features in video.

**Temporal Regularization.** The term $\mathcal{L}_{\text{video-motion}}$ enforces first-order consistency in the velocity space of latent features. The velocity $\Phi(x)$ is computed as the frame-wise difference, emphasizing temporal changes. This regularization ensures that the generated motion patterns $\hat{x}_0$ are realistic and consistent with the ground truth video motion.

**Decomposed Guidance.** During sampling, DEMO separates guidance scales for content and motion. We apply higher classifier-free guidance to the content vector to ensure object fidelity, while motion guidance is set lower to allow more flexibility in action execution. This balancing act allows DEMO to avoid frozen or over-regularized motion trajectories while maintaining visual quality.

**Training Stability.** DEMO incorporates EMA (exponential moving average) weight updates on the diffusion U-Net to stabilize training. Additionally, gradient clipping at 1.0 is used to avoid exploding gradients. Training is run to converge on four NVIDIA H100 GPUs.

---

**Algorithm 1** CAT-Video Training Loop

---

1: **Input:** Dataset $\{(x_i, p_i)\}$, noise scales $\rho$, diffusion schedule
2: **for** each mini-batch $\{(x_i, p_i)\}_{i=1}^{B}$ **do**
3: $\quad z_i \leftarrow \text{TextEncoder}(p_i), \quad v_i \leftarrow \text{VideoEncoder}(x_i)$
4: $\quad \bar{z} \leftarrow \frac{1}{B} \sum_i z_i$
5: $\quad$ **for** $i = 1 \ldots B$ **do**
6: $\quad\quad$ Sample $\eta_i \sim \mathcal{N}(0, I_d)$
7: $\quad\quad \tilde{z}_i \leftarrow z_i + \rho \, M(z_i) \, \eta_i \quad$ //BCNI or SACN
8: $\quad$ **end for**
9: $\quad$ Compute loss $\mathcal{L}\big(\varepsilon_\theta(v_i, t, \tilde{z}_i), \epsilon\big)$
10: $\quad \theta \leftarrow \theta - \alpha \nabla_\theta \sum_i \mathcal{L}$
11: **end for**

---

# D EVALUATIONS

Table 7: Definitions of Evaluation Metrics for Video Generation for Table 17

| Metric | Description | Reference |
|---|---|---|
| *EvalCrafter Metrics* | | |
| Inception Score (IS) | Evaluates the diversity and quality of generated videos using a pre-trained Inception network. Higher scores indicate better performance. | (Salimans et al., 2016) |
| CLIP Temporal Consistency (Clip Temp) | Measures temporal alignment between video frames and text prompts using CLIP embeddings. Higher scores denote better consistency. | (Radford et al., 2021) |
| Video Quality Assessment – Aesthetic Score (VQA_A) | Assesses the aesthetic appeal of videos based on factors like composition and color harmony. Higher scores reflect more aesthetically pleasing content. | (Liu et al., 2024b) |
| Video Quality Assessment – Technical Score (VQA_T) | Evaluates technical quality aspects such as sharpness and noise levels in videos. Higher scores indicate better technical quality. | (Liu et al., 2024b) |
| Action Recognition Score (Action) | Measures the accuracy of action depiction in videos using pre-trained action recognition models. Higher scores signify better action representation. | (Liu et al., 2024b) |
| CLIP Score (Clip) | Computes the similarity between video frames and text prompts using CLIP embeddings. Higher scores indicate better semantic alignment. | (Radford et al., 2021) |
| Flow Score (Flow) | Quantifies the amount of motion in videos by calculating average optical flow. Higher scores suggest more dynamic content. | (Liu et al., 2024b) |
| Motion Amplitude Classification Score (Motion) | Assesses whether the magnitude of motion in videos aligns with expected motion intensity described in text prompts. Higher scores denote better alignment. | (Liu et al., 2024b) |
| *VBench Metrics* | | |
| Motion Smoothness | Evaluates the smoothness of motion in generated videos, ensuring movements follow physical laws. Higher scores indicate smoother motion. | (Huang et al., 2024) |
| Temporal Flickering Consistency | Measures the consistency of visual elements across frames to detect flickering artifacts. Higher scores reflect better temporal stability. | (Huang et al., 2024) |
| Human Action Recognition Accuracy | Assesses the accuracy of human actions depicted in videos using pre-trained recognition models. Higher scores signify better action representation. | (Huang et al., 2024) |
| Dynamic Degree | Quantifies the level of dynamic content in videos, evaluating the extent of motion present. Higher scores indicate more dynamic scenes. | (Huang et al., 2024) |
| *Common Video Metrics* | | |
| Fréchet Video Distance (FVD) | Measures the distributional distance between real and generated videos using features from a pre-trained network. Lower scores indicate better quality. | (Unterthiner et al., 2019) |
| Learned Perceptual Image Patch Similarity (LPIPS) | Evaluates perceptual similarity between images using deep network features. Lower scores denote higher similarity. | (Zhang et al., 2018) |
| Structural Similarity Index Measure (SSIM) | Assesses image similarity based on luminance, contrast, and structure. Higher scores reflect greater similarity. | (Wang et al., 2004) |
| Peak Signal-to-Noise Ratio (PSNR) | Calculates the ratio between the maximum possible power of a signal and the power of corrupting noise. Higher scores indicate better quality. | (Huynh-Thu & Ghanbari, 2008) |

Table 8: Summary of video datasets used in experiments.

| Dataset | # Videos | Duration | Resolution | Splits (Train/Val/Test) | Ref. |
|---|---|---|---|---|---|
| **WebVid-2M** | 2,617,758 | 10–30s | up to 4K | 2,612,518 / 5,240 / — | (Bain et al., 2021) |
| **MSR-VTT** | 10,000 | 15s | 320×240 | 6,513 / 497 / 2,990 | (Xu et al., 2016) |
| **UCF101** | 13,320 | 7–10s | 320×240 | 9,537 / — / 3,783 | (Soomro et al., 2012) |
| **MSVD** | 1,970 | 1–60s | 480×360 | 1,200 / 100 / 670 | (Chen & Dolan, 2011) |

**Dataset Notes.**

- **WebVid-2M:** A large-scale dataset comprising over 2.6 million videos with weak captions scraped from the web. Resolutions vary, with some videos up to 4K; durations range from 10 to 30 seconds.
- **MSR-VTT:** Contains 10,000 video clips ( 15s each) at 320×240 resolution, each paired with 20 captions. Standard split: 6,513 train / 497 val / 2,990 test.
- **UCF101:** Action recognition dataset with 13,320 videos across 101 classes. Durations range 7–10 seconds, resolution is 320×240. Commonly split as 9,537 train / 3,783 test.
- **MSVD:** 1,970 YouTube videos (1–60s) at 480×360 resolution. Each video has multiple captions. Standard split: 1,200 train / 100 val / 670 test.

Table 9: Zero-Shot Cross-Dataset Evaluation Protocols for T2V generation (Wang et al., 2023a).

| Dataset | Evaluation Protocol |
|---|---|
| **UCF101** | Generate 100 videos per class using class labels as prompts. |
| **MSVD** | Generate one video per sample in the full test split, using a reproducibly sampled caption as the prompt for each video. |
| **MSR-VTT** | Generate 2,048 videos sampled from the test set, each with a reproducibly sampled caption used as the prompt. |
| **WebVid-2M** | Generate videos using the validation set, where each video is conditioned on its paired caption. |

Table 10: Full quantitative results across FVD (Unterthiner et al., 2019), VBench (Huang et al., 2024), and EvalCrafter (Liu et al., 2024b) metrics.

| Corruption | FVD ↓ | VBench | | | | EvalCrafter | | | | | | | |
|---|---|---|---|---|---|---|---|---|---|---|---|---|---|
| | | Smooth | Flicker | Human | Dynamic | IS | Clip Temp | VQA_A | VQA_T | Action | Clip | Flow | Motion |
| BCNI | **360.32** | **0.9612** | **0.9681** | **0.8920** | **0.8281** | **15.28** | 99.63 | **20.12** | 12.27 | **65.09** | 19.58 | **6.45** | 60.0 |
| Gaussian | 400.29 | 0.5748 | 0.9536 | 0.8340 | 0.6685 | 14.57 | **99.68** | 17.07 | **14.28** | 60.21 | **19.73** | 4.75 | **62.0** |
| Uniform | 443.22 | 0.5718 | 0.9367 | 0.8500 | 0.7695 | 14.22 | 99.65 | 17.45 | 13.35 | 64.71 | 19.58 | 6.38 | **62.0** |
| Clean | 520.32 | 0.5686 | 0.9476 | 0.8260 | 0.7290 | 14.65 | 99.66 | 14.81 | 12.10 | 63.90 | 19.66 | 4.96 | 60.0 |

**Metrics.** FVD: Fréchet Video Distance; IS: Inception Score; Clip Temp: CLIP Temporal Consistency; VQA_A: Video Quality Assessment – Aesthetic Score; VQA_T: Video Quality Assessment – Technical Score; Action: Action Recognition Score; Clip: CLIP Similarity Score; Flow: Flow Score / Optical Flow Consistency Score; Motion: Motion Amplitude Classification Score; Smooth: Motion Smoothness; Flicker: Temporal Flickering Consistency; Human: Human Action Recognition Accuracy; Dynamic: Dynamic Degree. Metric definitions are provided in Table 7

# E    FURTHER RESULTS

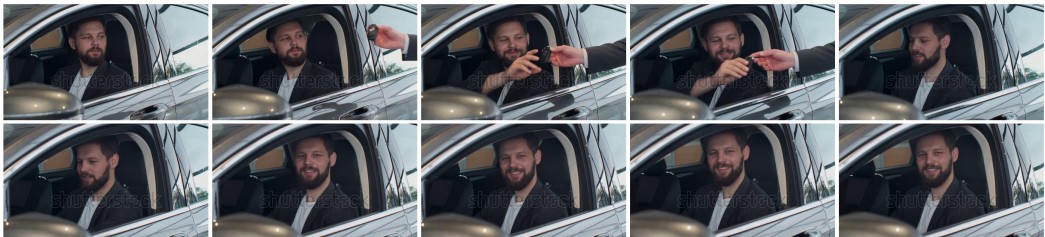

**A sales manager hands over car keys to a seated customer.**

Figure 3: **Visual Representation of Video Captions.** The extracted frames depict the scene described by the original captions before corruption. The video illustrates a sales manager handing over car keys to a man seated in the driver's seat. This serves as a reference to understand how different noise levels impact text descriptions of the same visual content. Text corruption effects are depicted in Table 11.

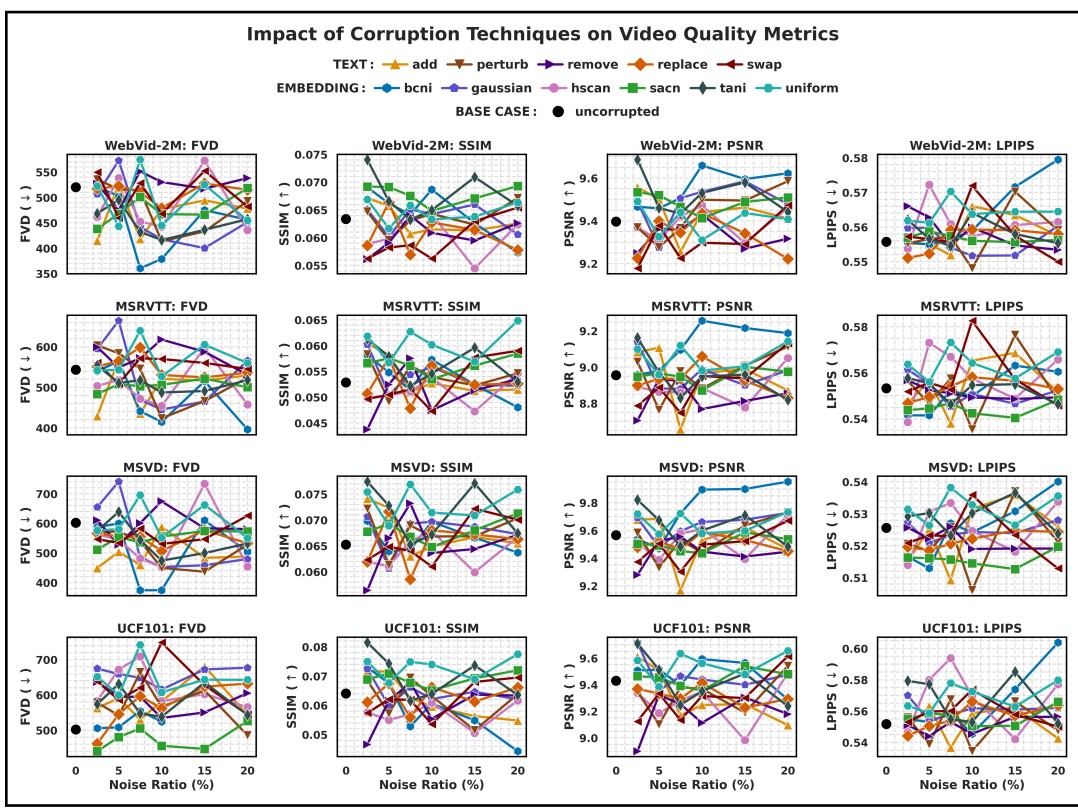

Figure 4: **Ablation Corruption Experiments**. Impact of corruption techniques on video quality metrics (FVD, SSIM, PSNR, LPIPS) across four standard text-to-video datasets: WebVid-2M, MSRVTT, MSVD, and UCF101. Text-level and embedding-level corruptions are evaluated at varying noise ratios.

Table 11: Effect of Corruption Techniques on Captions by Noise Ratio (%).

| Noise Ratio | Caption |
| --- | --- |
| **Noise Ratio = 2.5%** | |
| *Clean* | Sales manager handing over the keys to man that sitting in the car. |
| *Swap* | Sales the handing over manager keys to man that sitting in the car. |
| *Add* | Sales manager handing over the keys to man that sitting in the owkip car. |
| *Replace* | Sales manager handing over atoii keys to man that sitting in the car. |
| *Perturb* | Sales manager handing over the keys to man that max in the car. |
| *Remove* | Sales manager handing over the keys to man that sitting in car. |
| **Noise Ratio = 5%** | |
| *Clean* | Sales manager handing over the keys to man that sitting in the car. |
| *Swap* | Sales manager handing to the keys over man that sitting in the car. |
| *Add* | Sales manager handing over the keys to fjogc man that sitting in the car. |
| *Replace* | Sales manager handing bkwlj the keys to man that sitting in the car. |
| *Perturb* | Sales manager handing over the keys to man that jn in the car. |
| *Remove* | Sales handing over the keys to man that sitting in the car. |
| **Noise Ratio = 7.5%** | |
| *Clean* | Sales manager handing over the keys to man that sitting in the car. |
| *Swap* | Sales manager handing keys the over to man that sitting in the car. |
| *Add* | nuabx Sales manager handing over the keys to man that sitting in the car. |
| *Replace* | Sales manager handing over the keys to man that sitting in viukq car. |
| *Perturb* | Sales manager handing over the keys to man that sitting in the un. |
| *Remove* | Sales manager handing over the keys man that sitting in the car. |
| **Noise Ratio = 10%** | |
| *Clean* | Sales manager handing over the keys to man that sitting in the car. |
| *Swap* | Sales manager handing over the keys to man that sitting in the car. |
| *Add* | Sales manager handing over the keys nfgco to man that sitting in the car. |
| *Replace* | oybix manager handing over the keys to man that sitting in the car. |
| *Perturb* | Sales manager handing over the keys to man that mkn in the car. |
| *Remove* | Manager handing over the keys to man that sitting in the car. |
| **Noise Ratio = 15%** | |
| *Clean* | Sales manager handing over the keys to man that sitting in the car. |
| *Swap* | Sales manager handing over the keys to in that sitting man the car. |
| *Add* | Sales manager handing over fuibu the keys to man that sitting in the car. |
| *Replace* | Sales manager handing zsmko the keys to man that sitting in the car. |
| *Perturb* | Sales manager handing over the keys to kbys that sitting in the car. |
| *Remove* | Sales manager handing over the keys to man that sitting in the. |
| **Noise Ratio = 20%** | |
| *Clean* | Sales manager handing over the keys to man that sitting in the car. |
| *Swap* | The manager handing over the keys sitting man that to in Sales car. |
| *Add* | Sales manager handing over the keys svlkq to man that cijet sitting in the car. |
| *Replace* | Sales manager handing over nibke irico to man that sitting in the car. |
| *Perturb* | Sales manager sittdng over the keys to man keqs sitting in the car. |
| *Remove* | Manager handing over the to man that sitting in the car. |

Table 12: Zero-shot video generation results with a **diffusion** backbone on WebVid-2M (val), MSR-VTT, MSVD, and UCF101, evaluated using FVD. FVMD/CMMD are in Table 13
Legend: 1st , 2nd , 3rd , 4th , 5th .

| Method | WebVid-2M | | | | | | | MSR-VTT | | | | | | |
| | 0% | 2.5% | 5% | 7.5% | 10% | 15% | 20% | 0% | 2.5% | 5% | 7.5% | 10% | 15% | 20% |
|---|---|---|---|---|---|---|---|---|---|---|---|---|---|---|
| Add | 520.32 | 414.83 | 525.53 | 418.24 | 476.76 | 494.30 | 478.16 | 543.33 | 429.05 | 567.65 | 435.25 | 520.89 | 516.80 | 529.33 |
| BCNI | 520.32 | 521.24 | 502.45 | 360.32 | 378.87 | 475.01 | 456.14 | 543.33 | 539.93 | 564.00 | 441.31 | 414.49 | 515.12 | 396.35 |
| GAP | 520.32 | 525.52 | 593.64 | 520.54 | 424.69 | 512.10 | 573.37 | 543.33 | 613.22 | 710.24 | 620.46 | 575.66 | 857.56 | 999.89 |
| Gaussian | 520.32 | 506.56 | 572.67 | 441.69 | 417.60 | 400.29 | 451.67 | 543.33 | 595.08 | 664.45 | 468.79 | 445.29 | 464.91 | 565.83 |
| HSCAN | 520.32 | 461.39 | 538.34 | 452.74 | 442.29 | 572.78 | 435.63 | 543.33 | 503.60 | 517.88 | 471.87 | 454.01 | 597.32 | 457.68 |
| Perturb | 520.32 | 535.48 | 516.19 | 514.50 | 413.61 | 433.20 | 492.83 | 543.33 | 603.43 | 584.07 | 545.70 | 423.51 | 466.08 | 523.68 |
| Remove | 520.32 | 527.24 | 475.15 | 551.35 | 530.37 | 517.51 | 538.09 | 543.33 | 599.39 | 548.84 | 572.04 | 617.93 | 587.65 | 530.84 |
| Replace | 520.32 | 517.43 | 522.32 | 510.42 | 479.21 | 527.73 | 514.80 | 543.33 | 551.22 | 564.57 | 598.50 | 530.50 | 524.62 | 536.22 |
| SACN | 520.32 | 438.19 | 467.93 | 500.92 | 467.14 | 466.18 | 518.43 | 543.33 | 440.28 | 507.88 | 502.69 | 506.20 | 446.78 | 500.23 |
| Swap | 520.32 | 549.72 | 459.72 | 528.22 | 467.92 | 552.33 | 481.93 | 543.33 | 560.49 | 508.46 | 571.54 | 569.84 | 560.17 | 545.16 |
| TANI | 520.32 | 468.31 | 495.34 | 432.20 | 416.11 | 436.27 | 457.33 | 543.33 | 553.77 | 510.79 | 517.47 | 487.28 | 490.87 | 517.75 |
| Uniform | 520.32 | 522.36 | 443.22 | 574.35 | 444.71 | 525.22 | 454.79 | 543.33 | 541.80 | 543.46 | 639.83 | 526.85 | 605.27 | 559.93 |

| Method | MSVD | | | | | | | UCF-101 | | | | | | |
| | 0% | 2.5% | 5% | 7.5% | 10% | 15% | 20% | 0% | 2.5% | 5% | 7.5% | 10% | 15% | 20% |
|---|---|---|---|---|---|---|---|---|---|---|---|---|---|---|
| Add | 602.39 | 449.50 | 503.85 | 459.82 | 588.26 | 483.82 | 488.12 | 501.91 | 562.83 | 590.60 | 530.54 | 581.48 | 681.15 | 542.83 |
| BCNI | 602.39 | 587.59 | 599.44 | 374.34 | 374.52 | 610.38 | 504.35 | 501.91 | 505.54 | 508.13 | 554.73 | 523.93 | 926.35 | 921.69 |
| GAP | 602.39 | 665.77 | 829.36 | 704.69 | 662.16 | 1112.23 | 1357.74 | 501.91 | 498.14 | 612.76 | 1023.15 | 1246.66 | 1460.14 | 1717.69 |
| Gaussian | 602.39 | 654.73 | 740.79 | 485.30 | 452.82 | 458.69 | 479.63 | 501.91 | 674.62 | 659.27 | 648.41 | 615.28 | 672.25 | 677.13 |
| HSCAN | 602.39 | 562.76 | 545.31 | 481.12 | 452.57 | 733.60 | 454.03 | 501.91 | 583.43 | 671.78 | 708.75 | 582.24 | 446.78 | 565.22 |
| Perturb | 602.39 | 568.32 | 573.24 | 540.22 | 449.78 | 436.90 | 528.26 | 501.91 | 578.80 | 541.24 | 664.29 | 566.35 | 612.56 | 485.16 |
| Remove | 602.39 | 610.61 | 528.18 | 600.97 | 674.23 | 583.37 | 580.83 | 501.91 | 636.93 | 594.68 | 594.66 | 534.72 | 550.24 | 605.32 |
| Replace | 602.39 | 566.27 | 555.65 | 554.65 | 507.17 | 576.98 | 532.82 | 501.91 | 461.58 | 545.23 | 596.56 | 561.97 | 623.54 | 632.04 |
| SACN | 602.39 | 511.24 | 554.20 | 535.55 | 555.61 | 574.29 | 572.48 | 501.91 | 440.28 | 480.29 | 504.89 | 455.65 | 446.78 | 526.23 |
| Swap | 602.39 | 547.05 | 533.62 | 582.41 | 530.68 | 546.70 | 625.97 | 501.91 | 638.17 | 585.97 | 619.91 | 748.43 | 627.28 | 544.03 |
| TANI | 602.39 | 574.24 | 639.00 | 544.01 | 474.27 | 499.91 | 532.19 | 501.91 | 573.51 | 631.22 | 547.25 | 538.13 | 635.20 | 543.12 |
| Uniform | 602.39 | 575.76 | 580.59 | 695.59 | 551.99 | 662.51 | 550.73 | 501.91 | 651.64 | 599.53 | 742.18 | 607.23 | 643.22 | 642.74 |

Table 13: Zero-shot video generation results with a **diffusion** backbone on WebVid-2M (val), MSR-VTT, MSVD, and UCF101, evaluated using FVMD and CMMD. (FVD Table 12)

Legend: 1st , 2nd , 3rd , 4th , 5th .

*(a) FVMD (↓)*

| Method | WebVid-2M | | | | | | | MSR-VTT | | | | | | |
|---|---|---|---|---|---|---|---|---|---|---|---|---|---|---|
| | 0% | 2.5% | 5% | 7.5% | 10% | 15% | 20% | 0% | 2.5% | 5% | 7.5% | 10% | 15% | 20% |
| Add | 7119.88 | 4554.06 | 4708.06 | 6462.25 | 6862.60 | 5250.99 | 5415.74 | 9296.45 | 8422.97 | 7382.50 | 8716.41 | 8264.62 | 8345.86 | 8865.10 |
| BCNI | 7119.88 | 5171.73 | 4277.48 | 2930.58 | 2642.15 | 6339.48 | 3578.96 | 9296.45 | 8963.76 | 8188.26 | 5709.43 | 6853.90 | 11755.95 | 5956.47 |
| GAP | 7119.88 | 4519.42 | 4715.57 | 10239.91 | 12630.97 | 15288.51 | 11246.24 | 9296.45 | 8235.40 | 12550.56 | 23882.52 | 21371.65 | 20259.74 | 8035.42 |
| Gaussian | 7119.88 | 6253.58 | 3532.48 | 4240.57 | 3458.60 | 3916.42 | 3524.58 | 9296.45 | 12805.36 | 7530.45 | 8075.93 | 5224.22 | 7263.80 | 6512.54 |
| HSCAN | 7119.88 | 4298.57 | 3790.34 | 3091.29 | 2813.96 | 5358.07 | 2920.34 | 9296.45 | 7880.24 | 5678.14 | 7639.59 | 6984.37 | 10815.09 | 7199.65 |
| Perturb | 7119.88 | 5389.99 | 6515.52 | 4120.26 | 4579.37 | 4080.31 | 5463.62 | 9296.45 | 7614.27 | 8636.10 | 7487.41 | 8425.08 | 7075.52 | 10060.26 |
| Remove | 7119.88 | 3880.19 | 8046.34 | 6074.67 | 6864.34 | 6730.27 | 4422.09 | 9296.45 | 7646.77 | 11801.99 | 10174.45 | 9212.60 | 8238.40 | 7205.45 |
| Replace | 7119.88 | 4492.70 | 4039.02 | 4844.42 | 5294.52 | 5188.01 | 5571.67 | 9296.45 | 7194.60 | 6619.72 | 8259.47 | 7604.41 | 7971.50 | 9842.79 |
| SACN | 7119.88 | 3071.20 | 4044.40 | 5371.88 | 3013.97 | 3013.97 | 4268.20 | 9296.45 | 6409.46 | 7623.23 | 8032.28 | 5754.56 | 5705.31 | 7389.99 |
| Swap | 7119.88 | 7167.32 | 5224.20 | 4548.15 | 6694.89 | 6715.66 | 4309.75 | 9296.45 | 9720.78 | 8447.80 | 7827.10 | 7900.02 | 9680.74 | 6837.29 |
| TANI | 7119.88 | 5019.23 | 3387.44 | 3392.34 | 5566.76 | 6511.16 | 4518.37 | 9296.45 | 9454.21 | 7169.30 | 6611.25 | 8909.69 | 11133.71 | 7906.84 |
| Uniform | 7119.88 | 6514.07 | 3263.99 | 4322.78 | 2832.32 | 5499.35 | 2803.55 | 9296.45 | 11826.79 | 5396.16 | 8122.32 | 6727.26 | 11868.59 | 6334.83 |

| Method | MSVD | | | | | | | UCF-101 | | | | | | |
|---|---|---|---|---|---|---|---|---|---|---|---|---|---|---|
| | 0% | 2.5% | 5% | 7.5% | 10% | 15% | 20% | 0% | 2.5% | 5% | 7.5% | 10% | 15% | 20% |
| Add | 6112.60 | 5704.31 | 6815.98 | 5792.23 | 8845.01 | 5667.24 | 6724.64 | 4254.70 | 5815.48 | 6554.78 | 4182.14 | 6581.79 | 9804.16 | 6345.49 |
| BCNI | 6112.60 | 3787.22 | 3700.55 | 5420.46 | 4818.96 | 6766.14 | 6406.56 | 4254.70 | 5826.69 | 5215.35 | 5247.57 | 4130.11 | 6345.10 | 9278.69 |
| GAP | 6112.60 | 4066.78 | 4935.46 | 9498.63 | 11503.44 | 15229.36 | 10595.61 | 4254.70 | 6667.66 | 8543.87 | 16087.88 | 12182.65 | 6415.62 | 9264.86 |
| Gaussian | 6112.60 | 7137.37 | 5579.96 | 5551.08 | 5100.71 | 5488.52 | 5192.65 | 4254.70 | 8055.59 | 4379.23 | 4270.27 | 5365.68 | 5443.15 | 3370.32 |
| HSCAN | 6112.60 | 4814.12 | 4042.47 | 4813.74 | 4764.39 | 5178.11 | 3853.20 | 4254.70 | 9241.11 | 4414.51 | 7001.98 | 6988.44 | 5875.14 | 5879.88 |
| Perturb | 6112.60 | 7013.32 | 7121.45 | 7126.01 | 4765.87 | 6047.34 | 6527.22 | 4254.70 | 5082.29 | 6731.57 | 7073.45 | 5171.44 | 5264.66 | 5957.73 |
| Remove | 6112.60 | 6234.95 | 7795.59 | 6794.81 | 7805.36 | 8168.63 | 7407.37 | 4254.70 | 6650.62 | 6951.19 | 6354.22 | 5705.31 | 7077.52 | 6084.74 |
| Replace | 6112.60 | 5145.45 | 6530.55 | 7274.75 | 7498.75 | 7693.98 | 6852.13 | 4254.70 | 4224.68 | 5810.55 | 6430.62 | 5709.42 | 6648.93 | 6663.29 |
| SACN | 6112.60 | 4121.10 | 3542.42 | 3475.19 | 3953.00 | 4056.64 | 3599.20 | 4254.70 | 3497.47 | 3928.77 | 4071.79 | 3384.40 | 3434.59 | 4908.81 |
| Swap | 6112.60 | 7388.58 | 6953.22 | 6386.29 | 8285.71 | 7162.17 | 6339.92 | 4254.70 | 6674.92 | 5237.32 | 7565.82 | 7565.82 | 6117.41 | 3791.82 |
| TANI | 6112.60 | 5714.78 | 5433.25 | 4556.20 | 5755.43 | 6250.03 | 5638.06 | 4254.70 | 5066.60 | 6059.36 | 5437.29 | 5934.73 | 8188.66 | 4786.29 |
| Uniform | 6112.60 | 6540.08 | 4925.55 | 5838.66 | 5362.51 | 5066.98 | 4471.67 | 4254.70 | 5536.85 | 3325.36 | 3639.71 | 4156.79 | 12700.47 | 5365.23 |

*(b) CMMD (↓)*

| Method | WebVid-2M | | | | | | | MSR-VTT | | | | | | |
|---|---|---|---|---|---|---|---|---|---|---|---|---|---|---|
| | 0% | 2.5% | 5% | 7.5% | 10% | 15% | 20% | 0% | 2.5% | 5% | 7.5% | 10% | 15% | 20% |
| Add | 0.534 | 0.535 | 0.585 | 0.576 | 0.580 | 0.685 | 0.616 | 0.831 | 0.914 | 0.955 | 0.901 | 0.893 | 1.035 | 0.929 |
| BCNI | 0.534 | 0.652 | 0.659 | 0.601 | 0.618 | 0.798 | 1.025 | 0.831 | 0.975 | 0.998 | 0.938 | 1.009 | 1.198 | 1.471 |
| GAP | 0.534 | 0.628 | 0.715 | 0.953 | 1.258 | 1.708 | 2.078 | 0.831 | 0.943 | 1.004 | 1.386 | 1.757 | 2.260 | 2.590 |
| Gaussian | 0.534 | 0.571 | 0.561 | 0.569 | 0.585 | 0.575 | 0.599 | 0.831 | 0.911 | 0.825 | 0.835 | 0.840 | 0.823 | 0.838 |
| HSCAN | 0.534 | 0.564 | 0.620 | 0.615 | 0.606 | 0.672 | 0.621 | 0.831 | 0.806 | 0.906 | 0.923 | 0.933 | 0.971 | 0.967 |
| Perturb | 0.534 | 0.568 | 0.609 | 0.582 | 0.573 | 0.632 | 0.597 | 0.831 | 0.863 | 0.865 | 0.884 | 0.965 | 0.905 | 0.911 |
| Remove | 0.534 | 0.671 | 0.633 | 0.620 | 0.614 | 0.576 | 0.574 | 0.831 | 0.923 | 0.980 | 1.002 | 0.934 | 0.814 | 0.892 |
| Replace | 0.534 | 0.567 | 0.582 | 0.628 | 0.581 | 0.588 | 0.600 | 0.831 | 0.843 | 0.882 | 0.908 | 0.913 | 0.874 | 0.913 |
| SACN | 0.534 | 0.583 | 0.613 | 0.631 | 0.592 | 0.583 | 0.618 | 0.831 | 0.913 | 1.001 | 0.999 | 0.921 | 0.905 | 0.978 |
| Swap | 0.534 | 0.606 | 0.623 | 0.599 | 0.651 | 0.585 | 0.534 | 0.831 | 0.863 | 0.955 | 0.885 | 0.939 | 0.923 | 0.858 |
| TANI | 0.534 | 0.619 | 0.543 | 0.588 | 0.609 | 0.598 | 0.593 | 0.831 | 0.930 | 0.825 | 0.866 | 0.939 | 0.955 | 0.890 |
| Uniform | 0.534 | 0.563 | 0.495 | 0.552 | 0.513 | 0.589 | 0.531 | 0.831 | 0.967 | 0.810 | 0.934 | 0.876 | 0.937 | 0.928 |

| Method | MSVD | | | | | | | UCF-101 | | | | | | |
|---|---|---|---|---|---|---|---|---|---|---|---|---|---|---|
| | 0% | 2.5% | 5% | 7.5% | 10% | 15% | 20% | 0% | 2.5% | 5% | 7.5% | 10% | 15% | 20% |
| Add | 0.814 | 0.953 | 0.919 | 0.957 | 0.842 | 0.947 | 0.860 | 1.189 | 1.340 | 1.455 | 1.282 | 1.403 | 1.703 | 1.463 |
| BCNI | 0.814 | 1.020 | 0.970 | 0.901 | 0.936 | 1.059 | 1.270 | 1.189 | 1.344 | 1.371 | 1.436 | 1.513 | 1.983 | 2.464 |
| GAP | 0.814 | 0.942 | 0.912 | 1.207 | 1.686 | 2.267 | 2.532 | 1.189 | 1.329 | 1.524 | 2.180 | 2.564 | 2.937 | 3.281 |
| Gaussian | 0.814 | 0.898 | 0.865 | 0.894 | 0.897 | 0.855 | 0.869 | 1.189 | 1.426 | 1.209 | 1.222 | 1.217 | 1.276 | 1.207 |
| HSCAN | 0.814 | 0.869 | 0.943 | 0.936 | 0.948 | 0.987 | 0.942 | 1.189 | 1.334 | 1.432 | 1.401 | 1.348 | 1.435 | 1.415 |
| Perturb | 0.814 | 0.871 | 0.888 | 0.880 | 0.985 | 0.862 | 0.893 | 1.189 | 1.323 | 1.326 | 1.456 | 1.338 | 1.520 | 1.429 |
| Remove | 0.814 | 0.881 | 0.972 | 0.985 | 0.938 | 0.848 | 0.917 | 1.189 | 1.562 | 1.521 | 1.547 | 1.424 | 1.239 | 1.436 |
| Replace | 0.814 | 0.850 | 0.929 | 0.843 | 0.871 | 0.847 | 0.896 | 1.189 | 1.235 | 1.322 | 1.469 | 1.427 | 1.388 | 1.431 |
| SACN | 0.814 | 0.972 | 1.056 | 1.014 | 0.980 | 0.951 | 0.996 | 1.189 | 1.231 | 1.312 | 1.318 | 1.211 | 1.237 | 1.312 |
| Swap | 0.814 | 0.877 | 0.933 | 0.891 | 0.817 | 0.919 | 0.863 | 1.189 | 1.378 | 1.484 | 1.400 | 1.451 | 1.442 | 1.200 |
| TANI | 0.814 | 0.978 | 0.896 | 0.918 | 0.950 | 1.014 | 0.955 | 1.189 | 1.403 | 1.280 | 1.271 | 1.396 | 1.394 | 1.259 |
| Uniform | 0.814 | 1.003 | 0.826 | 0.967 | 0.883 | 0.947 | 0.905 | 1.189 | 1.399 | 1.153 | 1.383 | 1.234 | 1.480 | 1.270 |

Table 14: **Model-Dataset Evaluations (Autoregressive).** FVD comparisons across noise ratios.

| Noise ratio (%) | WebVid-2M | | | | MSRVTT | | | | MSVD | | | | UCF101 | | | |
|---|---|---|---|---|---|---|---|---|---|---|---|---|---|---|---|---|
| | BCNI | SACN | Gaussian | Uniform | BCNI | SACN | Gaussian | Uniform | BCNI | SACN | Gaussian | Uniform | BCNI | SACN | Gaussian | Uniform |
| 2.5 | 327.81 | 292.14 | 363.54 | 368.80 | 396.11 | 361.02 | 391.27 | 412.02 | 565.37 | 573.05 | 656.65 | 543.96 | 1037.14 | 862.88 | 1048.83 | 1018.44 |
| 5 | 275.19 | 322.14 | 293.01 | 253.06 | 400.49 | 369.94 | 427.37 | 389.57 | 531.34 | 570.54 | 632.93 | 561.80 | 881.91 | 957.71 | 952.19 | 1281.83 |
| 7.5 | 362.83 | 257.88 | 308.67 | 375.15 | 433.68 | 393.27 | 290.98 | 420.13 | 577.22 | 578.79 | 501.88 | 584.92 | 917.38 | 1064.79 | 847.36 | 1186.03 |
| 10 | 278.55 | 320.32 | 247.33 | 223.98 | 358.25 | 380.07 | 353.43 | 348.07 | 533.46 | 602.07 | 567.17 | 494.31 | 996.46 | 1058.34 | 1116.46 | 936.53 |
| 15 | 257.94 | 344.58 | 200.24 | 293.23 | 367.25 | 403.95 | 280.93 | 425.16 | 515.19 | 590.35 | 385.81 | 525.57 | 1116.84 | 1123.45 | 760.64 | 1050.79 |
| 20 | 293.76 | 294.70 | 242.54 | 335.24 | 410.50 | 409.28 | 309.85 | 356.70 | 543.66 | 602.15 | 468.73 | 551.68 | 1003.75 | 1123.45 | 1115.95 | 929.29 |
| Clean | | 315.16 | | | | 431.79 | | | | 629.91 | | | | 1309.04 | | |

Table 15: Zero-shot video generation results with an **autoregressive** backbone on WebVid-2M (val), MSR-VTT, MSVD, and UCF101, evaluated using FVD. FVMD/CMMD are in Table 16.

Legend: 1st , 2nd , 3rd , 4th , 5th .

| Method | 0% | 2.5% | 5% | 7.5% | 10% | 15% | 20% | 0% | 2.5% | 5% | 7.5% | 10% | 15% | 20% |
|---|---|---|---|---|---|---|---|---|---|---|---|---|---|---|
| | WebVid-2M | | | | | | | MSR-VTT | | | | | | |
| BCNI | 315.16 | 327.81 | 275.19 | 362.83 | 278.55 | 257.94 | 293.76 | 431.79 | 396.11 | 400.49 | 433.68 | 358.25 | 367.25 | 410.50 |
| Gaussian | 315.16 | 363.54 | 293.01 | 308.67 | 247.33 | 200.24 | 242.54 | 431.79 | 391.27 | 427.37 | 290.98 | 353.43 | 280.93 | 309.85 |
| HSCAN | 315.16 | 280.57 | 235.67 | 309.19 | 255.31 | 276.77 | 270.45 | 431.79 | 401.81 | 323.32 | 380.49 | 302.86 | 410.67 | 289.70 |
| SACN | 315.16 | 292.14 | 322.14 | 257.88 | 320.32 | 344.58 | 294.70 | 431.79 | 361.02 | 369.94 | 393.27 | 380.07 | 403.95 | 409.28 |
| Uniform | 315.16 | 368.80 | 253.06 | 375.15 | 223.98 | 293.23 | 335.24 | 431.79 | 412.02 | 389.57 | 420.13 | 348.07 | 425.16 | 356.70 |
| TANI | 315.16 | 274.95 | 321.23 | 440.86 | 333.94 | 269.16 | 328.18 | 431.79 | 291.68 | 320.70 | 727.46 | 400.74 | 306.55 | 547.16 |
| | MSVD | | | | | | | UCF-101 | | | | | | |
| BCNI | 629.91 | 565.37 | 531.34 | 577.22 | 533.46 | 515.19 | 543.66 | 1309.04 | 1037.14 | 881.91 | 917.38 | 996.46 | 1116.84 | 1003.75 |
| Gaussian | 629.91 | 656.65 | 632.93 | 501.88 | 567.17 | 385.81 | 468.73 | 1309.04 | 1048.83 | 952.19 | 847.36 | 1116.46 | 760.64 | 1115.95 |
| HSCAN | 629.91 | 592.00 | 490.74 | 550.16 | 500.58 | 396.80 | 527.12 | 1309.04 | 1049.31 | 1012.38 | 992.21 | 877.72 | 757.90 | 1029.77 |
| SACN | 629.91 | 573.05 | 570.54 | 578.79 | 602.07 | 590.35 | 602.15 | 1309.04 | 862.88 | 957.71 | 1064.79 | 1058.34 | 883.25 | 1123.45 |
| Uniform | 629.91 | 543.96 | 561.80 | 584.92 | 494.31 | 525.57 | 551.68 | 1309.04 | 1018.44 | 1281.83 | 1186.03 | 936.53 | 1050.79 | 929.29 |
| TANI | 629.91 | 493.75 | 483.31 | 738.09 | 564.89 | 477.53 | 745.06 | 1309.04 | 831.43 | 937.59 | 1363.07 | 1150.66 | 976.03 | 1077.35 |

Table 16: Zero-shot video generation results with an **autoregressive** backbone on WebVid-2M (val), MSR-VTT, MSVD, and UCF101, evaluated using FVMD and CMMD. FVD (Table 15)

Legend: 1st , 2nd , 3rd , 4th , 5th .

*(a) FVMD (↓)*

| Method | 0% | 2.5% | 5% | 7.5% | 10% | 15% | 20% | 0% | 2.5% | 5% | 7.5% | 10% | 15% | 20% |
|---|---|---|---|---|---|---|---|---|---|---|---|---|---|---|
| | WebVid-2M | | | | | | | MSR-VTT | | | | | | |
| BCNI | 7529.07 | 7531.96 | 8990.52 | 11585.45 | 11075.10 | 10698.51 | 9857.85 | 8851.37 | 7232.94 | 8887.03 | 9241.40 | 9495.64 | 10667.96 | 9100.20 |
| Gaussian | 7529.07 | 11515.04 | 8782.35 | 10521.78 | 9579.66 | 11147.15 | 8572.26 | 8851.37 | 10871.33 | 8290.57 | 9161.28 | 8265.19 | 9687.71 | 8606.21 |
| HSCAN | 7529.07 | 6905.52 | 9505.46 | 8328.17 | 7296.80 | 10169.49 | 6788.56 | 8851.37 | 8164.07 | 9180.58 | 8925.60 | 9296.64 | 8080.37 | 7760.92 |
| SACN | 7529.07 | 10064.75 | 13494.61 | 11620.58 | 8553.72 | 11129.95 | 11800.13 | 8851.37 | 9920.05 | 11478.01 | 10516.12 | 8590.54 | 9794.76 | 8779.26 |
| Uniform | 7529.07 | 11398.63 | 11125.29 | 9002.75 | 10515.16 | 10299.02 | 9129.10 | 8851.37 | 9439.82 | 10291.41 | 9155.02 | 10175.21 | 8694.50 | 7548.65 |
| TANI | 7529.07 | 7732.30 | 9739.71 | 8194.09 | 11086.04 | 8780.55 | 7940.12 | 8851.37 | 7830.28 | 7283.66 | 8458.28 | 10300.80 | 7898.44 | 7897.32 |
| | MSVD | | | | | | | UCF-101 | | | | | | |
| BCNI | 11800.27 | 10502.44 | 11831.83 | 14540.26 | 13525.42 | 15224.07 | 13389.00 | 13548.23 | 10649.17 | 12097.68 | 14426.77 | 14295.73 | 14288.39 | 14207.08 |
| Gaussian | 11800.27 | 15553.98 | 12230.33 | 14202.35 | 13032.53 | 14911.84 | 12585.22 | 13548.23 | 15049.47 | 12185.18 | 13222.32 | 12484.33 | 13791.56 | 13406.85 |
| HSCAN | 11800.27 | 11462.00 | 12623.01 | 12921.38 | 12176.66 | 11980.06 | 10923.45 | 13548.23 | 10194.11 | 12913.33 | 12222.82 | 11310.21 | 11771.52 | 11969.79 |
| SACN | 11800.27 | 14389.67 | 16150.93 | 16143.49 | 12333.49 | 14847.25 | 13545.14 | 13548.23 | 13554.59 | 15212.87 | 14119.03 | 13082.39 | 14204.25 | 13718.58 |
| Uniform | 11800.27 | 14731.30 | 15053.17 | 12835.55 | 13594.96 | 12305.03 | 11313.64 | 13548.23 | 14745.46 | 13337.44 | 13538.99 | 12141.23 | 13706.67 | 12712.40 |
| TANI | 11800.27 | 11510.36 | 11388.21 | 11368.97 | 15558.99 | 10535.25 | 12007.48 | 13548.23 | 9901.83 | 12631.76 | 11425.83 | 14335.79 | 12783.32 | 10736.95 |

*(b) CMMD (↓)*

| Method | 0% | 2.5% | 5% | 7.5% | 10% | 15% | 20% | 0% | 2.5% | 5% | 7.5% | 10% | 15% | 20% |
|---|---|---|---|---|---|---|---|---|---|---|---|---|---|---|
| | WebVid-2M | | | | | | | MSR-VTT | | | | | | |
| BCNI | 0.734 | 0.679 | 0.595 | 0.529 | 0.573 | 0.569 | 0.561 | 1.813 | 1.807 | 1.720 | 1.667 | 1.555 | 1.716 | 1.628 |
| Gaussian | 0.734 | 0.656 | 0.607 | 0.709 | 0.480 | 0.427 | 0.536 | 1.813 | 1.588 | 1.756 | 1.678 | 1.619 | 1.417 | 1.698 |
| HSCAN | 0.734 | 0.629 | 0.581 | 0.616 | 0.624 | 0.584 | 0.574 | 1.813 | 1.715 | 1.665 | 1.718 | 1.755 | 1.668 | 1.675 |
| SACN | 0.734 | 0.676 | 0.626 | 0.587 | 0.649 | 0.532 | 0.484 | 1.813 | 1.740 | 1.738 | 1.567 | 1.706 | 1.580 | 1.564 |
| Uniform | 0.734 | 0.673 | 0.570 | 0.618 | 0.491 | 0.613 | 0.554 | 1.813 | 1.721 | 1.741 | 1.758 | 1.580 | 1.619 | 1.680 |
| TANI | 0.734 | 0.548 | 0.626 | 0.733 | 0.524 | 0.773 | 0.626 | 1.813 | 1.543 | 1.735 | 1.847 | 1.672 | 1.717 | 1.772 |
| | MSVD | | | | | | | UCF-101 | | | | | | |
| BCNI | 1.776 | 1.745 | 1.665 | 1.658 | 1.588 | 1.715 | 1.674 | 2.698 | 2.657 | 2.561 | 2.473 | 2.492 | 2.737 | 2.771 |
| Gaussian | 1.776 | 1.667 | 1.808 | 1.700 | 1.636 | 1.431 | 1.709 | 2.698 | 2.506 | 2.575 | 2.597 | 2.540 | 2.404 | 2.819 |
| HSCAN | 1.776 | 1.715 | 1.661 | 1.740 | 1.714 | 1.674 | 1.597 | 2.698 | 2.708 | 2.718 | 2.711 | 2.541 | 2.367 | 2.569 |
| SACN | 1.776 | 1.752 | 1.708 | 1.571 | 1.840 | 1.633 | 1.612 | 2.698 | 2.651 | 2.564 | 2.682 | 2.514 | 2.574 | 2.678 |
| Uniform | 1.776 | 1.722 | 1.683 | 1.733 | 1.575 | 1.586 | 1.657 | 2.698 | 2.589 | 2.843 | 2.817 | 2.344 | 2.532 | 2.616 |
| TANI | 1.776 | 1.590 | 1.748 | 1.772 | 1.643 | 1.728 | 1.773 | 2.698 | 2.504 | 2.657 | 2.759 | 2.759 | 2.697 | 2.675 |

Table 17: VBench and EvalCrafter Results. Baselines (Wang et al., 2023a; Ruan et al., 2024).

| Method | #Params | #Videos | VBench (↑) | | | | EvalCrafter (Quality) | | | | EvalCrafter (Consistency) | | | |
|---|---|---|---|---|---|---|---|---|---|---|---|---|---|---|
| | | | Motion Smoothness | Temporal Flickering | Human Action | Dynamic Degree | IS | ClipT | VQA_A | VQA_T | Action | Clip | Flow | Motion |
| *Baselines (zero-shot)* | | | | | | | | | | | | | | |
| ModelScopeT2V | 1.7B | 10M | 96.19 | 96.02 | 90.40 | 62.50 | 14.60 | — | 15.12 | 16.88 | 75.88 | — | 2.51 | 44 |
| ModelScopeT2V fine-tuned | 1.7B | 10M | 96.38 | 96.35 | 90.40 | 63.75 | 14.92 | — | 15.89 | 16.39 | 74.23 | — | 2.72 | 40 |
| DEMO w/o $\mathcal{L}_{\text{video-motion}}$ | 2.3B | 10M | — | — | — | — | 17.13 | — | 18.78 | 15.12 | 76.20 | — | 3.11 | 48 |
| DEMO | 2.3B | 10M | 96.09 | 94.63 | 90.60 | 68.90 | 17.57 | — | 19.28 | 15.65 | 78.22 | — | 4.89 | 58 |
| *CAT (ours)* | | | | | | | | | | | | | | |
| BCNI | 2.3B | 2M | 96.12 | 96.81 | 89.20 | 82.81 | 15.28 | 99.63 | 20.12 | 12.27 | 65.09 | 19.58 | 6.45 | 60.0 |
| Gaussian | 2.3B | 2M | 57.48 | 95.36 | 83.40 | 66.85 | 14.57 | 99.68 | 17.07 | 14.28 | 60.21 | 19.73 | 4.75 | 62.0 |
| Uniform | 2.3B | 2M | 57.18 | 93.67 | 85.00 | 76.95 | 14.22 | 99.65 | 17.45 | 13.35 | 64.71 | 19.58 | 6.38 | 62.0 |
| Clean | 2.3B | 2M | 56.86 | 94.76 | 82.60 | 72.90 | 14.65 | 99.66 | 14.81 | 12.10 | 63.90 | 19.66 | 4.96 | 60.0 |

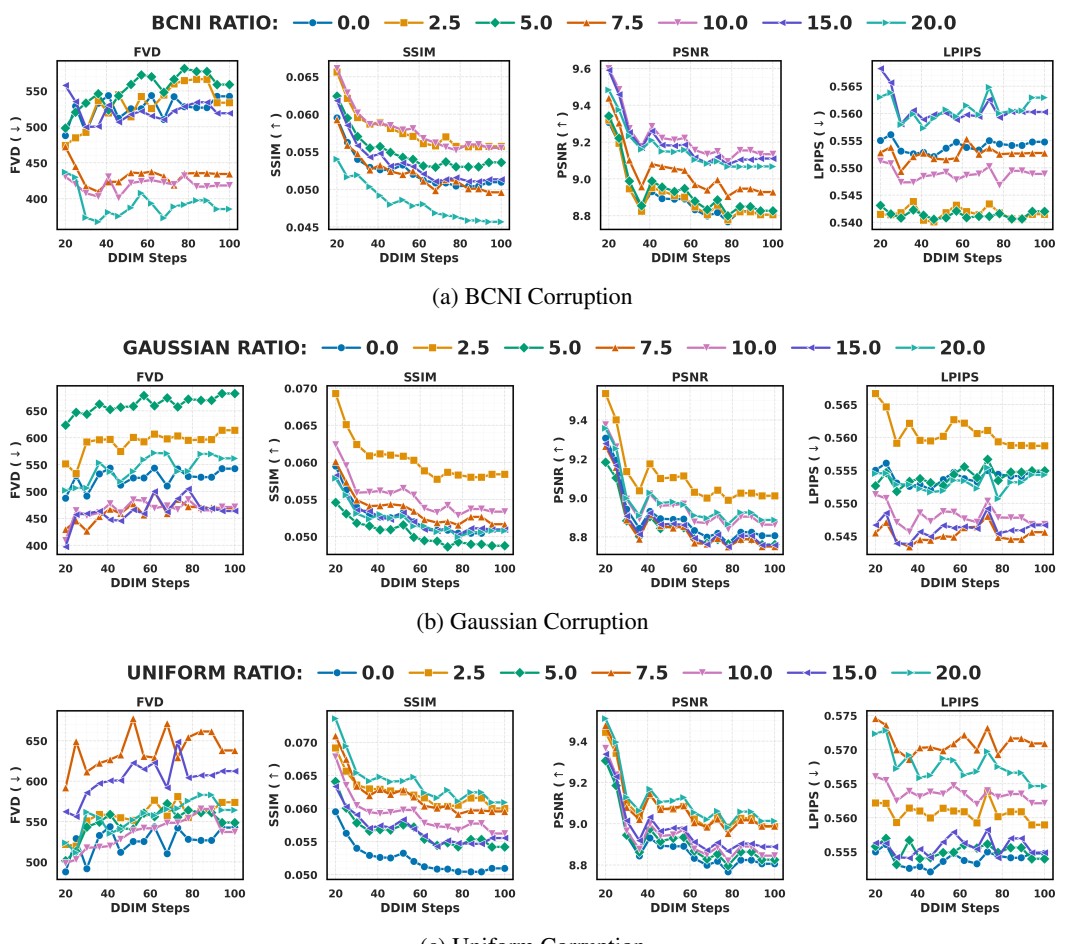

Figure 5: **Effect of DDIM Steps on Video Quality Metrics.** Each subfigure shows how FVD, SSIM, PSNR, and LPIPS vary with DDIM sampling steps across different corruption techniques.

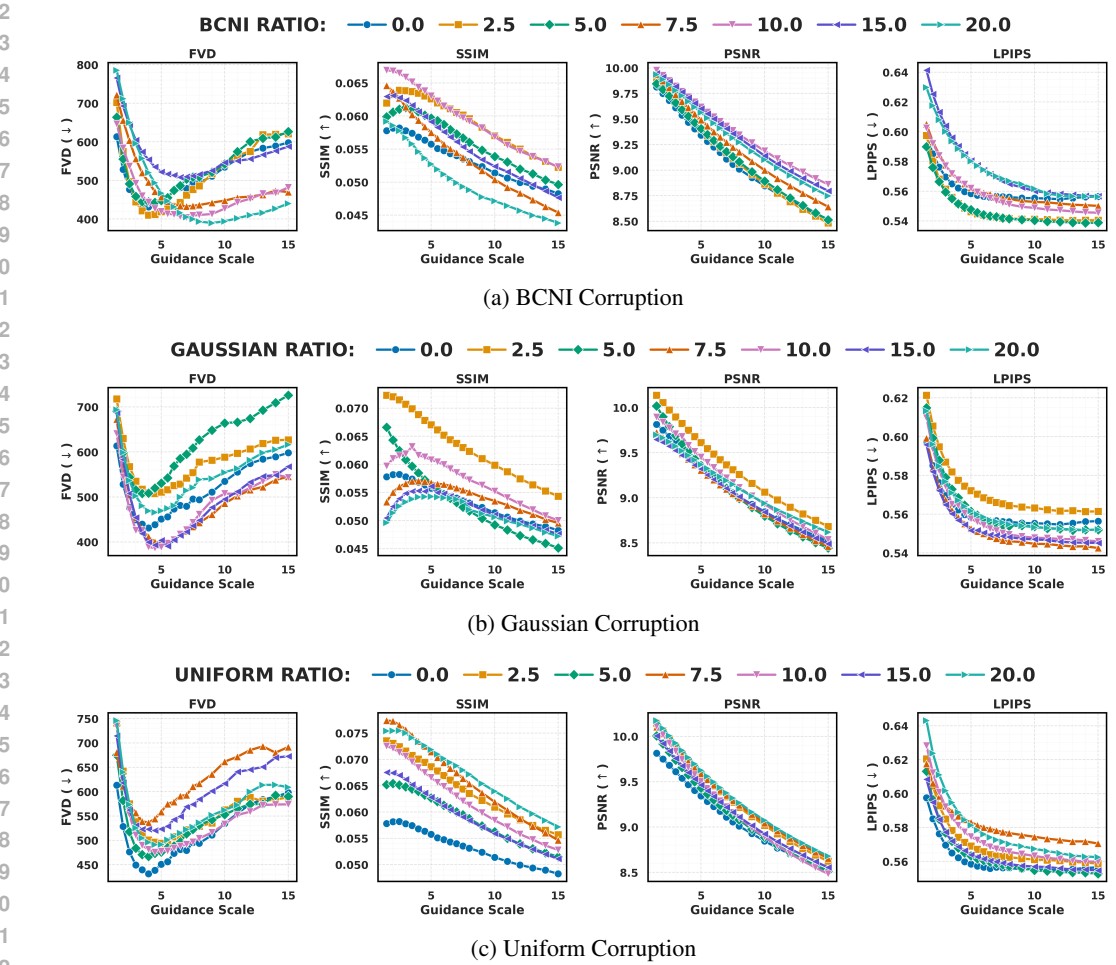

Figure 6: **Effect of Guidance Scale on Video Quality Metrics.** Each subfigure shows how FVD, SSIM, PSNR, and LPIPS vary with the guidance scale across different corruption techniques. Lower FVD and LPIPS and higher SSIM and PSNR indicate better generation quality.

Table 18: Averaged results across seeds. We report mean $\pm$ std for FVD, SSIM, PSNR, and LPIPS across representative corruption settings (full results in Appendix).

| Method | FVD $\downarrow$ | SSIM $\uparrow$ | PSNR $\uparrow$ | LPIPS $\downarrow$ |
|---|---|---|---|---|
| add_2.5 | $414.8 \pm 24.6$ | $0.0671 \pm 0.0004$ | $9.56 \pm 0.01$ | $0.5552 \pm 0.0015$ |
| swap_7.5 | $528.2 \pm 29.5$ | $0.0587 \pm 0.0004$ | $9.23 \pm 0.01$ | $0.5556 \pm 0.0012$ |
| replace_20 | $514.8 \pm 30.9$ | $0.0578 \pm 0.0004$ | $9.22 \pm 0.01$ | $0.5579 \pm 0.0013$ |
| perturb_10 | $413.6 \pm 11.6$ | $0.0643 \pm 0.0004$ | $9.50 \pm 0.01$ | $0.5480 \pm 0.0014$ |
| remove_15 | $517.5 \pm 35.7$ | $0.0595 \pm 0.0003$ | $9.27 \pm 0.01$ | $0.5547 \pm 0.0012$ |
| bcni_10 | $\mathbf{378.9 \pm 21.3}$ | $0.0687 \pm 0.0003$ | $9.66 \pm 0.01$ | $0.5589 \pm 0.0010$ |
| bcni_7.5 | $\mathbf{360.3 \pm 18.2}$ | $0.0642 \pm 0.0003$ | $9.51 \pm 0.01$ | $0.5562 \pm 0.0009$ |
| sacn_10 | $467.1 \pm 16.6$ | $0.0648 \pm 0.0004$ | $9.41 \pm 0.01$ | $0.5560 \pm 0.0009$ |
| sacn_15 | $466.2 \pm 15.5$ | $0.0671 \pm 0.0003$ | $9.49 \pm 0.01$ | $0.5556 \pm 0.0010$ |
| gaussian_10 | $417.6 \pm 23.8$ | $0.0642 \pm 0.0003$ | $9.54 \pm 0.01$ | $0.5517 \pm 0.0011$ |
| uniform_10 | $444.7 \pm 22.2$ | $0.0633 \pm 0.0005$ | $9.31 \pm 0.01$ | $0.5637 \pm 0.0010$ |

| WebVid-2M | | | | | | |
|---|---|---|---|---|---|---|
| **Operator** | Sens. ↓ | Var ↓ | Low | Mid | High | Risk |
| Gaussian | 55.2 | 1850 | 0.081 | 0.164 | 0.092 | 0.0 |
| Uniform | 42.7 | 1191 | 0.003 | 0.004 | 0.002 | 0.0 |
| TANI | 19.8 | 77.4 | 0.002 | 0.029 | **0.961** | 0.98 |
| SACN (ours) | **8.7** | **34.1** | **0.992** | 0.219 | 0.007 | 0.02 |
| BCNI (ours) | 49.6 | 2109 | 0.064 | **0.352** | **0.421** | 0.0 |

| MSR-VTT | | | | | | |
|---|---|---|---|---|---|---|
| **Operator** | Sens. ↓ | Var ↓ | Low | Mid | High | Risk |
| Gaussian | 68.5 | 2439 | 0.095 | 0.197 | 0.089 | 0.0 |
| Uniform | 38.6 | 1360 | 0.001 | 0.003 | 0.002 | 0.0 |
| TANI | 16.9 | 62.6 | 0.000 | 0.017 | 0.006 | **0.99** |
| SACN (ours) | **9.1** | **36.7** | **0.999** | 0.243 | 0.006 | 0.01 |
| BCNI (ours) | 51.0 | 2392 | 0.070 | **0.337** | **0.435** | 0.0 |

| MSVD | | | | | | |
|---|---|---|---|---|---|---|
| **Operator** | Sens. ↓ | Var ↓ | Low | Mid | High | Risk |
| Gaussian | 80.7 | 3899 | 0.051 | 0.104 | 0.147 | 0.30 |
| Uniform | 52.5 | 2378 | 0.0002 | 0.002 | 0.019 | 0.0 |
| TANI | 43.6 | 1328 | 0.058 | 0.029 | 0.020 | 0.0 |
| SACN (ours) | **12.8** | **101** | 0.008 | 0.0001 | 0.0 | **0.99** |
| BCNI (ours) | 95.8 | 7791 | 0.128 | **0.429** | **0.317** | 0.0 |

| UCF101 | | | | | | |
|---|---|---|---|---|---|---|
| **Operator** | Sens. ↓ | Var ↓ | Low | Mid | High | Risk |
| Gaussian | **17.9** | 201 | 0.0 | 0.0 | 0.0002 | 0.0 |
| Uniform | 46.3 | 2133 | 0.0 | 0.0002 | 0.0002 | 0.0 |
| TANI | 40.0 | 1597 | 0.001 | 0.014 | 0.127 | 0.0 |
| SACN (ours) | 26.9 | 719 | **0.447** | **0.851** | **0.452** | 0.18 |
| BCNI (ours) | 78.2 | 5982 | 0.241 | 0.114 | 0.066 | 0.0 |

Table 19: **Cross-dataset corruption robustness.** Each dataset (WebVid-2M, MSR-VTT, MSVD, UCF101) is reported independently. Columns show sensitivity (slope of FVD degradation), residual variance (fit stability), and win probabilities across corruption regimes (Low = 2.5–5%, Mid = 7.5–10%, High = 15–20%), plus a risk-adjusted robustness score. SACN achieves lowest sensitivity and variance across datasets, confirming smoother and more reliable degradation. BCNI dominates in mid/high regimes, especially on WebVid, MSR-VTT, and MSVD. Baselines collapse early, while TANI peaks only under extreme corruption but lacks stability.

