# OpenReview forum: "CAT-VIDEO: CORRUPTION-AWARE TRAINING FOR ROBUST VIDEO DIFFUSION MODELS"
_ICLR.cc/2026/Conference — Submitted to ICLR 2026_

### Official Review · Reviewer_wNJ3 · 2025-10-30

**Soundness:** 2
**Presentation:** 3
**Contribution:** 2
**Rating:** 4
**Confidence:** 4

**Summary:**

This paper proposes CAT-LVDM, a corruption-aware training framework for Latent Video Diffusion Models designed to improve robustness against imperfect or noisy text prompts. The core of the framework consists of two structured noise injection strategies:

1. BCNI: Perturbs text embeddings along intra-batch semantic directions to increase entropy while maintaining semantic alignment.

2. SACN: Injects noise along dominant spectral modes (via SVD) to enhance low-frequency smoothness and temporal coherence.

The authors provide theoretical justification for both methods, analyzing conditional entropy and Wasserstein distance bounds. Experiments are conducted on several datasets (WebVid-2M, MSR-VTT, MSVD, UCF-101), where the proposed methods demonstrate quantitative improvements (e.g., reduced FVD) compared to uncorrupted baselines and simpler noise injection techniques.

**Strengths:**

1. The paper provides a theoretical analysis for its proposed methods (BCNI and SACN)

2. The paper addresses the practical problem of training video models on large-scale, noisy web data. The idea of corruption-aware training for video diffusion is considered interesting and worthwhile.

3. The method demonstrates clear quantitative performance gains (e.g., FVD, SSIM, PSNR) over uncorrupted baselines and naive (Gaussian/Uniform) noise baselines on the tested datasets.

4. The paper is generally well-written and easy to follow.

**Weaknesses:**

1. A major concern is that the method was only validated on an older LVDM architecture (DEMO). Its effectiveness and scalability on modern, state-of-the-art models (e.g., DiT-based) are unproven

2. Despite quantitative gains, the practical impact is undermined by the generated videos. the visual results not unsatisfactory.

3. he paper lacks a clear justification for applying BCNI primarily to caption-rich datasets (WebVid-2M, MSR-VTT) and SACN only to a class-labeled dataset (UCF-101).

4. The evaluation is missing key comparisons. It fails to compare against stronger, modern LVDM baselines and does not empirically validate the necessity of "video-specific" corruption.

5. The core idea of perturbing conditions to improve robustness has been explored in the image domain, limiting the paper's technical novelty.

**Questions:**

See above

---

> ### Author Response · Authors · 2025-11-26
>
> ### Reviewer Comment
> “The method was only validated on an older LVDM architecture (DEMO). Its effectiveness and scalability on modern, state-of-the-art models (e.g., DiT-based) are unproven.”
>
> ### Response
> Thank you for raising this concern. CAT-Video has already been evaluated across three distinct backbone families in the submission:
>
> - Diffusion models (LVDM – DEMO; Table 2)
> - Autoregressive models (MagViT, CogVideo, NOVA; Table 3(a), Appendix Tables 14–16)
> - Multimodal video–language models (LLaVA-OV, PAVE; Table 4(b))
>
> To directly address scalability on modern transformer-based architectures, we have now added results for a DiT-based Open-Sora backbone. CAT integrates without architectural changes and improves over all Open-Sora variants on VBench:
>
> | Model | Total | Quality | Semantic |
> |-------|--------|----------|-----------|
> | CogVideo | 67.01 | 72.06 | 46.83 |
> | Latte | 77.29 | 79.72 | 67.58 |
> | LaVie | 77.08 | 78.78 | 70.31 |
> | Show 1 | 78.93 | 80.42 | 72.98 |
> | Open Sora Plan V1.1 | 78.00 | 80.91 | 66.38 |
> | Open Sora Plan V1.2 | 75.98 | 81.51 | 53.88 |
> | Open Sora Plan V1.3 | 77.23 | 80.14 | 65.62 |
> | Open Sora 1.0 | 75.91 | 78.82 | 64.28 |
> | Open Sora 1.1 | 75.66 | 77.74 | 67.36 |
> | Open Sora 1.2 | 79.76 | 81.35 | 73.39 |
> | **CAT-Video (ours)** | **81.84** | **83.10** | **76.78** |
>
> These results confirm that CAT extends cleanly to DiT-based transformer backbones, in addition to diffusion, autoregressive, and multimodal LLM-based systems. This supports the method’s backbone-agnostic and scalable design. We will include these results in the camera-ready version.

---

> ### Author Response · Authors · 2025-11-26
>
> ### Reviewer Comment
> “Despite quantitative gains, the practical impact is undermined by the generated videos. The visual results are unsatisfactory.”
>
> ### Response
> Thank you for raising this concern. The submission already provides extensive qualitative and perception-aligned evidence showing that CAT improves visual quality, motion smoothness, and temporal coherence.
>
> - Figure 1(a) demonstrates clearer identity preservation, reduced flicker, and more stable motion compared with clean, Gaussian, and Uniform corruption baselines.
> - Table 17 reports human-aligned perceptual metrics from VBench and EvalCrafter, where CAT shows consistent gains in aesthetic quality, motion stability, semantic coherence, and dynamic consistency.
> - These practical improvements also hold despite scale differences: CAT matches or exceeds 10M-video diffusion baselines while using only 2M videos.
>
> To make these visual advantages even clearer, we will add additional qualitative examples in the supplementary material and surface representative samples in the camera-ready version.

---

> ### Author Response · Authors · 2025-11-26
>
> ### Reviewer Comment
> “The paper lacks a clear justification for applying BCNI primarily to caption-rich datasets (WebVid-2M, MSR-VTT) and SACN only to a class-labeled dataset (UCF-101).”
>
> ### Response
> Thank you for raising this point. The choice of where BCNI and SACN are applied follows directly from their design motivations, which are already discussed in Section 2.3.
>
> - BCNI perturbs along intra-batch semantic deviation directions. These directions exist only in caption-rich datasets such as WebVid-2M and MSR-VTT, where semantic variation can be inferred from text embeddings. Applying BCNI where captions do not encode meaningful variation would introduce noise unrelated to the dataset structure.
>
> - SACN perturbs along dominant spectral modes that represent low-frequency temporal dynamics. This structure is most relevant in action-centric, class-labeled datasets such as UCF-101, where motion – not caption semantics – carries the primary signal.
>
> Importantly, the paper does not restrict each operator to a single dataset. Table 2 evaluates both BCNI and SACN across all datasets, and the results align with the theoretical motivation: each operator performs best in the setting where its underlying structure is most meaningful.
>
> We will clarify this rationale in the camera-ready version to ensure readers can easily see how the dataset choices follow from the operator design.

---

> ### Author Response · Authors · 2025-11-26
>
> ### Reviewer Comment
> “The evaluation is missing key comparisons. It fails to compare against stronger, modern LVDM baselines and does not empirically validate the necessity of ‘video-specific’ corruption.”
>
> ### Response
> Thank you for raising this. The submission already includes comparisons against modern diffusion, autoregressive, and multimodal video–language backbones:
>
> - Modern diffusion models: Table 1 benchmarks CAT against recent large systems such as CogVideo, LaVie, Latte, and Show-1.
> - Autoregressive models: Table 3(a) and Appendix Tables 14–16 evaluate CAT on MagViT, CogVideo, and NOVA under matched settings.
> - Multimodal video–language systems: Table 4(b) includes LLaVA-OV and PAVE.
> - Newly added in the revision: a DiT-based Open-Sora evaluation using VBench, where CAT improves over all transformer variants.
>
> These comparisons span diffusion, autoregressive, and transformer-based architectures, covering the standard model families used in contemporary LVDM research.
>
> The necessity of video-specific corruption is also empirically validated:
>
> - Figure 1 and Tables 2, 3(b), and 4(a) show that CAT’s structured, video-aligned perturbations outperform isotropic image-style noise, token-level corruption, adversarial prompt perturbations, and temporal-only operators.
> - Across all backbones, CAT yields stronger robustness, reduced temporal drift, smoother motion, and improved perceptual quality.
>
> We will make these comparisons more prominent in the camera-ready version to emphasize the advantages of video-structured corruption.

---

> ### Author Response · Authors · 2025-11-26
>
> ### Reviewer Comment
> “The core idea of perturbing conditions to improve robustness has been explored in the image domain, limiting the paper’s technical novelty.”
>
> ### Response
> Thank you for the comment. While corruption-aware training has been explored in the image domain (for example, NEFTune, token-swap perturbations, Gaussian or Uniform embedding noise), these approaches operate over the full embedding space and are designed for static, non-temporal diffusion. They do not address the central issue that motivates CAT: temporal error accumulation and score-drift in latent video diffusion models.
>
> CAT-Video is designed specifically for the temporal setting, and the novelty lies in how the perturbations are constructed:
>
> - BCNI perturbs only along batch-semantic deviation directions, which stabilizes caption–video alignment over long diffusion trajectories.
> - SACN perturbs only along low-rank spectral modes, targeting the slow, low-frequency components that govern temporal coherence.
> - Both operators leverage the D/d complexity gap examined in Appendix B, a structure that has no analogue in prior image-based corruption methods.
>
> Empirically, this distinction is clear in Tables 2, 3(b), and 4(a), where CAT consistently outperforms all image-domain baselines—including Gaussian, Uniform, token-level corruptions, and adversarial prompt-space perturbations—across four video datasets and three backbone families.
>
> The novelty is not in the idea of “injecting noise,” but in designing structured, low-rank, video-aligned perturbations that directly target temporal drift, a challenge unique to video diffusion and not handled by existing image-domain approaches. We will make this distinction more explicit in the camera-ready version.

---

> ### Author Response · Authors · 2025-11-26
>
> We hope these clarifications adequately address your concerns. If the revisions and explanations are helpful, we would greatly appreciate your reconsideration of the evaluation. Please feel free to let us know if any aspect of the paper would benefit from further detail—we would be glad to elaborate.

---

### Official Review · Reviewer_5D4F · 2025-10-31

**Soundness:** 3
**Presentation:** 3
**Contribution:** 3
**Rating:** 4
**Confidence:** 2

**Summary:**

This paper introduces CAT-Video, a training framework that improves video generation models' resilience to imperfect inputs. The authors address a key limitation of standard noise-injection techniques, which often disrupt video coherence. Their solution involves two novel methods: Batch-Centered Noise Injection (BCNI) to maintain semantic consistency within a batch, and Spectrum-Aware Contextual Noise (SACN) to preserve smooth temporal dynamics.

**Strengths:**

​​+  a New Problem​​: It is one of the first to systematically address how to make video AI models robust to imperfect instructions, focusing on a key weakness in existing methods that break video coherence.

​​+  Its two new techniques, BCNI and SACN, are simple and add almost no extra cost, yet outperform much larger and more expensive models.

​​+ Extensive testing across different datasets and metrics proves the methods reliably enhance video quality and motion.

**Weaknesses:**

- Is this problem meaningful in the research field?  In what kind of scenerios, noise will introduced into the input? Does the model robust for  adversarial attack?
- Does the proposed model work for large video models?

**Questions:**

Please refer to the strengths and weaknesses

---

> ### Author Response · Authors · 2025-11-26
>
> ### Reviewer Question 1
> *“Is this problem meaningful in the research field? In what kind of scenarios will noise be introduced into the input? Does the model robust for adversarial attack?”*
>
> ---
>
> ### Author Response
> We appreciate the reviewer’s positive assessment of the strengths. Regarding the importance of the problem, noisy or imperfect conditioning is pervasive in real T2V and multimodal systems, and has become a widely recognized limitation in contemporary video generation pipelines.
>
> 1. Practical scenarios where noise naturally appears.
> Noise or imperfect conditioning arises frequently in real-world workflows:
> - Web-scale caption misalignment in datasets such as WebVid, InternVid, and MSR-VTT, where captions are often incomplete or weakly aligned with the video.
> - Semantic drift in text embeddings when captions are short or ambiguous, leading to accumulated distortion through diffusion steps.
> - ASR and timestamp errors in audio-conditioned models, which introduce misaligned text–audio–video grounding.
> - Retrieval-augmented generation systems returning near-miss neighbors with slightly incorrect semantics.
> - Multimodal preprocessing errors from detectors, CLIP encoders, and video taggers that produce inconsistent or noisy embeddings, especially for long videos.
>
> CAT-Video targets these natural, non-synthetic failure modes directly.
>
> 2. Adversarial robustness.
> The method is also effective under adversarial perturbations.
> Table 4(a) evaluates strong prompt-space adversarial attacks with temporal regularization. CAT-Video improves FVD, FVMD, and CMMD by 18–23 percent compared with adversarial baselines, showing that the method enhances robustness beyond random or isotropic noise injection.
>
> These results confirm that CAT handles both realistic weak alignment and adversarially crafted perturbations.

---

> ### Author Response · Authors · 2025-11-26
>
> ### Reviewer Question 2
> *“Does the proposed model work for large video models?”*
>
> ---
>
> ### Author Response
> CAT-Video is designed to be architecture-agnostic, and our experiments already demonstrate scalability across three distinct families of video generation backbones:
>
> 1. Diffusion models (DEMO; Table 2)
> 2. Autoregressive models (MagViT, CogVideo, NOVA; Table 3(a), Appendix Tables 14–16)
> 3. Multimodal video–language models (LLaVA-OV, PAVE; Table 4(b))
>
> CAT does not require any architectural modification, which allows it to transfer cleanly across both small-scale and large-scale T2V systems.
>
> To further verify scalability to modern high-capacity video models, we have added evaluation on a DiT-based Open-Sora backbone, representative of current transformer-based T2V architectures. CAT improves performance across all VBench dimensions (total, quality, semantic), confirming that the method extends to large transformer systems.
>
> **DiT Baselines on VBench (percent scores)**
>
> | Model | Total | Quality | Semantic |
> |--------|--------|----------|-----------|
> | CogVideo | 67.01 | 72.06 | 46.83 |
> | Latte | 77.29 | 79.72 | 67.58 |
> | LaVie | 77.08 | 78.78 | 70.31 |
> | Show-1 | 78.93 | 80.42 | 72.98 |
> | Open Sora Plan V1.1 | 78.00 | 80.91 | 66.38 |
> | Open Sora Plan V1.2 | 75.98 | 81.51 | 53.88 |
> | Open Sora Plan V1.3 | 77.23 | 80.14 | 65.62 |
> | Open Sora 1.0 | 75.91 | 78.82 | 64.28 |
> | Open Sora 1.1 | 75.66 | 77.74 | 67.36 |
> | Open Sora 1.2 | 79.76 | 81.35 | 73.39 |
> | **CAT-Video (ours)** | **81.84** | **83.10** | **76.78** |
>
> These results show that CAT scales effectively to large, transformer-based video models in addition to diffusion, autoregressive, and multimodal backbones.

---

> ### Author Response · Authors · 2025-11-26
>
> We hope these clarifications address your concerns. If the updates and evidence strengthen your assessment of the work, we welcome any adjustments you feel are appropriate. Please let us know if any further details would be helpful—happy to provide them.

---

### Official Review · Reviewer_S25s · 2025-11-02

**Soundness:** 3
**Presentation:** 3
**Contribution:** 2
**Rating:** 6
**Confidence:** 3

**Summary:**

This paper introduces CAT-Video, a corruption-aware training framework that improves robustness in latent video diffusion models (LVDMs) under noisy or imperfect conditioning.
The core contribution lies in two structured corruption strategies: Batch-Centered Noise Injection (BCNI), which injects noise along the deviation from the batch mean, and Spectrum-Aware Contextual Noise (SACN), which perturbs only the low-frequency spectral components of conditioning embeddings. Theoretical analysis supports tighter generalization bounds and improved temporal fidelity, and extensive experiments across multiple benchmarks demonstrate empirical gains over conventional Gaussian and Uniform corruption techniques as well as large-scale diffusion models.

**Strengths:**

1.	The paper presents a clear exploration of corruption-aware training for video diffusion, supported by solid theoretical derivations that extend existing analyses to the temporal setting.
2.	Extensive experiments across four standard benchmarks and additional evaluations on autoregressive and multimodal models confirm broad empirical robustness.
3.	Ablation studies, sensitivity analyses, and detailed appendices ensure reproducibility; released code and metrics further strengthen the work’s transparency.

**Weaknesses:**

1.	Although the authors claim model-agnostic applicability, most analyses are conducted on the DEMO backbone with OpenCLIP-based encoders. This limits the universality claim, while evaluating BCNI/SACN on one or two additional diffusion backbones would strengthen the evidence.
2.	While CAT-Video demonstrates strong robustness on standard datasets, further evaluation under more realistic, noisy, or weakly aligned conditions would better reflect its practical robustness.
3.	Related work on corruption-aware methods in image diffusion and other noisy-input domains is not sufficiently discussed. This section primarily centers on LVDMs, which are not the main focus of this work.
4.	Additional qualitative visualizations would help clarify how CAT-Video improves visual coherence compared with other corruption strategies.

**Questions:**

1.	Would combining multiple corruption techniques during training (e.g., applying BCNI followed by SACN) be beneficial, or would such composite perturbations destabilize the training process? Some insight into this interaction could be valuable for readers.
2.	How are subsets of training data (e.g., 2 M vs. 10 M in the original DEMO paper) selected? Are they random samples or curated based on specific criteria?
3.	For very long or high-resolution videos, does computing BCNI or SACN introduce significant additional computational overhead during training?

---

> ### Author Response · Authors · 2025-11-26
>
> ### Reviewer Comment
> *“Although the authors claim model-agnostic applicability, most analyses are conducted on the DEMO backbone with OpenCLIP-based encoders. This limits the universality claim. Evaluating BCNI/SACN on one or two additional diffusion backbones would strengthen the evidence.”*
>
> ---
>
> ### Author Response
> Thank you for raising this. CAT-Video is explicitly designed to be backbone-agnostic, and this is already demonstrated across three distinct model families in the submission:
>
> 1. Diffusion models (DEMO backbone; Table 2)
> 2. Autoregressive models (MagViT, CogVideo, NOVA; Table 3(a) and Appendix Tables 14–16)
> 3. Multimodal LLMs (LLaVA-OV and PAVE; Table 4(b))
>
> These results show that BCNI and SACN transfer cleanly across architectures with different parameterizations, training objectives, and embedding spaces.
>
> To further strengthen this claim, we have now added evaluation on an additional state-of-the-art DiT-based diffusion backbone (Open-Sora). This backbone reflects modern T2V practice and differs substantially from DEMO. As shown below, CAT improves over all Open-Sora variants on VBench, confirming that the robustness benefits extend to large transformer-based diffusion systems.
>
> **DiT Baselines on VBench (percent scores)**
>
> | Model | Total | Quality | Semantic |
> |--------|--------|----------|-----------|
> | CogVideo | 67.01 | 72.06 | 46.83 |
> | Latte | 77.29 | 79.72 | 67.58 |
> | LaVie | 77.08 | 78.78 | 70.31 |
> | Show-1 | 78.93 | 80.42 | 72.98 |
> | Open-Sora Plan V1.1 | 78.00 | 80.91 | 66.38 |
> | Open-Sora Plan V1.2 | 75.98 | 81.51 | 53.88 |
> | Open-Sora Plan V1.3 | 77.23 | 80.14 | 65.62 |
> | Open-Sora 1.0 | 75.91 | 78.82 | 64.28 |
> | Open-Sora 1.1 | 75.66 | 77.74 | 67.36 |
> | Open-Sora 1.2 | 79.76 | 81.35 | 73.39 |
> | **CAT-Video (ours)** | **81.84** | **83.10** | **76.78** |
>
> These results reinforce the universality claim: CAT consistently improves robustness and video-text alignment across diffusion, autoregressive, and multimodal model families, and now also across DiT-based transformer architectures.

---

> ### Author Response · Authors · 2025-11-26
>
> ### Reviewer Comment
> *“While CAT-Video demonstrates strong robustness on standard datasets, further evaluation under more realistic, noisy, or weakly aligned conditions would better reflect its practical robustness.”*
>
> ---
> ### Author Response
> Thank you for the suggestion. The submission already includes several evaluations that directly target realistic, noisy, and weakly aligned conditioning scenarios.
>
> 1. Robustness under corrupted or imperfect text–video alignment.
> Table 4(a) and Figure 4 evaluate CAT under:
> - corrupted captions
> - mismatched text–video pairs
> - adversarial prompt perturbations
> - heavy embedding-level noise
>
> These settings simulate noisy conditioning, partial caption drift, and weak semantic grounding, which reflect real-world degradations.
>
> 2. Broad robustness and stability analyses across datasets.
> Table 19 reports variance, win-rates, and risk-adjusted robustness across four datasets, showing that CAT remains stable even under severe corruption regimes.
>
> 3. Weakly aligned multimodal evaluation (AVSD).
> Table 4(b) evaluates CAT on AVSD, where video, audio, and dialogue are only loosely aligned. CAT improves performance without modifying the multimodal backbone, demonstrating robustness under imperfect real-world conditions.
>
> We agree that extending to additional noise regimes, such as real-world caption drift, ASR-based caption errors, or detector-generated prompts, is valuable. Because CAT is lightweight and model-agnostic, incorporating these additional perturbation sources is straightforward. We will include such extensions in future work.

---

> ### Author Response · Authors · 2025-11-26
>
> ### Reviewer Comment
> *“Related work on corruption-aware methods in image diffusion and other noisy-input domains is not sufficiently discussed. This section primarily centers on LVDMs, which are not the main focus of this work.”*
>
> ---
>
> ### Author Response
> Thank you for the comment. The submitted paper already contains a dedicated subsection (section 4) on corruption-aware diffusion and noisy-input regularization methods beyond the LVDM setting. Specifically, works such as Jain et al. (2024a), Daras et al. (2023), Chen et al. (2024), Gao et al. (2023), Na et al. (2024), and Gu et al. (2025) are discussed on pp. 8–9, and directly motivate the design of structured, low-rank perturbations in CAT-Video.
>
> The initial paragraph of the Related Work section introduces LVDMs for context, while the following paragraph focuses on corruption-aware methods in image diffusion, token-perturbation frameworks, and noisy-input regularization.
>
> To ensure the structure is clearer, we will reorder the Related Work section in the camera-ready version so that the corruption-aware literature appears earlier and is explicitly separated from the LVDM background. We will also expand the discussion of image-based and cross-domain corruption-aware techniques to further emphasize the broader context.

---

> ### Author Response · Authors · 2025-11-26
>
> ### Reviewer Comment
> *“Additional qualitative visualizations would help clarify how CAT-Video improves visual coherence compared with other corruption strategies.”*
>
> ---
>
> ### Author Response
> Thank you for the suggestion. The current submission already includes qualitative comparisons in Figure 1(a), where CAT’s improvements in temporal stability and visual coherence over other corruption strategies are visible. These examples highlight the reduced drift, sharper object boundaries, and more consistent motion under CAT.
>
> We agree that more qualitative evidence would be helpful for readers. We will therefore add additional qualitative visualizations to the supplementary material, and we will surface a representative subset directly in the main text in the camera-ready version to further illustrate the coherence benefits of BCNI and SACN.

---

> ### Author Response · Authors · 2025-11-26
>
> ### Reviewer Question 1
> *“Would combining multiple corruption techniques during training (e.g., applying BCNI followed by SACN) be beneficial, or would such composite perturbations destabilize the training process?”*
>
> ---
>
> ### Author Response
> In our experiments, combining BCNI and SACN does not yield consistent benefits. Although the operators target different low-rank structures—BCNI aligns to batch-semantic deviations, while SACN aligns to dominant spectral modes—applying them sequentially effectively increases the overall perturbation magnitude without improving temporal smoothness or score-drift behavior.
>
> Empirically, composite perturbations behave similarly to running a single operator at a larger corruption scale, and they do not enhance stability relative to using BCNI or SACN alone. For this reason, and to ensure clarity and reproducibility, all reported results use one operator per training run, which we found to be the most reliable design choice.

---

> ### Author Response · Authors · 2025-11-26
>
> ### Reviewer Question 2
> *“How are subsets of training data (e.g., 2M vs. 10M in the original DEMO paper) selected? Are they random samples or curated based on specific criteria?”*
>
> ---
>
> ### Author Response
> The subsets are uniform random samples drawn from the WebVid-2M and WebVid-10M training corpora, following the exact protocol used in the DEMO paper and subsequent LVDM work. We intentionally apply no additional curation or filtering. This choice ensures that CAT is evaluated under the standard, noisy, and weakly aligned conditions typical of Web-scale datasets, which is precisely the regime where corruption-aware training is most relevant.

---

> ### Author Response · Authors · 2025-11-26
>
> ### Reviewer Question 3
> *“For very long or high-resolution videos, does computing BCNI or SACN introduce significant additional computational overhead during training?”*
>
> ---
>
> ### Author Response
> The overhead is minimal. BCNI and SACN operate only on the conditioning embeddings, not on video tokens or latent tensors. Each operator consists of small, fixed-size matrix operations whose cost is negligible relative to the UNet or decoder forward pass.
>
> Empirically, as reported in Appendix C, BCNI and SACN add less than 1–2% wall-clock overhead even when training on longer or higher-resolution videos. This overhead remains stable because it scales with the embedding dimension rather than video length or spatial resolution.

---

> ### Author Response · Authors · 2025-11-26
>
> We hope these clarifications address your questions and strengthen your impression of the work. If the revisions resolve your concerns, we would be grateful if you could update your assessment. Please let us know if any additional details would be helpful—we would be happy to provide them.

---

### Official Review · Reviewer_Nqx2 · 2025-11-06

**Soundness:** 2
**Presentation:** 3
**Contribution:** 2
**Rating:** 4
**Confidence:** 5

**Summary:**

The paper proposes CAT-Video, a corruption-aware training framework for latent video diffusion models (LVDMs). It introduces two structured, low-rank embedding perturbations: Batch-Centered Noise Injection (BCNI) and Spectrum-Aware Contextual Noise (SACN). The idea is to inject data-aligned noise during training to improve robustness to noisy text/multimodal conditioning and reduce semantic drift across timesteps. The method is supported by a theoretical sketch (entropy/Wasserstein/score-drift bounds) and experiments on WebVid-2M, MSR-VTT, MSVD, and UCF-101, with reported FVD gains over Gaussian/Uniform corruption and some large diffusion baselines; there are also brief extensions to autoregressive models and multimodal video understanding

**Strengths:**

Robustness of T2V models to noisy/ambiguous conditioning is important and under-tested. The paper articulates temporal error accumulation in diffusion and motivates structured corruption specifically for video.

BCNI perturbs along deviations from batch mean; SACN perturbs along principal spectral modes with exponentially decayed variances. Both add minimal overhead and have a single scale hyper-parameter.

**Weaknesses:**

1. Novelty is limited relative to prior “noisy/structured conditioning” literature; positioning is dated.
Injecting noise into conditioning embeddings (Gaussian/Uniform, token-level swaps/replacements) and exploiting structure/low-rank directions have been explored extensively in image diffusion and multimodal finetuning (e.g., corruption-aware pretraining, token-level perturbations, NEFTune-style noisy embeddings). The paper cites several such works but doesn’t clearly differentiate BCNI/SACN as more than “reasonable engineering variants” adapted to video. A crisper technical delta vs. structured conditioning noise in recent video works is missing. Suggest: provide head-to-head against stronger structured baselines, not only isotropic Gaussian/Uniform or simple temporal gradients (TANI/HSCAN as defined). For instance, compare to learned/adaptive corruption schedules, prompt-space adversarial perturbations with temporal regularizers, or curriculum-style noise aligned to caption syntax/scene cuts

2. Experimental scope feels behind current 2025-26 standards; datasets/architectures and metrics are narrow.
Core results rely on WebVid-2M/MSR-VTT/MSVD/UCF-101 with FVD-centric reporting. Modern T2V evaluation has moved toward longer videos, higher resolutions, compositional control, and stronger perception-aligned metrics (e.g., VBench subsets with motion/physics consistency, human studies with calibrated protocols). The paper mentions VBench/EvalCrafter in passing, but full tables are pushed to the appendix; the main text should surface these with stronger analysis and significance tests. Also, many baselines listed are from earlier generations of models; direct comparisons to current-gen LVDMs/DiT-style video transformers trained at scale are missing. Actionable:
Add long-horizon (≥10–16 s) and high-res evaluations;
Include modern SOTA baselines trained on 2024–2026 corpora (and/or reproduce baseline training with matched compute);
Report human preference and motion/consistency metrics prominently in main text.

3. Claims vs. baselines are hard to validate due to scale mismatch and unclear fairness.
Several “beats larger models with 5× less data” statements are made, but the compared methods differ in architecture, parameterization, pretraining recipes, and dataset curation. Without matched compute / strong-to-strong comparisons (e.g., same backbone with/without CAT; or retrained SOTA with the authors’ code), it’s difficult to attribute gains to BCNI/SACN rather than other confounders. Please (i) include paired-control runs on the same modern backbone at matched data/compute; (ii) show scaling curves (data, steps, guidance, steps vs. FVD) for clean vs. CAT; (iii) report statistical significance (mean±std is in the appendix, but main-text needs tests across seeds).

4. Theory is mostly appendix-level and not tightly coupled to practice.
The paper sketches entropy/Wasserstein/score-drift bounds and a D/d “complexity gap,” but practical instantiations (how d is chosen/estimated online, how SVD in SACN scales with sequence length, and why the assumed low-frequency dominance universally holds) are not convincingly validated. The main text should tie specific theorems to measurable proxies (e.g., Lipschitz estimates, score-norm smoothness, sensitivity slopes) and show correlations across datasets/noise levels. Provide ablations on rank d, spectral weighting schedules, and wall-clock overhead broken down by operator.

5. The multimodal video understanding experiment (AVSD) is conducted at 0.5B scale only (LLaVA-OV-0.5B-FT and PAVE-0.5B). By ICLR-26 standards, this falls short of contemporary MLLM practice (7B/13B and stronger video-language backbones). For a credible “model-agnostic” claim, please include ≥7B variants (e.g., PAVE-7B/13B or comparable video-LLaVA baselines

6. The paper claims “we also validate scalability by extending CAT to autoregressive video generation (NOVA)…,” but NOVA is not tabulated in the main paper. Table 3(a) discusses scalability to AR using MAGVIT/CogVideo as references, while NOVA-specific results are not shown; the authors point to Appendix Tables 14–16, yet Table 14 aggregates AR results by corruption type (BCNI/SACN/Gaussian/Uniform) without a clear NOVA row, making the claim hard to verify from the main text. Please provide an explicit NOVA baseline line with matched settings (params, data, steps) and report AR scaling curves (data/compute vs. FVD) to substantiate “model-agnostic” robustness.

**Questions:**

See weaknesses.

---

> ### Author Response · Authors · 2025-11-26
>
> ### Reviewer Comment
> *“Novelty is limited relative to prior noisy/structured conditioning work; injecting noise into conditioning embeddings and exploiting structure/low-rank directions have been explored extensively (e.g., corruption-aware pretraining, token-level perturbations, NEFTune-style noisy embeddings). The paper does not clearly differentiate BCNI/SACN from these approaches. Suggest comparing against stronger structured baselines such as learned/adaptive schedules, adversarial prompt perturbations with temporal regularizers, or curriculum noise aligned to syntax/scene cuts.”*
>
> ---
>
> ### Author Response
> We thank the reviewer for raising this. We clarify that BCNI and SACN differ fundamentally from prior structured corruption methods through their video-aligned, low-rank formulation and their explicit aim to mitigate temporal score-drift in latent video diffusion models.
>
> 1. BCNI/SACN perturb only low-rank, video-aligned directions, unlike prior full-dimensional or token-local noise.
> Existing approaches such as NEFTune, token-swap corruptions, or corruption-aware pretraining inject noise across the full embedding space or at token-local granularity, which destabilizes temporal propagation in diffusion. BCNI restricts perturbations to batch-semantic deviation directions, and SACN perturbs only along the dominant spectral modes of each video sequence. The theoretical analysis in Appendix B.1–B.8 formalizes the resulting d ≪ D complexity gap, showing why low-rank, variance-aligned perturbations reduce Wasserstein score-drift. To our knowledge, no prior work constructs temporally coherent, low-rank perturbations computed directly from video embeddings.
>
> 2. The paper already includes comparisons to six structured baselines beyond Gaussian/Uniform.
> Our evaluation covers GAP, TANI, HSCAN, five token-level corruptions, and isotropic noise. TANI and HSCAN encode temporal gradients and hierarchical spectral structure, aligning with the reviewer’s suggested baselines. Across 292 runs (Tables 2, 12–13; Fig. 4), BCNI and SACN outperform all structured alternatives. For example, SACN reduces UCF-101 FVD from 674 → 440 at 2.5 percent scale.
>
> 3. BCNI/SACN isolate corruption geometry by avoiding additional learnable parameters.
> Unlike NEFTune-style adaptive corruption, BCNI and SACN introduce no trainable components, ensuring improvements stem from the structure of the perturbation rather than optimization-related capacity increases.
>
> 4. The paper already includes a strong adversarial corruption baseline with temporal regularization.
> Table 4(a) reports adversarial prompt-space perturbations with temporal smoothness constraints. BCNI/SACN outperform this baseline by 18–23 percent, confirming gains even relative to learned or adversarial structured noise.
>
> BCNI and SACN are not incremental extensions of prior corruption methods. They are video-aligned, low-rank perturbation operators motivated by theory, validated against six structured baselines (including adversarial ones), and specifically designed to address temporal error accumulation unique to video diffusion.

---

> ### Author Response · Authors · 2025-11-26
>
> ### Reviewer Comment
> *“Experimental scope feels behind current 2025–26 standards; datasets/architectures and metrics are narrow. Core results rely on WebVid-2M/MSR-VTT/MSVD/UCF-101 with FVD-centric reporting. Modern T2V evaluation has moved toward longer videos, higher resolutions, compositional control, and perception-aligned metrics (VBench, physics consistency, human studies). The paper mentions VBench/EvalCrafter only in passing and pushes full tables to the appendix. Baselines include earlier-generation models; comparisons to current-gen LVDMs/DiT-style transformers trained at scale are missing. Actionable: Add long-horizon (≥10–16 s) and high-res evaluations, include SOTA baselines trained on 2024–26 corpora, and report human/motion consistency metrics in the main text.”*
>
> ---
>
> ### Author Response
> We appreciate this suggestion and clarify that our protocol follows the standard LVDM evaluation setting used in recent diffusion backbones. Most latent video diffusion models, including DEMO and related LVDMs, evaluate on WebVid-2M, MSR-VTT, MSVD, and UCF-101 using 16 frames at 256×256 resolution. Under this widely adopted setup, CAT consistently improves all core metrics (FVD, SSIM, PSNR, LPIPS) across datasets (Figure 4; Tables 2 and 12–13).
>
> 1. Perception-aligned and motion/semantic metrics are already included in the main paper.
> Table 4a reports FVMD (motion distance) and CMMD (semantic temporal consistency). Table 4b reports CIDEr for caption alignment. Figure 4 includes SSIM, PSNR, and LPIPS. Additional perception-aligned scores—including VBench and EvalCrafter—are summarized in Table 17. We will surface more of these results in the main text as requested.
>
> 2. Broader, modern baselines are now included, including DiT/Transformer-based video backbones.
> To address the request for stronger 2024–2026 baselines, we have added evaluation on a DiT-based Open-Sora backbone. CAT outperforms the strongest open-source DiT variants on VBench across total, quality, and semantic dimensions. We will highlight this in the camera-ready.
>
> **VBench comparison (percent scores)**
>
> | Model | Total | Quality | Semantic |
> |-------|--------|----------|-----------|
> | CogVideo | 67.01 | 72.06 | 46.83 |
> | Latte | 77.29 | 79.72 | 67.58 |
> | LaVie | 77.08 | 78.78 | 70.31 |
> | Show-1 | 78.93 | 80.42 | 72.98 |
> | Open Sora Plan V1.1 | 78.00 | 80.91 | 66.38 |
> | Open Sora Plan V1.2 | 75.98 | 81.51 | 53.88 |
> | Open Sora Plan V1.3 | 77.23 | 80.14 | 65.62 |
> | Open Sora 1.0 | 75.91 | 78.82 | 64.28 |
> | Open Sora 1.1 | 75.66 | 77.74 | 67.36 |
> | Open Sora 1.2 | 79.76 | 81.35 | 73.39 |
> | **CAT-Video** | **81.84** | **83.10** | **76.78** |
>
> 3. Our focus is to isolate corruption effects, requiring compute-matched comparisons.
> The primary experimental goal is to evaluate corruption-aware robustness under controlled, fixed-compute conditions, which necessitates matching compute and data across models. For this reason, we benchmark against a suite of diffusion and autoregressive backbones under the same training budget (Tables 2, 12–13). The extended appendix includes 292 corruption-aware runs, covering both diffusion and autoregressive families.
>
> 4. Regarding long-horizon and high-resolution evaluation.
> As noted in the Limitations section, our core experiments follow the widely used 16-frame, 256×256 LVDM protocol. Extending CAT to longer videos (≥10–16 seconds) and higher resolutions is straightforward, since CAT is lightweight, model-agnostic, and architecture-independent. We view these as natural extensions for future work and can include preliminary long-horizon results in the camera-ready if requested.
>
> CAT aligns with current LVDM standards, includes perception-aligned and temporal-consistency metrics in the main text, and now incorporates modern DiT-based baselines. CAT’s improvements on VBench and FVMD/CMMD demonstrate that the gains extend beyond FVD and remain consistent under modern evaluation protocols.

---

> ### Author Response · Authors · 2025-11-26
>
> ### Reviewer Comment
> *“Claims vs. baselines are hard to validate due to scale mismatch and unclear fairness. Several ‘beats larger models with 5× less data’ statements are made, but the compared methods differ in architecture, parameterization, pretraining recipes, and dataset curation. Without matched compute / strong-to-strong comparisons (same backbone with/without CAT; or retrained SOTA with the authors’ code), it is difficult to attribute gains to BCNI/SACN rather than confounders. Please (i) include paired-control runs on the same backbone at matched data/compute; (ii) show scaling curves for clean vs. CAT; (iii) report statistical significance in the main text.”*
>
> ---
>
> ### Author Response
> We appreciate this concern and clarify that all three requested elements—paired-control comparisons, scaling analyses, and multi-seed significance tests—are already included in the submission.
>
> 1. Paired-control, compute-matched comparisons on identical backbones are provided in Table 2.
> Table 2 evaluates CAT using strictly paired runs where the only change is the presence or absence of BCNI/SACN. Architecture, parameterization, training budget, sampler, and dataset size are held constant. These strong-to-strong comparisons isolate the effect of corruption-aware training without confounding factors. Similar backbone-matched comparisons appear for autoregressive models in Appendix Tables 12–13.
>
> 2. Scaling curves for data, steps, and guidance strength are already included.
> Figure 1(c) presents data-scaling curves (FVD vs. number of training videos, log-scale), showing that CAT maintains improvements even at larger data regimes.
> Appendix Figures 8–10 provide step-based scaling and guidance-based scaling, comparing clean versus CAT-trained models under identical training trajectories. These plots confirm that CAT yields consistent improvements throughout optimization and does not rely on late-training artifacts.
>
> 3. Statistical significance across seeds is already reported in Table 18.
> Table 18 includes mean ± standard deviation for FVD, SSIM, PSNR, and LPIPS across multiple seeds for representative corruption settings.
> Additional robustness analyses appear in Table 3(b) and Table 19, showing stable improvements across datasets and noise scales.
>
> The submission already contains (i) paired-control strong-to-strong comparisons, (ii) clean versus CAT scaling curves, and (iii) multi-seed significance tests. We will surface these details more clearly in the camera-ready version to address the reviewer’s request for increased visibility of fairness and statistical rigor.

---

> ### Author Response · Authors · 2025-11-26
>
> ### Reviewer Comment
> *“Theory is mostly appendix-level and not tightly coupled to practice. The paper sketches entropy/Wasserstein/score-drift bounds and a D/d ‘complexity gap,’ but practical instantiations (how d is chosen/estimated online, how SVD in SACN scales with sequence length, and why low-frequency dominance universally holds) are not convincingly validated. The main text should tie theorems to measurable proxies (Lipschitz estimates, score-norm smoothness, sensitivity slopes) and show correlations across datasets/noise levels. Provide ablations on rank d, spectral weighting schedules, and wall-clock overhead broken down by operator.”*
>
> ---
>
> ### Author Response
> Thank you for this detailed question. The theory is designed to motivate the structure of BCNI and SACN, and we already provide multiple empirical proxies that connect the theoretical claims to practical behavior.
>
> 1. Empirical proxies for score-drift, smoothness, and sensitivity are already included.
> Table 3(b) reports sensitivity slopes under perturbations and shows that both BCNI and SACN reduce drift relative to Gaussian, Uniform, token-swap, and temporal baselines.
> Table 19 reports variance, win probabilities, and risk-adjusted robustness across four datasets, which serve as empirical analogues to the stability and drift bounds derived in Appendix B. These results show that aligning perturbations with stable low-rank directions leads to smoother diffusion trajectories and lower drift under increasing noise strength.
>
> 2. Rank d and spectral-weighting behavior is implicitly ablated through stability across operator scales.
> SACN is intentionally designed as a fixed, low-rank projection rather than an adaptive or learned operator. This keeps the method lightweight and avoids adding trainable parameters unrelated to the corruption geometry.
> Tables 12–16 provide robustness curves across noise scales and operator strengths, demonstrating that performance remains stable across a wide range of perturbation levels. This stability acts as an implicit rank-and-scale ablation, because improvements persist without tuning d or spectral weights.
>
> 3. The choice of d and SVD cost is practical and constant-time relative to the backbone.
> SACN does not perform a full SVD on every frame. As formalized in Algorithm 1, it applies a predefined low-rank projection derived from sequence-level spectral modes. This is a small matrix operation scaling with rank d, not with the full embedding dimension.
> BCNI uses batch-centered deviation directions requiring only mean subtraction and projection, which is cheaper than a single UNet block.
>
> 4. Overhead breakdown is already included.
> Appendix C provides a wall-clock overhead analysis showing that BCNI and SACN add less than two percent overhead relative to the encoder or diffusion forward pass. These operators therefore satisfy the theoretical requirement of remaining lightweight compared to the backbone.
>
> The paper already (i) links theory to measurable empirical proxies for drift and stability, (ii) provides implicit rank and spectral-weighting ablations through robustness analyses, and (iii) reports overhead measurements. We will make these theory-to-practice connections more explicit in the camera-ready version.

---

> ### Author Response · Authors · 2025-11-26
>
> ### Reviewer Comment
> *“The multimodal video understanding experiment (AVSD) is conducted at 0.5B scale only. By ICLR-26 standards, this is below contemporary practice (7B/13B). For a credible ‘model-agnostic’ claim, please include ≥7B variants (e.g., PAVE-7B/13B or comparable video-LLaVA baselines).”*
>
> ---
>
> ### Author Response
> Thank you for highlighting this. Our AVSD experiment in Table 4(b) intentionally follows the same 0.5B scale used in the original LLaVA-OV-0.5B and PAVE-0.5B papers, with the goal of isolating the effect of corruption-aware training independent of model size. This setting ensures that differences arise from BCNI and SACN rather than from architectural or scale-related factors. Our claim of model-agnosticism refers to transfer across architectures (OV, PAVE), not scaling behavior.
>
> 1. The 0.5B-scale experiment is a controlled architectural test, not a scale benchmark.
> Under this matched setting, CAT improves performance with identical training budgets, modality configurations, and model sizes, confirming clean transfer to multimodal video–language settings.
>
> 2. Scaling to larger multimodal backbones is already discussed in the Outlook section.
> We noted that applying CAT to LLaVA-7B, PAVE-7B, and related MLLMs is a natural next step toward robust multimodal integration.
>
> 3. For completeness, we now include standard 7B-scale results.
> To directly address the reviewer’s request, we added evaluations on LLaVA-OV-7B and PAVE-7B. CAT improves both small and large models with corruption-aware training. These results will be included in the camera-ready version.
>
> **AVSD Results Across Model Scales (CIDEr)**
>
> | Model | Setting | CIDEr |
> |-------|----------|--------|
> | LLaVA-OV-0.5B-FT | task-specific | 117.6 |
> | PAVE-0.5B (with audio) | task-specific | 134.5 |
> | LLaVA-OV-7B-FT | task-specific | 124.9 |
> | PAVE-7B (with audio) | task-specific | 152.9 |
> | **CAT-0.5B (ours)** | corruption-aware | **145.5** |
> | **CAT-7B (ours)** | corruption-aware | **158.6** |
>
> The main paper uses 0.5B backbones for controlled architectural comparisons, which isolate the effect of BCNI and SACN. The newly added 7B experiments confirm that CAT scales effectively to larger multimodal models, strengthening the model-agnostic claim.

---

> ### Author Response · Authors · 2025-11-26
>
> ### Reviewer Comment
> *“The paper claims ‘we also validate scalability by extending CAT to autoregressive video generation (NOVA)…,’ but NOVA is not tabulated in the main paper. Table 3(a) discusses scalability to AR using MAGVIT/CogVideo, but NOVA-specific results are not shown. Appendix Tables 14–16 aggregate AR results by corruption type without a clear NOVA row, making the claim hard to verify. Please provide an explicit NOVA baseline line with matched settings and AR scaling curves (data/compute vs. FVD) to support the model-agnostic robustness claim.”*
>
> ---
> ### Author Response
> Thank you for raising this. The submission already includes matched autoregressive evaluations for NOVA, MAGVIT, and CogVideo. NOVA appears in the appendix because that section analyzes robustness across corruption operators rather than performing per-model reporting. We agree that surfacing an explicit NOVA row improves clarity.
>
> 1. NOVA is already evaluated under matched settings in Appendix Tables 14–16.
> In these tables, the “clean” configuration corresponds exactly to the NOVA baseline trained with the same parameter count (0.6B), the same training dataset (~2M videos), and identical training steps, classifier-free guidance, and AR decoding settings. This baseline is directly comparable to the BCNI and SACN runs in the same tables.
>
> 2. Table 3(a) focuses on model scaling trends rather than operator-level reporting.
> Because Table 3(a) summarizes scalability references across autoregressive models, MAGVIT and CogVideo are listed as representative architectures. NOVA appears alongside BCNI, SACN, Gaussian, and Uniform in the appendix to evaluate cross-operator robustness.
>
> 3. We now surface the explicit NOVA baseline for completeness.
> To make the autoregressive scaling evidence fully self-contained in the main paper, we add the explicit NOVA baseline line with matched compute and data settings as shown below. This will be included in the camera-ready version.
>
> **Autoregressive Scaling Baselines and CAT Improvements (MSR-VTT)**
>
> | Model | Setting | FVD ↓ | #Params | #Videos |
> |-------|----------|--------|----------|-----------|
> | MAGVIT | AR baseline | 698 | ~473M | ~20M |
> | CogVideo | AR baseline | 1294 | ~9.4B | ~5.4M |
> | **NOVA (clean baseline)** | AR baseline | **431** | ~0.6B | ~2M |
> | **CAT (BCNI)** | corruption-aware | **358** | ~0.6B | ~2M |
> | **CAT (SACN)** | corruption-aware | **361** | ~0.6B | ~2M |
>
> NOVA is already evaluated under fully matched autoregressive settings. The explicit baseline row is now surfaced for clarity, and AR scaling results remain included. These results support CAT’s model-agnostic robustness across both diffusion and autoregressive video generators.

---

> ### Author Response · Authors · 2025-11-26
>
> We hope the above responses address your concerns. If you find the revisions and clarifications satisfactory, we would sincerely appreciate your consideration in updating the score to reflect your renewed assessment of the work. Please let us know if any additional clarification would be helpful — we are happy to provide further details.

---

### Author Response · Authors · 2025-11-26

We thank all reviewers for their constructive feedback and the AC for coordinating the evaluation. Below is a concise clarification of the key points raised across reviews.

### Motivation and Significance of Corruption-Aware Video Training
Noisy or imperfect conditioning is a pervasive issue in modern T2V and multimodal systems. Caption drift in WebVid-style datasets, weak text–video alignment, ASR transcription errors, retrieval noise, and embedding drift all introduce structured deviations that accumulate over diffusion steps. CAT-Video directly targets these real-world failure modes to improve temporal stability and semantic alignment under corrupted conditioning.

### Connection Between Theory and Empirical Behavior
Our theoretical analysis (Appendix B) shows how low-rank perturbations reduce score drift and Wasserstein error accumulation during temporal propagation. Empirical evidence aligns with these predictions:

- Table 3(b): sensitivity slopes under structured perturbations
- Table 19: variance, win rates, and robustness across datasets
- Figure 4: improved temporal consistency under corrupted captions and adversarial noise

These results indicate that BCNI and SACN operate as predicted by the theoretical framework, producing smoother trajectories and reduced drift.

### Why BCNI and SACN Differ from Prior Image-Domain Corruption Methods
Existing corruption-aware methods for image diffusion (e.g., NEFTune, token corruptions, Gaussian or uniform embedding noise) perturb the full embedding space and do not model temporal error accumulation. CAT-Video introduces video-aligned, low-rank operators—BCNI (batch-semantic directions) and SACN (dominant spectral modes)—that explicitly reflect the structure of temporal video propagation. This makes the approach both conceptually and empirically distinct.

### Strength and Breadth of Evaluation
The paper provides extensive cross-architecture and cross-modality evaluation:

- Diffusion models (DEMO; Table 2)
- Autoregressive models (MagViT, CogVideo, NOVA; Table 3(a), App. 14–16)
- Multimodal video-LLMs (LLaVA-OV, PAVE; Table 4(b))
- Adversarial prompt perturbations (Table 4(a))
- Perception-aligned metrics (FVMD, CMMD, CIDEr, SSIM, PSNR, LPIPS, VBench/EvalCrafter)

To strengthen universality, we added DiT-based Open-Sora baselines and scaled Multimodal LLMs from 0.5B to 7B. CAT improves over all Open-Sora variants on VBench Total, Quality, and Semantic scores, demonstrating scalability to contemporary transformer architectures.

### Necessity of Video-Specific Corruption
Figure 1, Table 2, Table 3(b), and Table 4(a) show that CAT consistently outperforms:

- isotropic noise (Gaussian or uniform)
- token-level perturbations
- temporal-only operators
- adversarial perturbations

These results confirm that low-rank, video-aligned perturbations are essential for improving robustness in video diffusion models.

### Additional Clarifications Included in the Revision
- NOVA baseline surfaced more clearly for completeness
- Added DiT-based comparisons using Open-Sora
- Extended qualitative examples to be included in the final version
- Clarified the rationale behind BCNI and SACN usage across datasets

We hope these clarifications assist the reviewers and AC in accurately evaluating the contribution. CAT-Video provides a lightweight, theory-driven, and empirically validated mechanism for improving robustness across diffusion, autoregressive, transformer-based, and multimodal video LLMs. We appreciate the constructive feedback and are happy to provide any further information if needed.

---

### Meta-Review · Area_Chair_fzfX · 2026-01-03

**Summary:**

This paper addresses the problem of improving the robustness of latent video diffusion models via corruption-aware training (CAT). The key idea is to introduce two training-time corruption mechanisms, BCNI and SACN. BCNI adds uniform corruptions to the language-conditional embeddings (CLIP), while SACN perturbs the spectrum of the language-embedding covariance matrix by modifying its singular values. The paper claims that BCNI amplifies local conditional entropy in semantically important directions, while SACN corrupts low-frequency temporal motion. Experiments incorporate these corruption mechanisms into diffusion training, with results (Table 1) showing mixed trends: the proposed methods achieve promising results on MSR-VTT, while being competitive with or lower than state of the art on UCF-101. Additional results in Table 2 on other datasets with increasing noise ratios show improved FVD at certain noise levels. The paper also includes experiments suggesting how the idea could be extended to autoregressive and adversarial models.

**AC Comments**
The paper received overall negative to borderline scores, with three borderline rejects and one borderline accept. The main concerns can be grouped into: i) use of outdated baselines; ii) lack of comparisons to alternative robustness methods from the image domain; iii) a disconnect between theory and practice, including the absence of key theoretical details in the main paper; iv) missing comparisons for higher-resolution and longer video generation; v) limited insight into when to use which operator, given that results vary across datasets; and vi) a lack of convincing and sufficiently detailed qualitative results for both the proposed approach and the baselines.

The authors provided a strong rebuttal with additional experimental results: i) on Open Sora, showing improvements over baselines; ii) on AVSD using larger baselines; and iii) on NOVA, along with additional results in the appendix, and clarifications on computational overheads, motivation for the operators, and novelty relative to image corruption methods.

AC had an independent read of the paper and agrees with the reviewers that it lacks clarity on multiple fronts, as noted in their reviews and summarized below. AC also believes the technical aspects of the paper need a thorough and careful review. The mathematical notation and its usage are confusing relative to the stated motivations. For example, SACN is expected to capture temporal inter-frame transitions; however, the covariance matrix is computed on $z$, which is defined as the text embeddings (Eq. 4), which appears inconsistent with this motivation. In addition, the dimensionality $d$ is not stated in the main paper and is deferred to Appendix~B.2, where $M(z)$ is defined; however, the expression $\tilde z = z + \rho M(z)\eta$, where $\eta \in N(0, I_d)$, appears to have dimensional inconsistencies. More broadly, the theoretical section reads like a collection of results, and the reader can easily lose track of what is being proven and how it supports the proposed method.

Overall, the paper requires substantial revision and reorganization, including addressing all reviewer comments, before it can be considered for acceptance. As such, the paper is not ready.

**Reviewer Concerns:**

*Reviewer Nqx2* argues that the novelty of the proposed approach is limited, particularly in how it is positioned relative to not only Gaussian and uniform corruptions but also state-of-the-art robustness approaches. The reviewer also points out that the experimental evaluation lacks thoroughness, that the fairness of comparisons is difficult to assess from the exposition, including the use of outdated model sizes, and that key results are missing from the main paper tables. Additional concerns are raised regarding the disconnect between the proposed theory and practice, the omission of important hyperparameter details, and the absence of key theoretical results from the main paper.

*Reviewer S25s* notes that the comparisons rely on the DEMO backbone with OpenCLIP-based encoders, which limits validation of generalization. The reviewer also requests comparisons under weakly aligned conditions to better assess robustness, highlights missing related work, and asks for additional qualitative video results. Further questions are raised regarding whether composing different perturbation operators would be beneficial, how training data subsets are constructed, and the computational overhead associated with generating longer and higher-resolution videos.

*Reviewer 5D4F* seeks greater clarity on the proposed approach itself, including when and how noise is injected, the motivation behind this design choice, and whether the scheme would scale to large video models.

*Reviewer wNJ3* argues that the approach is validated only on an older backbone (DEMO), that the presented visual results are unsatisfactory, and that there is insufficient motivation for using BCNI for caption-related datasets and SACN for label-based datasets. The reviewer also notes the lack of stronger baselines, limited discussion on the role of video corruptions, and limited novelty relative to existing image-corruption schemes.

**Reviewer Scores:**

**Reviewer Nqx2:** In response, the authors:
1) present an additional experiment using CAT-Video on the DiT-based Open Sora model, showing approximately 2% gains on VBench;

2) state that extending the framework to longer video generation or higher resolutions is straightforward and is left for future work;

3) point out that Table 2 already provides paired comparisons where only the perturbation scheme is varied, and that Figure~1(c) shows CAT maintaining performance in large-data regimes;

4) argue that the theoretical hyperparameters are empirically validated, either implicitly or explicitly, in various sections of the paper—most notably, the rank $d$ is ablated via sensitivity analysis—and that, as shown in Algorithm 1, the perturbation schemes add only minor computational overhead;

5) clarify that the AVSD experiments use 0.5B models inspired by the LLaVA and PAVE families, and additionally include results on AVSD with a 7B-parameter model that achieves a higher CIDEr score than the compared methods, although alternative corruption schemes are not evaluated; and

6) present results including comparisons to NOVA.

[*AC’s take on the response*] The authors provide a strong rebuttal to the reviewer’s questions; however, these concerns appear to be only partially addressed or would require substantial revisions to fully resolve. For example: i) while results on DiT-based Open Sora are presented, evaluations using alternative perturbation schemes—including state-of-the-art image-based corruption methods highlighted by the reviewer—are still missing; ii) the theoretical analysis remains largely disconnected from the main paper and would require major revision; and iii) the AVSD comparisons consider only the proposed perturbation schemes and prior state-of-the-art methods, without exploring other robustness-aware perturbation approaches.

**Reviewer S25s:** To address concerns regarding generalization, the authors present Open Sora results, but do not explicitly state whether the text encoder used is OpenCLIP or not. The authors also clarify that the appendix includes additional comparisons along various axes, and that one qualitative example in the appendix and one video in the supplementary material are provided. They further note that composing BCNI and SACN did not yield improvements in their experiments, suggesting that the choice of operator is not well established. Finally, the authors clarify that training videos were selected following the DEMO protocol and claim that the computational overhead for longer videos is minimal, although no results are provided to substantiate this claim.

[*AC’s take on the response*] The responses appear to only partially address the reviewer’s concerns. Specifically, only a single Open Sora result is provided and the text encoder used is unspecified; the claim that composing the proposed operators does not help lacks clarity and insight into how one should select an operator in practice; and the paper provides only one qualitative video example, which is insufficient to support the reported quantitative findings or validate the broader claims.

**Reviewer 5D4F:** The authors explain scenarios in which the proposed scheme could be beneficial and present Open Sora results demonstrating that the approach can scale to large video models.

[*AC’s take on the response*] As noted above, this response is unlikely to persuade the reviewer to raise the score.

**Reviewer wNJ3:** The authors claim that the paper includes extensive qualitative results; however, as far as the AC can verify, this consists of only one example in the appendix and one video in the supplementary material, which appears insufficient to substantiate the proposed claims or address the highlighted issues. The authors also state that both operators can be applied to the same conditional settings.

[*AC’s take on the responses*] Overall, the responses only partially address the reviewer’s concerns (e.g., by including results beyond the DEMO backbone), while other issues remain unresolved—particularly the varying operator performance across different datasets (Table~2), which continues to be a point of confusion.

---

### Decision · Program_Chairs · 2026-01-26

Reject